# Whole genome comparison of a large collection of mycobacteriophages reveals a continuum of phage genetic diversity

Welkin H Pope[1†], Charles A Bowman[1†], Daniel A Russell[1†], Deborah Jacobs-Sera[1], David J Asai[2], Steven G Cresawn[3], William R Jacobs Jr[4], Roger W Hendrix[1], Jeffrey G Lawrence[1], Graham F Hatfull[1]*, Science Education Alliance Phage Hunters Advancing Genomics and Evolutionary Science, Phage Hunters Integrating Research and Education, Mycobacterial Genetics Course

[1]Department of Biological Sciences, University of Pittsburgh, Pittsburgh, United States; [2]Howard Hughes Medical Institute, Chevy Chase, United States; [3]Department of Biology, James Madison University, Harrisonburg, United States; [4]Department of Microbiology and Immunology, Albert Einstein College of Medicine, Bronx, United States

**Abstract** The bacteriophage population is large, dynamic, ancient, and genetically diverse. Limited genomic information shows that phage genomes are mosaic, and the genetic architecture of phage populations remains ill-defined. To understand the population structure of phages infecting a single host strain, we isolated, sequenced, and compared 627 phages of *Mycobacterium smegmatis*. Their genetic diversity is considerable, and there are 28 distinct genomic types (clusters) with related nucleotide sequences. However, amino acid sequence comparisons show pervasive genomic mosaicism, and quantification of inter-cluster and intra-cluster relatedness reveals a continuum of genetic diversity, albeit with uneven representation of different phages. Furthermore, rarefaction analysis shows that the mycobacteriophage population is not closed, and there is a constant influx of genes from other sources. Phage isolation and analysis was performed by a large consortium of academic institutions, illustrating the substantial benefits of a disseminated, structured program involving large numbers of freshman undergraduates in scientific discovery.

*For correspondence: gfh@pitt.edu

†These authors contributed equally to this work

**Competing interests:** The authors declare that no competing interests exist.

## Introduction

Bacteriophages are the dark matter of the biological universe, forming a vast, ancient, dynamic, and genetically diverse population, replete with genes of unknown function (*Pedulla et al., 2003*). Phages are the most abundant organisms in the biosphere, and the ~$10^{31}$ tailed phage particles participate in ~$10^{23}$ infections per second on a global scale, with the entire population turning over every few days (*Suttle, 2007*). The population is not only vast and dynamic, but comparisons of virion structures suggest that it is also extremely old (*Krupovic and Bamford, 2010*). It is thus not surprising that bacteriophages are genetically highly diverse, although their comparative genomics has lagged behind that of other microbes, largely due to the lack of individual isolates for genomic analyses (*Hatfull and Hendrix, 2011*). To date, there are approximately 2000 completely sequenced bacteriophage genomes in the GenBank database, a small number relative to the more than 30,000 sequenced prokaryotic genomes (http://www.ncbi.nlm.nih.gov/genome/browse/), in spite of phage genomes being only 1–5% of the size of their host genomes.

Double-stranded DNA tailed phages are proposed to have evolved with common ancestry but with different phages having differential access to a large common gene pool (*Hendrix et al., 1999*).

**eLife digest** Viruses are unable to replicate independently. To generate copies of itself, a virus must instead invade a target cell and commandeer that cell's replication machinery. Different viruses are able to invade different types of cell, and a group of viruses known as bacteriophages (or phages for short) replicate within bacteria. The enormous number and diversity of phages in the world means that they play an important role in virtually every ecosystem.

Despite their importance, relatively little is known about how different phage populations are related to each other and how they evolved. Many phages contain their genetic information in the form of strands of DNA. Using genetic sequencing to find out where and how different genes are encoded in the DNA can reveal information about how different viruses are related to each other. These relationships are particularly complicated in phages, as they can exchange genes with other viruses and microbes.

Previous studies comparing the genomes—the complete DNA sequence—of reasonably small numbers of phages that infect the *Mycobacterium* group of bacteria have found that the phages can be sorted into 'clusters' based on similarities in their genes and where these are encoded in their DNA. However, the number of phages investigated so far has been too small to conclude how different clusters are related. Are the clusters separate, or do they form a 'continuum' with different genes and DNA sequences shared between different clusters?

Here, Pope, Bowman, Russell et al. compare the individual genomes of 627 bacteriophages that infect the bacterial species *Mycobacterium smegmatis*. This is by far the largest number of phage genomes analyzed from a single host species. The large number of genomes analyzed allowed a much clearer understanding of the complexity and diversity of these phages to be obtained. The isolation, sequencing and analysis of the hundreds of *M. smegmatis* bacteriophage genomes was performed by an integrated research and education program, called the Science Education Alliance Phage Hunters Advancing Genomics and Evolutionary Science (SEA-PHAGES) program. This enabled thousands of undergraduate students from different institutions to contribute to the phage discovery and sequencing project, and co-author the report. SEA-PHAGES therefore shows that it is possible to successfully incorporate genuine scientific research into an undergraduate course, and that doing so can benefit both the students and researchers involved.

The results show that while the genomes could be categorized into 28 clusters, the genomes are not completely unrelated. Instead, a spread of diversity is seen, as genes and groups of genes are shared between different clusters. Pope, Bowman, Russell et al. further reveal that the phage population is in a constant state of change, and continuously acquires genes from other microorganisms and viruses.

Phage genomes are typified by their mosaic architectures generated by gene loss and gain through horizontal genetic exchange; however, the parameters influencing access to the common gene pool are numerous and likely include host range, genome size, replication mode, and life style (temperate vs lytic). Migration to new hosts is probably common, but is affected by local host diversity and mutation rates, as well as resistance mechanisms such as receptor availability, restriction, CRISPRs, and abortive infection systems (*Buckling and Brockhurst, 2012*; *Jacobs-Sera et al., 2012*; *Hoskisson et al., 2015*). Constraints on gene acquisition may also be imposed by synteny—particularly among virion structural genes—and by size limits of DNA packaging (*Juhala et al., 2000*; *Hatfull and Hendrix, 2011*).

We have previously described comparative analyses of modest numbers of mycobacteriophages and shown that they can be sorted by nucleotide sequence and gene content comparisons into groups of closely related genomes referred to as 'clusters' (designated Cluster A, B, C, etc.); phages without any close relatives are referred to as 'singletons'. Some of the clusters can be further divided into subclusters (e.g., Subcluster A1, A2, A3, etc.) according to nucleotide sequence relatedness (*Pedulla et al., 2003*; *Hatfull et al., 2006*, *2010*; *Pope et al., 2011b*). The genomes are mosaic whereby individual phages are constructed as assemblages of modules, many of which are single genes (*Pedulla et al., 2003*). Each mycobacteriophage cluster has features particular to that cluster (e.g., regulatory systems, repeated sequences, tRNA genes, etc. [*Pope et al., 2011a*, *2011b*, *2013*, *2014a*, *2014b*]), but

because of the pervasive mosaicism, the relationships among phages within clusters and between clusters are complex. Collections of phages have been isolated on other hosts such as *Bacillus* spp., *Escherichia coli*, *Pseudomonas* spp., *Propionibacterium* spp. and *Staphylococcus* spp. (*Kwan et al., 2005*, *2006*; *Kropinski et al., 2007*; *Marinelli et al., 2012*; *Hatfull et al., 2013*; *Grose and Casjens, 2014*; *Grose et al., 2014*; *Lee et al., 2014*) and these can be similarly divided into clusters based on DNA similarity. Recent analysis of 337 phages infecting 31 bacterial species within the *Enterobacteriaceae* (*Grose and Casjens, 2014*) reveals 56 clusters of phage genomes. It is thus clear that there is substantial diversity within the phage population, even when comparing phages of a common host and which are expected to be in direct genetic contact with each other in their natural environment (*Hatfull and Hendrix, 2011*). Nonetheless, the numbers of genomes isolated on a particular host generally are too small to define the nature and the size of the populations at large with any substantial resolution.

Viral metagenomic studies provide valuable insights into phage diversity and population dynamics, but typically generate few complete genome sequences or any specific information relating viral genomes to specific bacterial hosts (*Hambly and Suttle, 2005*; *Rodriguez-Brito et al., 2010*; *Mokili et al., 2012*). A recent analysis of *Synechococcus* phages using metagenomic analysis coupled with viral tagging showed that there are multiple 'populations' of these phages (similar to the clusters described above), but suggested that these represent distinct groups of related phages rather than a continuous spectrum of diversity (*Deng et al., 2014*). This differs from prior predictions that the phage population as a whole likely spans a continuum of diversity—albeit with uneven representation of different groups of related phages—because of genomic mosaicism (*Hendrix, 2003*; *Hatfull, 2010*, *2012*). However, as the *Synechococcus* phage data are derived from a single sample using a single host, it is unclear if this extends to phages of other hosts (*Deng et al., 2014*).

Here we describe the comparative analysis of a large number of completely sequenced mycobacteriophage genomes and demonstrate that they represent a spectrum of diversity and do not constitute discrete populations. Rarefaction analyses of their constituent genes are consistent with populations of gene families shared among mycobacteriophages being augmented by the introduction of new gene families from outside sources. The assembling of a large and highly informative collection of bacteriophages by a consortium of students and faculty at multiple institutions demonstrates that a course-based research experience (CRE) can be successfully implemented at large scale without compromising the authenticity or richness of a scientific investigation imbued with discovery and project ownership.

## Results and discussion

### A genome-by-genome approach to defining phage diversity

Exploring phage diversity using a genome-by-genome approach has notable advantages and some potential disadvantages. The main advantage is that complete genome sequences give information about genome length and composition, providing key insights into genome mosaicism and how genome segments are shared and exchanged. A difficulty is that there are not large extant phage collections available for most bacterial hosts, and isolation, purification, and characterization of phages can be slow and time-consuming. Because isolation typically requires plaque formation and growth in the laboratory, some naturally occurring phages may escape isolation using standard methods. Thus, although the diversity of phages isolated and propagated in the laboratory may not capture all types of phage, it represents a minimum, not a maximum, index of diversity.

### Authentic research in a CRE

The 2012 report from the President's Council of Advisors on Science and Technology (PCAST) focused on the poor retention of undergraduate students in science, technology, engineering and mathematics (STEM) as an impediment to meeting US economic demands (*PCAST, 2012*). One of the PCAST recommendations is to replace traditional introductory laboratory courses with research-based experiences that would inspire freshman students and promote STEM retention. A powerful strategy is to engage students in scientific discovery through CREs. The successful implementation of this strategy depends on (i) identifying research questions that can engage students in contributing genuine advances in scientific knowledge without requiring prior expert

knowledge, and (ii) designing the project so that large numbers of students can participate in a meaningful fashion.

We have previously described the Howard Hughes Medical Institute (HHMI) Science Education Alliance Phage Hunters Advancing Genomics and Evolutionary Science (SEA-PHAGES) program, in which beginning undergraduate students isolate, purify, sequence, annotate, and compare bacteriophages, and have described its educational advantages (*Jordan et al., 2014*). By taking advantage of the massive diversity of the phage population so that each student can isolate a unique phage, the program encourages student ownership of their science. And because the collective discoveries by many students generate new scientific insights, the program creates a scientific community of students engaged in authentic research.

The SEA-PHAGES program has contributed to the growth of the collection of sequenced mycobacteriophages to nearly 700 individual isolates (http://phagesdb.org), of which 627 were selected for a detailed analysis (*Supplementary file 1*). This is by far the largest collection of sequenced phage genomes for any single host and thus promises to substantially advance our understanding of phage diversity. The phages were isolated using either direct plating or by enrichment using *Mycobacterium smegmatis* mc$^2$155 as a host, and sequenced using next-generation approaches (see 'Materials and methods'). More than 5000 students—primarily freshmen—at 74 institutions have been involved since inception of the SEA-PHAGES program in 2008, and the phages isolated represent a broad geographical distribution (*Figure 1*) and a variety of viral morphotypes (http://phagesdb.org). The new insights gained from comparative genomic analyses of these phages—as described below—demonstrate the effectiveness of viral discovery and genomics as a model for CRE development.

## Assembling mycobacteriophages into clusters and subclusters

Using previously reported parameters based primarily on nucleotide sequence similarity spanning >50% genome length (*Hatfull et al., 2006*), the 627 genomes were assembled into 20 clusters (A–T) and eight singletons (with no close relatives) (*Figure 2*, *Supplementary file 1*); 11 clusters were subdivided into 2 to 11 subclusters (*Table 1*). There is considerable variation in cluster size with substantial differences in the numbers of genomes in each cluster (2–232), but there is relatively little variation in either genome length or the numbers of genes per genome in any given cluster (*Table 1*). Cluster assignment is of practical utility and is generally robust, with clustered phages typically sharing genome architectures, as noted for the *Enterobacteriacea* (*Grose and Casjens, 2014*). For example, Cluster A phages are similar in size and transcriptional organization, and share an unusual immunity system (*Brown et al., 1997*; *Pope et al., 2011b*). Cluster M phages all contain large numbers of tRNA genes (*Pope et al., 2014a*), Cluster K (*Pope et al., 2011a*) and Cluster O (*Cresawn et al., 2015*) phages have different but characteristic repeated sequences, and Cluster J phages have an unusual capsid with a triangulation (T) number of 13 (*Pope et al., 2013*). Therefore, the organization of related mycobacteriophages into clusters provides a framework for identifying and interpreting gene trafficking within and among potentially distinct groups of genomes.

## Gene content relationships among sequenced mycobacteriophages

Genome mosaicism is more apparent from comparison of gene product amino acid sequences than nucleotide sequence comparisons because of the accumulation of genome rearrangements over a longer period of evolution, during which indications of DNA similarity are lost. To compare mycobacteriophage gene contents we grouped related genes into protein families ('phamilies' or 'phams') using Phamerator (*Cresawn et al., 2011*), which we modified to use kClust (*Hauser et al., 2013*) so as to easily accommodate the large numbers of comparisons. The 69,633 genes assembled into 5205 phams of which 1613 (31%) are orphams (single-gene phamilies [*Hatfull et al., 2010*]). Approximately 25% of phams can be assigned functions in viral structure and assembly, DNA metabolism, integration, lysis, and regulation, but the vast majority are of unknown function. Representation of gene content relationships among all 627 phages as a network phylogeny reveals relationships that are in accord with the cluster and subcluster designations derived from nucleotide sequence comparisons (*Figure 3*). The multiple branches between clusters/subclusters reflect the phylogenetic complexities that arise from genome mosaicism, where genes within a genome have distinct evolutionary histories.

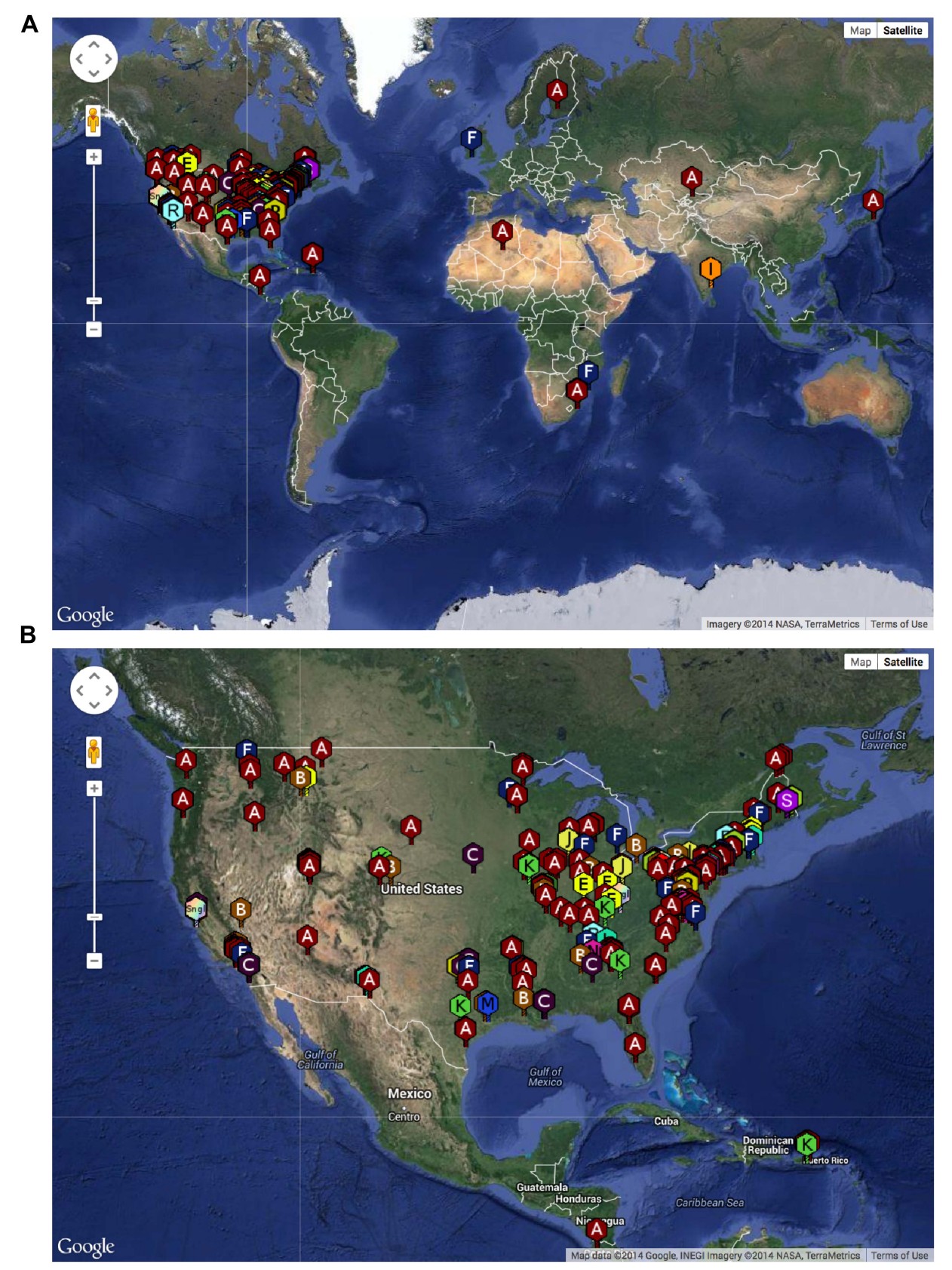

**Figure 1**. Geographical distribution of sequenced mycobacteriophages. (**A**) Locations of sequenced mycobacteriophages across the globe. (**B**) Locations

*Figure 1. continued on next page*

Figure 1. Continued
of sequenced mycobacteriophages across the United States. Colors and letter designations on the isolates refer to the cluster to which the genomes belong. Data from www.phagesdb.org.

The distribution of orphams (genes without mycobacteriophage homologues) provides additional support for cluster/subcluster assignments; *Figure 4*). A relatively high proportion of orphams is a characteristic of both singleton genomes and single-genome subclusters (*Figure 4*). At least 30% of genes in all of the singleton genomes are orphams, and the single-genome subclusters have a minimum of 15% orphams; genomes in other clusters and subclusters typically have fewer than 10% orphams (*Figure 4*). The presence of numerous orphams ensures that the lack of cluster inclusion did not result from sequence errors or insufficient or inappropriate gene annotation. Notable exceptions are Predator (Subcluster H1) and Mendokysei (Cluster T), both of which are in very small clusters/subclusters, and KayaCho (Subcluster B4). KayaCho may warrant separation into a new subcluster (e.g., B6), but overall the orpham distribution is consistent with the cluster/subcluster designations.

## The diversity of different clusters is highly varied

To determine the extent to which the various clusters/subclusters represent discrete groups, we generated a heat map showing pairwise shared gene content (*Figure 5*) and quantified the cluster/subcluster diversity (*Table 1*, *Figure 6*). The heat map strikingly illustrates that diversity is non-uniform, with genomes in some clusters (e.g., Subclusters B1, C1) being very closely related, whereas in others they display substantial differences (e.g., Subclusters A1, F1). The variation is also evident within the large Cluster A group, with some subclusters having low diversity (e.g., A4, A5, A6), some being highly diverse (e.g., A1, A2), and some plausibly further splitting into subgroups (A3) (*Figure 5*).

We quantified the cluster diversity using three different measures, CLuster Average Shared Phamilies (CLASP), Cluster Associated Phamilies (CAP), and Cluster Cohesion Index (CCI) (*Tables 1, 2*, *Figure 6A*). Both CAP (the number of phams present in *all* genomes within a cluster divided by the average number of genes per genome) and CCI (the average number of genes per genome as a percentage of the total number of phams in that cluster) show substantial variation between clusters (*Table 1*, S2), and little evidence for commonly conserved 'core genes', as suggested for T4-related phages (*Petrov et al., 2010*). However, both of these parameters are somewhat influenced by cluster/subcluster size, which varies from cluster to cluster. In contrast, CLASP (the percentage of phamilies shared between two genomes, then averaged across all possible pairs within a cluster or subcluster) is relatively insensitive to cluster/subcluster size (as seen by a resampling analysis; *Figure 6—figure supplement 1*), but still shows substantial variation from one cluster to another (*Table 1*, *Figure 6A*).

## The discreteness of different clusters is highly varied

The heat map of genome comparisons (*Figure 5*) also illustrates the degrees to which clusters and subclusters share gene content, a reflection of cluster discreteness, or how isolated discrete clusters are from each other. For example, although the Cluster A phages are highly diverse, they also appear relatively isolated and share relatively few genes with other clusters (*Figure 5*). In contrast, phages in Cluster E share substantial numbers of genes with other clusters, including those in Clusters F, J, L, P, and several singletons. We have quantified these relationships with the Cluster Isolation Index (CII, the percentage of phams present within a cluster that are not present in other mycobacteriophage genomes), which demonstrates the considerable variation in isolation from phages of other clusters/subclusters (*Table 1*, *Figure 6B*). For example, at one extreme, 84.6% of Cluster C gene phamilies are found only in Cluster C and not elsewhere. At the other extreme, only 23.8% of Cluster I gene phamilies are constrained to that cluster, with the remainder having relatives present in genomes in other clusters. Other clusters form a spectrum of relationships between these extremes (*Table 1*, *Figure 6B*), and clusters such as I and P—which share recognizable DNA sequence similarity (*Figure 2—figure supplement 1*)—share >60% of their genes with other phages (low CII values; *Table 1*). Thus, although some clusters could be considered as discrete groups—as reported for the

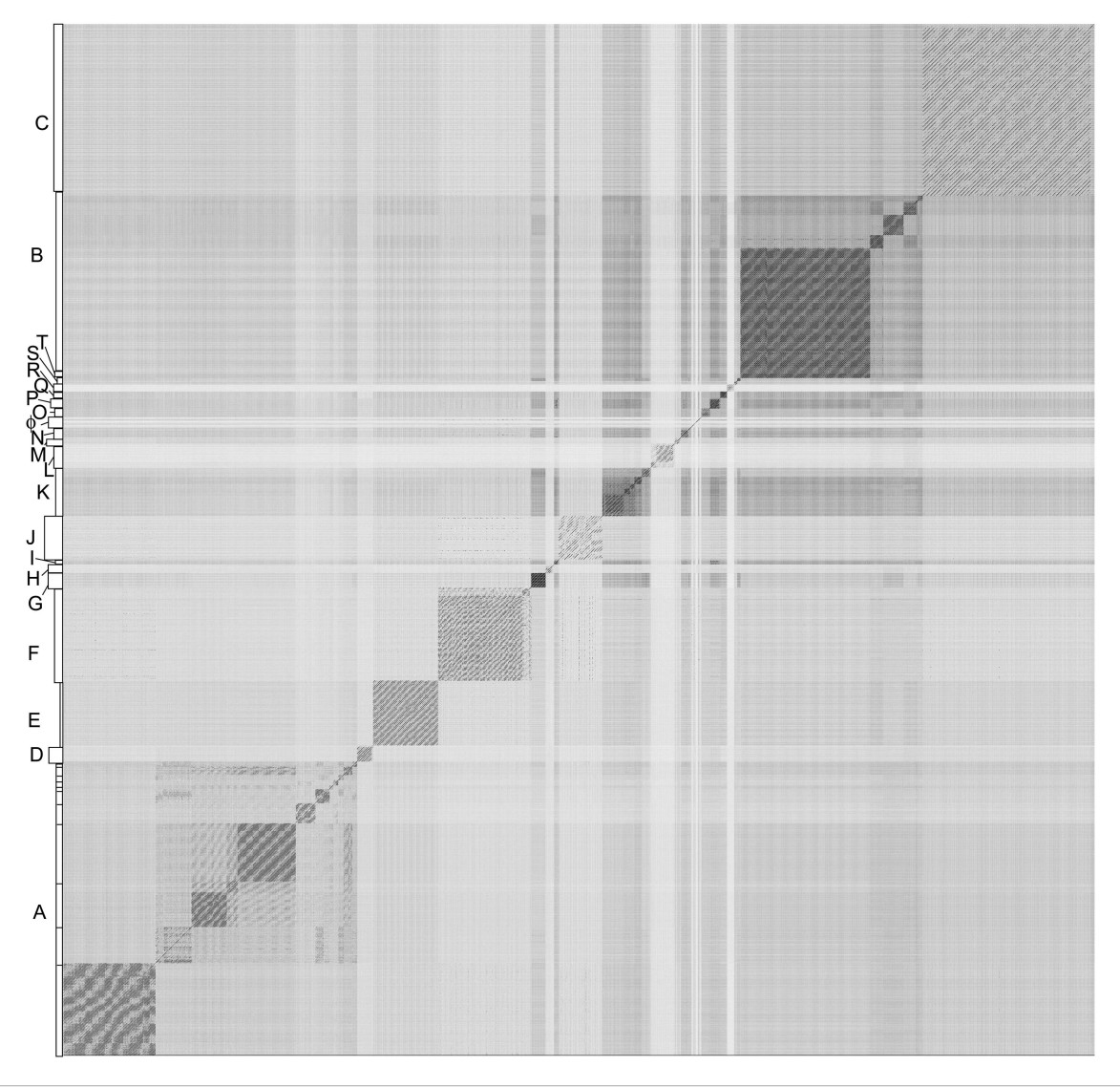

**Figure 2**. Nucleotide sequence comparison of 627 mycobacteriophages displayed as a dotplot. Complete genome sequences of 627 mycobacteriophages were concatenated into a single file which was compared with itself using Gepard (*Krumsiek et al., 2007*) and displayed as a dotplot using default parameters (word length, 10). The order of the genomes is as listed in *Supplementary file 1*. Nucleotide similarity is a primary component in assembling phages into clusters, which typically requires evident DNA similarity spanning more than 50% of the genome lengths.

The following source data and figure supplements are available for figure 2:

**Source data 1**. Concatenated DNA sequences for 627 phage genomes.

**Figure supplement 1**. Dotplot of phages in Clusters I, N, P and the singleton Sparky.

**Figure supplement 2**. Dotplot of Carcharodon, Che9c, Kheth, and Dori.

**Figure supplement 3**. Dotplot of Corndog, Brujita, SG4, Yoshi, and MooMoo.

*Synechococcus* phages (*Deng et al., 2014*)—this is far from being a universal or characteristic feature of groups of related phages.

Cluster isolation analyses reveal additional complexities arising from highly mosaic genomes. For example, the singleton Dori is clearly related to Cluster B phages (*Figure 3*) with which it shares

**Table 1.** Diversity and genetic isolation of mycobacteriophage genome clusters

| Cluster | # Subclusters | # Genomes | Average # genes* | Average length (bp) | Total phams† | Total genes | CLASP‡ | CAP§ | CCI# | CII¶ |
|---|---|---|---|---|---|---|---|---|---|---|
| A | 11 | 232 | 90 ± 5.3 | 51,514 | 1085 | 20,880 | 38.3 | 12.4 | 0.08 | 80.2 |
| B | 5 | 109 | 100.4 ± 4.5 | 68,653 | 421 | 10,944 | 66.2 | 23.2 | 0.24 | 81.0 |
| C | 2 | 45 | 231 ± 5.9 | 155,504 | 486 | 10,395 | 89.3 | 29.4 | 0.48 | 84.6 |
| D | 2 | 10 | 89.3 ± 6.4 | 64,965 | 147 | 893 | 88.1 | 64.3 | 0.61 | 71.4 |
| E | 1 | 35 | 141.9 ± 3.4 | 75,526 | 236 | 4967 | 87.2 | 63.8 | 0.60 | 59.3 |
| F | 3 | 66 | 105.3 ± 5.3 | 57,416 | 658 | 6950 | 54.4 | 4.9 | 0.16 | 55.8 |
| G | 1 | 14 | 61.5 ± 1.2 | 41,845 | 72 | 861 | 96.0 | 91.1 | 0.85 | 55.6 |
| H | 2 | 5 | 98.4 ± 5.7 | 69,469 | 207 | 492 | 61.6 | 31.5 | 0.48 | 67.6 |
| I | 2 | 4 | 78 ± 3.7 | 49,954 | 147 | 312 | 58.9 | 35.0 | 0.53 | 23.8 |
| J | 1 | 16 | 239.8 ± 9.3 | 110,332 | 530 | 3776 | 70.8 | 40.1 | 0.45 | 58.5 |
| K | 5 | 32 | 95.7 ± 4.6 | 59,720 | 411 | 3069 | 51.8 | 20.0 | 0.23 | 73.5 |
| L | 3 | 13 | 127.9 ± 6.5 | 75,177 | 246 | 1663 | 78.2 | 50.8 | 0.52 | 72.4 |
| M | 2 | 3 | 141 ± 8.8 | 81,636 | 201 | 423 | 73.5 | 63.0 | 0.70 | 69.2 |
| N | 1 | 7 | 69.1 ± 2.2 | 42,888 | 152 | 484 | 64.1 | 45.6 | 0.45 | 40.8 |
| O | 1 | 5 | 124.2 ± 3.1 | 70,651 | 151 | 621 | 90.6 | 83.3 | 0.82 | 64.2 |
| P | 2 | 9 | 78.8 ± 2.1 | 47,668 | 159 | 709 | 76.1 | 42.3 | 0.50 | 34.0 |
| Q | 1 | 5 | 85.2 ± 3.7 | 53,755 | 90 | 426 | 96.6 | 90.4 | 0.95 | 73.3 |
| R | 1 | 4 | 101.5 ± 2.5 | 71,348 | 117 | 406 | 91.4 | 84.8 | 0.87 | 71.8 |
| S | 1 | 2 | 109 ± 2.0 | 65,172 | 117 | 218 | 91.7 | 91.7 | 0.93 | 70.9 |
| T | 1 | 3 | 66.7 ± 2.4 | 42,833 | 83 | 200 | 86.1 | 82.5 | 0.80 | 62.7 |
| Dori | 1 | 1 | 94 | 64,613 | 94 | 94 | N/A | N/A | N/A | 35.8 |
| DS6A | 1 | 1 | 97 | 60,588 | 96 | 97 | N/A | N/A | N/A | 58.3 |
| Gaia | 1 | 1 | 194 | 90,460 | 193 | 194 | N/A | N/A | N/A | 58.0 |
| MooMoo | 1 | 1 | 98 | 55,178 | 98 | 98 | N/A | N/A | N/A | 31.6 |
| Muddy | 1 | 1 | 71 | 48,228 | 70 | 71 | N/A | N/A | N/A | 71.4 |
| Patience | 1 | 1 | 109 | 70,506 | 109 | 109 | N/A | N/A | N/A | 57.8 |
| Sparky | 1 | 1 | 93 | 63,334 | 93 | 93 | N/A | N/A | N/A | 48.4 |
| Wildcat | 1 | 1 | 148 | 78,296 | 148 | 148 | N/A | N/A | N/A | 69.6 |

*Average number of protein-coding genes per genome, with standard deviation.

†Total phams is the sum of all phamilies (groups of homologous mycobacteriophage genes) in that cluster.

‡The Cluster Averaged Shared Phamilies (CLASP) index is the average of the percentages of phamilies shared pairwise between genomes within a cluster.

§The Cluster-Associated Phamilies (CAP) index is the percentage of the average number of phamilies per genome within a cluster whose phamilies are present in every cluster member.

#The Cluster Cohesion Index (CCI) is generated by dividing the average number of genes per genome by the total number of phamilies (phams) in that cluster.

¶The Cluster Isolation Index (CII) is the percentage of phams that are present only in that cluster, and not present in other mycobacteriophages.

N/A: Not applicable.

limited DNA similarity (*Figure 2—figure supplement 2*) with 20–26% of its genes (*Figure 4—figure supplement 1*), but also has nucleotide similarity and shares genes with Cluster N and I2 phages, among others (*Figure 2—figure supplement 2*, *Figure 4—figure supplement 1*), as reflected in its low CII (*Table 1*, *Figure 6B*). Likewise, the singleton MooMoo has segments of DNA similarity and shares ~20% of its gene content (as determined by shared phams) with Cluster F phages (*Figure 3*, *Figure 2—figure supplement 3*, *Figure 4—figure supplement 1*), but also has similarity to Clusters N and I, as well as a low CII (*Table 1*, *Figure 6B*). It has low DNA similarity to Cluster O (*Figure 2—figure supplement 3*), but has several phams in common with the Cluster O phages, and

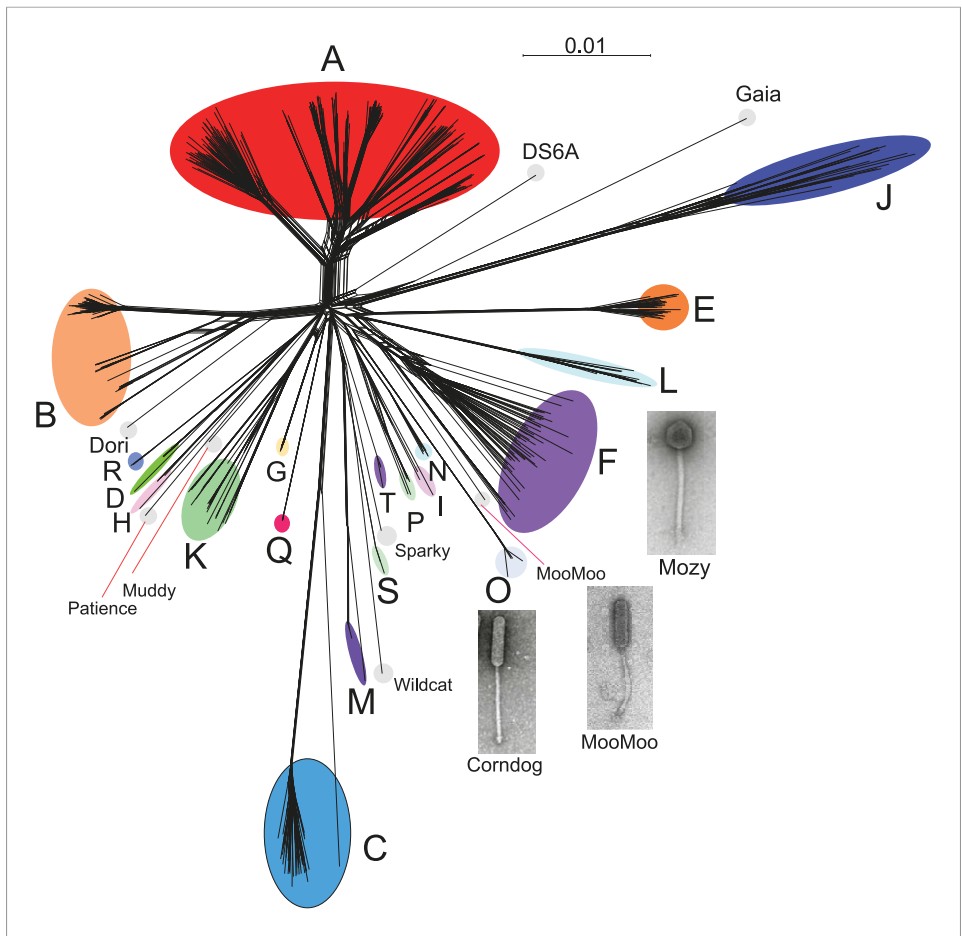

**Figure 3**. Network phylogeny of 627 mycobacteriophages based on gene content. Genomes of 627 mycobacteriophages were compared according to shared gene content using the Phamerator (*Cresawn et al., 2011*) database Mykobacteriophage_627, and displayed using SplitsTree (*Huson and Bryant, 2006*). Colored circles indicate grouping of phages labeled according to their cluster designations generated by nucleotide sequence comparison (*Figure 2*); singleton genomes with no close relatives are labeled but not circled. Micrographs show morphotypes of the singleton MooMoo, the Cluster F phage Mozy, and the Cluster O phage Corndog. With the exception of DS6A, all of the phages infect *Mycobacterium smegmatis* mc$^2$155.
The following source data is available for figure 3:

**Source data 1**. Nexus file containing phamily assignments for 627 phage genomes.

has the same unusual prolate morphology (*Figure 3*). Complex relationships are also seen in the singletons Gaia and Sparky (*Figure 4—figure supplement 2*).

Taken together, the analyses of both cluster diversity and cluster isolation show that mycobacteriophage populations contain a continuum of diversity, with non-uniform abundance of different types of phages. The prevalence of isolated phages may not necessarily reflect the proportions of different types of phages in the environment, but the availability of a large collection of isolated phages enables capture and whole genome analysis of relatively rare phages that are critical to understanding the complexities of genome relationships. We recently reported genomic analysis of the singleton mycobacteriophage Patience, which has a substantially lower GC% than its host (50.3% vs 67.4%), has a different codon usage profile, but is undergoing codon selection for growth in a high GC% environment (*Pope et al., 2014b*). If there is a flux of phage genomes and genes entering the mycobacterial neighborhood, then we predict that the phages of a single host do not reflect a closed system with discrete populations, but one that is open with ever-expanding diversity.

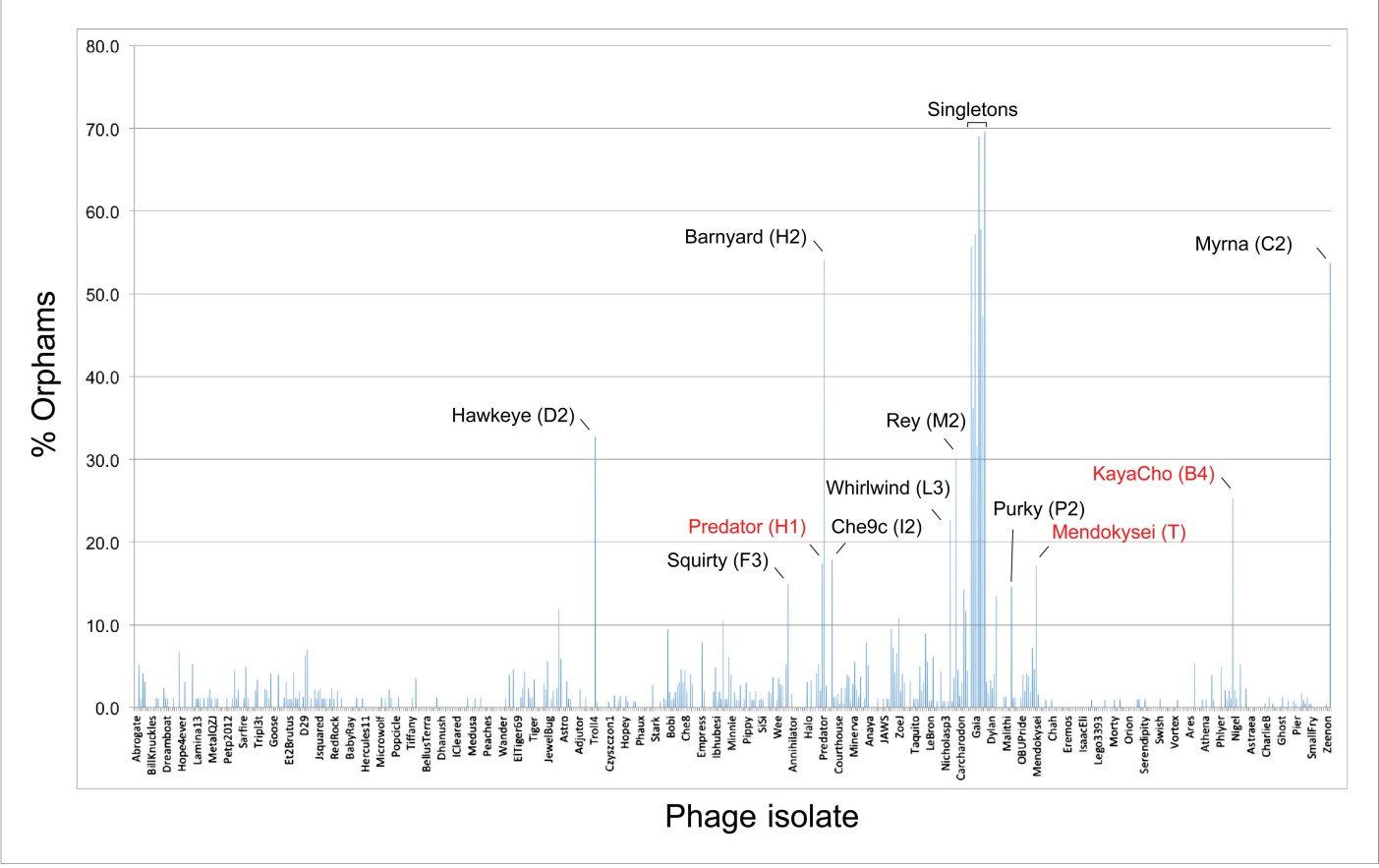

**Figure 4**. Proportions of orphams in mycobacteriophage genomes. The proportions of genes that are orphams (i.e., single-gene phamilies with no homologues within the mycobacteriophage dataset) are shown for each phage. The order of the phages is as shown in *Supplementary file 1*. All of the singleton genomes have >30% orphams, and most of the other genomes with relatively high proportions of orphams are the single-genome subclusters (*Table 2*) including Hawkeye (D2), Myrna (C2), Squirty (F3), Barnyard (H2), Che9c (I2), Whirlwind (L3), Rey (M2), and Purky (P2). Three phages shown in red type are not singletons or single-genome subclusters but have relatively high proportions of orphams. Predator and Mendokysei are members of the diverse and small clusters (five or fewer genomes) H and T, respectively; KayaCho is a member of Subcluster B4 but has a sufficiently high proportion of orphams to arguably warrant formation of a new subcluster, B6.

The following source data and figure supplements are available for figure 4:

**Source data 1**. Pham table containing phamily designations for 627 phage genomes.

**Figure supplement 1**. Shared gene content between Dori, MooMoo, and other mycobacteriophages.

**Figure supplement 2**. Shared gene content between Gaia, Sparky, and other mycobacteriophages.

## The mycobacteriophage population is not a closed system

Both the huge diversity of phamilies in mycobacteriophages and the high frequency of orphams suggest that genes are constantly added to phage genomes from outside sources just as genes are added to the genomes of their bacterial hosts via horizontal gene transfer. Such gene influx—for example, from host-jumping phages such as Patience (*Pope et al., 2014b*)—would provide genetic novelty and enable phages to adapt to their ever-changing hosts. To examine gene flux into the mycobacteriophage population, we performed a rarefaction analysis by re-sampling the gene phamilies within the phage population (*Figure 7*). Remarkably, the rarefaction curves of the entire collection—including the 95% confidence limits—do not fit a hyperbola as would be expected if the mycobacteriophages were limited to an isolated set of genes, and about 2.5 new gene phamilies

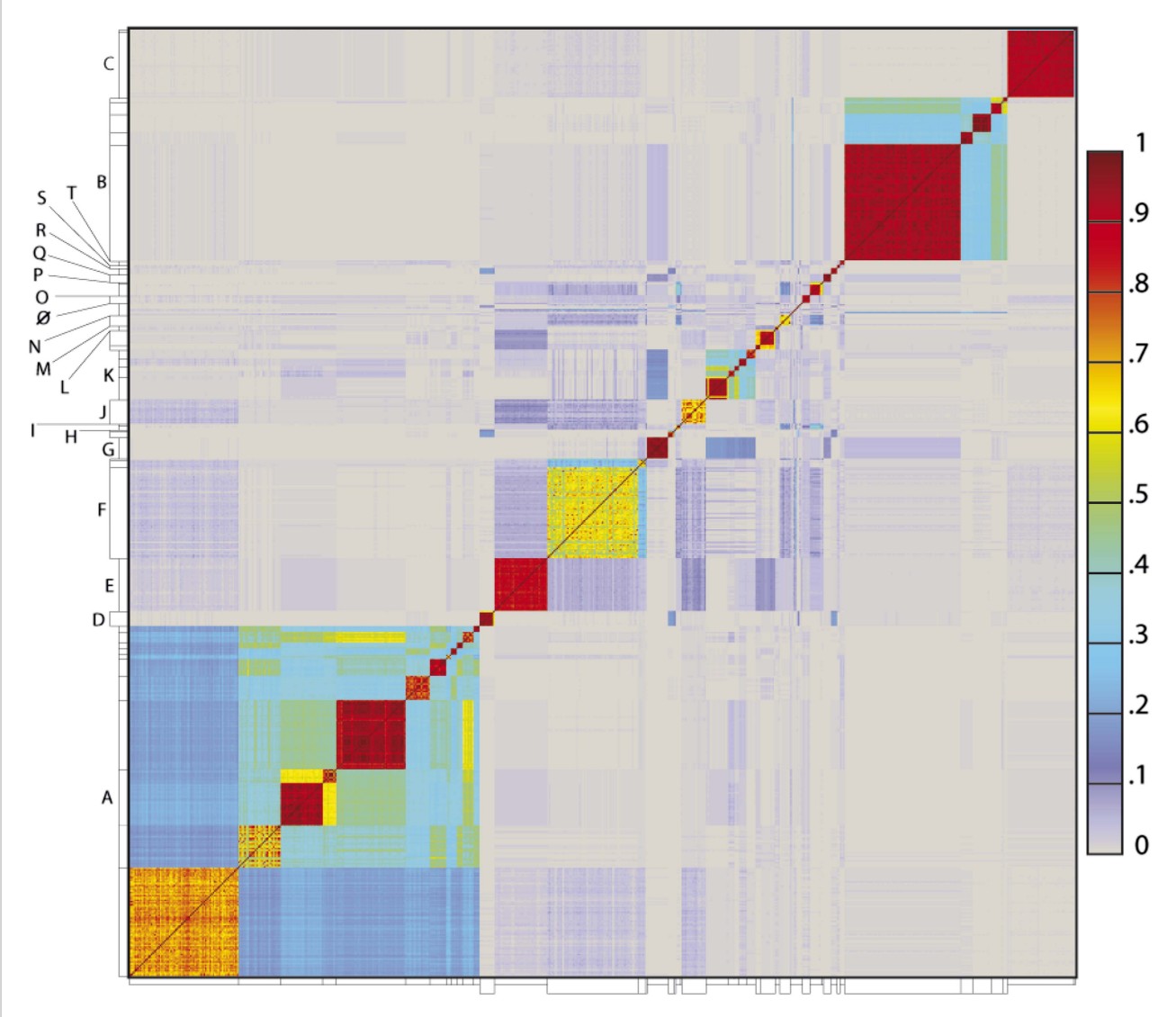

**Figure 5**. Heat map representation of shared gene content among 627 mycobacteriophages. The percentages of pairwise shared genes was determined using a Phamerator (*Cresawn et al., 2011*) database (Mykobacteriophage_627) populated with 627 completely sequenced phage genomes. The 69,574 genes were assembled into 5205 phamilies (phams) of related sequences using kClust, and the average proportions of shared phams calculated. Genomes are ordered on both axes according to their cluster and subcluster designations (*Supplementary file 1*) determined by nucleotide sequence similarities (*Figure 2*). The values (proportions of pairwise shared phams averaged between each partner) are colored as indicated.

The following source data is available for figure 5:

**Source data 1**. Dataset showing percentages of pairwise shared phamilies.

are predicted to be identified with each newly isolated phage (*Figure 7A*). Similar independent analyses on the phages of Cluster A or the phages of Cluster B show that this is also observed within these clusters (*Figure 7B,C*). Thus both individual clusters and the collection as a whole are not genetically fixed, but are in constant flux. While a hyperbola can model sampling of gene phamilies from a finite pool, it does not accommodate the influx of new phamilies. The addition of a linear term (see 'Materials and methods'), representing the introduction of new phamilies from outside sources, results in a non-asymptotic curve which predicts the continual identification of new phams even after large numbers of genomes have been sampled (R > 0.999; *Figure 7D*). This linear term acts as

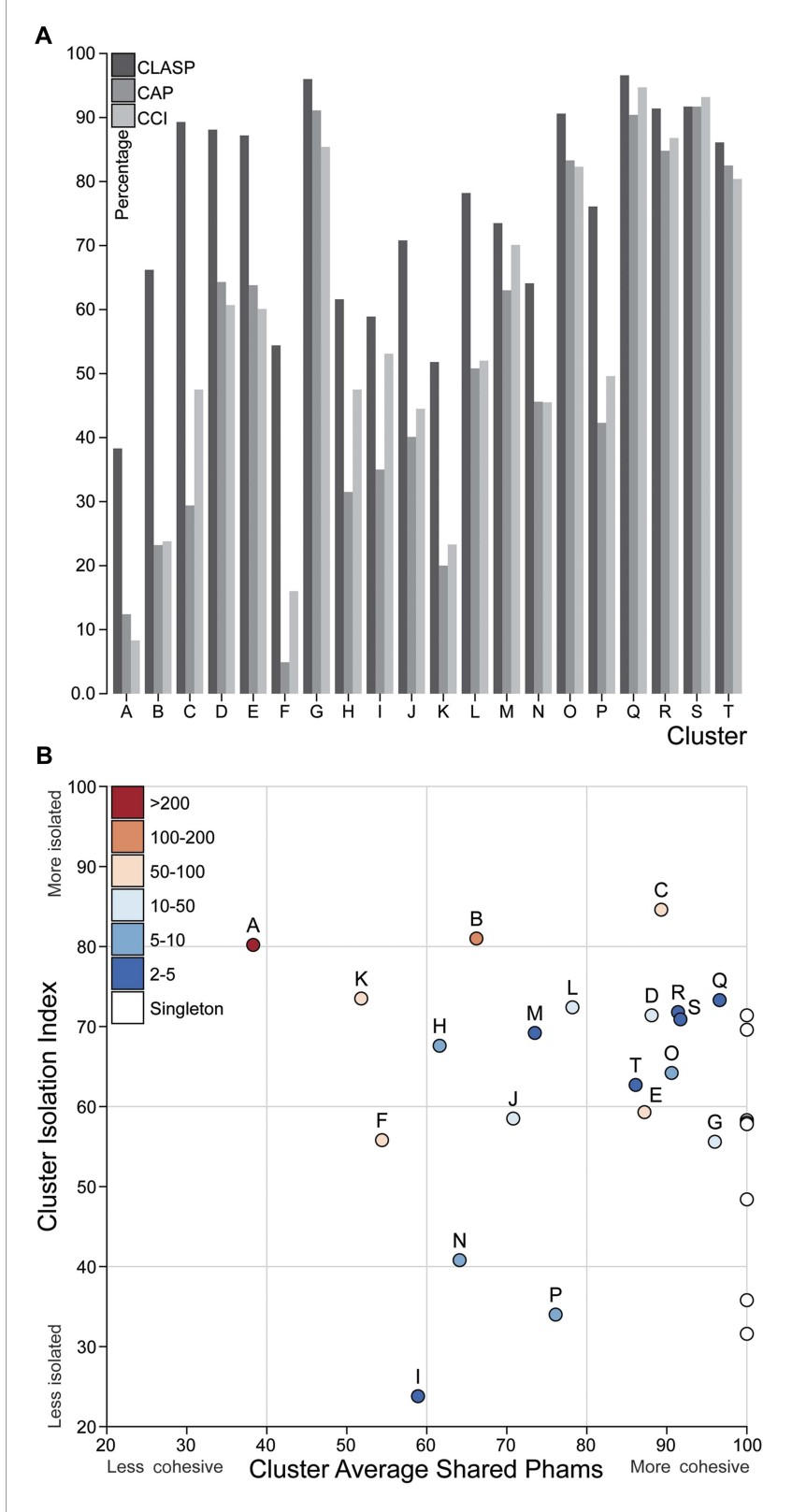

**Figure 6**. Cluster diversity and isolation. (**A**) The CLuster Averaged Shared Phamilies (CLASP; blue), Cluster Associated Phamilies (CAP; red) and Cluster Cohesion Index (CCI; green) values are plotted for each mycobacteriophage cluster. (**B**) The Cluster Isolation Index (CII) and CLASP values (both shown as percentages) are

*Figure 6. Continued*

plotted for each phage cluster. Singletons (white circles) are not individually labeled but correspond to the values shown in *Table 1*.

The following source data and figure supplements are available for figure 6:

**Source data 1**. Datasets showing numbers of CLuster Average Shard Phamilies.

**Figure supplement 1**. Resampling CLASP values for cluster diversity and size.

**Figure supplement 2**. Cluster diversity shown by Cluster-Associated Phamilies (CAP) and Cluster Phamily Variation (CPV) indices.

a surrogate for the linear range of a second hyperbolic curve, one representing the resampling of a much larger set of gene phamilies available for introduction into mycobacteriophage genomes. Unfortunately, the current dataset remains insufficient to confidently extrapolate to give an estimate of the total number of viral protein families in the biosphere, which has been previously estimated to be anywhere between a half a million and 2 billion (*Rohwer, 2003*; *Ignacio-Espinoza et al., 2013*).

We note that because of the generally slow pace of the advancement of phage genomics, we have little insight into the phage populations of other hosts. We retrieved all double-stranded DNA tailed phage genomes in GenBank that we could identify (a total of 1781), corresponding to about 120 host bacterial genera, with a median number of phages per host genus of two. Using similar parameters for pham building as described above, the 181,717 predicted genes assemble into 47,479 phamilies. The relatively low representation of each phamily (3.8 genes/phamily) compared to the mycobacteriophages (13.4 genes/phamily) is a further reflection of the gross under-sampling of the phage population as a whole.

## Implications for bacteriophage taxonomy

Bacteriophage taxonomic classification reflecting phylogeny presents substantial challenges because of genome mosaicism (*Lawrence et al., 2002*). Classification by viral morphology is well established, but may not accurately reflect the genetic relationships, as illustrated for the prolate-headed MooMoo (*Figure 3*). We also note that the mycobacteriophage myoviruses have a high CII and form a discrete group (*Table 1*) as do the *Synechococcus* myophages (*Deng et al., 2014*), perhaps reflecting a virulent lifestyle that constrains productive gene exchange; T4-related phages from diverse hosts share a core set of 15–20% of their genes, and whole genome comparisons reveal extensive mosaicism (*Petrov et al., 2010*). Host range mutability thus may differ in phages with different morphotypes, limiting access to the gene pool, and although grouping phages into clusters and subclusters provides analytical advantages because of the wide range in prevalence of different phages (*Table 1*), it is not suitable as a broadly applicable hierarchical taxonomic system. The comparative analysis of these mycobacteriophages thus supports reticulate taxonomies that more accurately reflect the phylogenetic complexities (*Lawrence et al., 2002*; *Lima-Mendez et al., 2007*).

## Implications for student learning through research experiences

A research experience can be a powerful vehicle that enables a person to gain an understanding of the process of science (*Hunter et al., 2007*). When the research experience occurs early and at a large scale, as described here, the focus can shift from selecting a few 'qualified' students to exploring the potential interests of many students. Clearly, an essential ingredient is the nature of the research project, as definitions of research may vary from an inquiry-based exercise to authentic research with the potential to contribute publishable findings. To optimize the educational benefits, the research project must be intellectually and technically accessible to beginning students (i.e., few prerequisites) and scalable so that many students can simultaneously make progress in parallel, yet independently (*Hatfull et al., 2006*). Importantly, each student's findings should contribute to a scientific question with integration of all students' discoveries advancing a scientific question of significance, as judged by scientific peer review. This, we

**Table 2**. Genometrics and Cluster Cohesion Indexes of mycobacteriophages

| Cluster | Subcluster | # Genomes | Average # genes | Average length (bp) | # Phams | CLASP* | CAP† | CCI‡ |
|---|---|---|---|---|---|---|---|---|
| A | | 232 | 90.0 | 51,514 | 1085 | 38.3 | 12.4 | 8.0 |
| | A1 | 72 | 91.2 | 51,954 | 416 | 72.3 | 36.9 | 22.0 |
| | A2 | 28 | 93.4 | 52,805 | 312 | 64.7 | 30.1 | 30.0 |
| | A3 | 37 | 87.7 | 50,325 | 163 | 81.1 | 48.8 | 54.0 |
| | A4 | 46 | 87.4 | 51,376 | 125 | 92.7 | 70.6 | 70.0 |
| | A5 | 16 | 86.0 | 50,531 | 152 | 81.4 | 58.7 | 57.0 |
| | A6 | 11 | 97.8 | 51,677 | 128 | 90.2 | 75.1 | 76.0 |
| | A7 | 3 | 84.3 | 52,941 | 115 | 74.9 | 64.4 | 73.0 |
| | A8 | 4 | 97.8 | 51,597 | 107 | 93.5 | 86.8 | 91.0 |
| | A9 | 4 | 96.0 | 52,838 | 106 | 92.7 | 83.4 | 91.0 |
| | A10 | 7 | 80.0 | 49,174 | 112 | 81.6 | 60.9 | 71.0 |
| | A11 | 4 | 98.5 | 52,260 | 113 | 93.6 | 88.3 | 87.0 |
| B | | 108 | 100.4 | 68,653 | 421 | 66.2 | 23.2 | 24.0 |
| | B1 | 77 | 101.8 | 68,532 | 144 | 93.2 | 72.9 | 71.0 |
| | B2 | 8 | 89.9 | 67,267 | 101 | 94.9 | 84.6 | 89.0 |
| | B3 | 12 | 102.8 | 68,698 | 121 | 96.3 | 84.7 | 85.0 |
| | B4 | 8 | 96.1 | 70,619 | 166 | 79.9 | 45.8 | 58.0 |
| | B5 | 3 | 96.3 | 70,033 | 108 | 91.7 | 87.2 | 89.0 |
| C | | 45 | 231.0 | 155,504 | 486 | 89.3 | 29.4 | 48.0 |
| | C1 | 44 | 231.0 | 155,297 | 345 | 91.9 | 73.2 | 67.0 |
| | C2 | 1 | 229.0 | 164,602 | 227 | N/A | N/A | N/A |
| D | | 10 | 89.3 | 64,965 | 147 | 88.1 | 64.3 | 61.0 |
| | D1 | 9 | 87.3 | 64,697 | 100 | 94.9 | 88.8 | 87.0 |
| | D2 | 1 | 107.0 | 67,383 | 107 | N/A | N/A | N/A |
| E | | 35 | 141.9 | 75,526 | 235 | 87.2 | 63.8 | 60.0 |
| F | | 66 | 105.3 | 57,416 | 658 | 54.4 | 4.9 | 16.0 |
| | F1 | 60 | 104.8 | 57,486 | 573 | 59.6 | 20.6 | 18.0 |
| | F2 | 5 | 110.8 | 55,996 | 207 | 65.7 | 49.0 | 54.0 |
| | F3 | 1 | 107.0 | 60,285 | 105 | N/A | N/A | N/A |
| G | | 14 | 61.5 | 41,845 | 72 | 96.0 | 91.1 | 85.0 |
| H | | 5 | 98.4 | 69,469 | 207 | 61.6 | 31.5 | 48.0 |
| | H1 | 4 | 95.8 | 69,137 | 131 | 81.9 | 67.9 | 73.0 |
| | H2 | 1 | 109.0 | 70,797 | 110 | N/A | N/A | N/A |
| I | | 4 | 78.0 | 49,954 | 147 | 58.9 | 35.0 | 53.0 |
| | I1 | 3 | 76.0 | 47,588 | 101 | 77.5 | 66.7 | 75.0 |
| | I2 | 1 | 84.0 | 57,050 | 84 | N/A | N/A | N/A |
| J | | 16 | 239.8 | 110,332 | 530 | 70.8 | 40.1 | 45.0 |
| K | | 33 | 95.7 | 59,720 | 411 | 51.8 | 20.0 | 23.0 |
| | K1 | 15 | 94.3 | 59,877 | 166 | 85.5 | 47.9 | 57.0 |
| | K2 | 4 | 96.3 | 56,597 | 128 | 85.2 | 77.7 | 75.0 |
| | K3 | 3 | 98.2 | 61,322 | 111 | 92.2 | 89.5 | 88.0 |
| | K4 | 5 | 94.0 | 57,865 | 106 | 93.7 | 87.2 | 89.0 |
| | K5 | 6 | 98.2 | 62,154 | 144 | 82.1 | 68.2 | 68.0 |
| L | | 13 | 127.9 | 75,177 | 246 | 78.2 | 50.8 | 52.0 |

*Table 2. Continued on next page*

*Table 2. Continued*

| Cluster | Subcluster | # Genomes | Average # genes | Average length (bp) | # Phams | CLASP* | CAP† | CCI‡ |
|---------|-----------|-----------|-----------------|---------------------|---------|--------|------|------|
|         | L1        | 3         | 123.7           | 74,050              | 135     | 92.6   | 88.8 | 92.0 |
|         | L2        | 9         | 129.3           | 75,456              | 170     | 90.1   | 72.2 | 76.0 |
|         | L3        | 1         | 128.0           | 76,050              | 126     | N/A    | N/A  | N/A  |
| M       |           | 3         | 141.0           | 81,636              | 201     | 73.5   | 63.0 | 70.0 |
|         | M1        | 2         | 135.0           | 80,593              | 138     | 96.6   | 96.6 | 98.0 |
|         | M2        | 1         | 153.0           | 83,724              | 152     | N/A    | N/A  | N/A  |
| N       |           | 7         | 69.1            | 42,888              | 152     | 64.1   | 45.6 | 45.0 |
| O       |           | 5         | 124.2           | 70,651              | 151     | 90.6   | 83.3 | 82.0 |
| P       |           | 9         | 78.8            | 47,668              | 159     | 76.1   | 42.3 | 50.0 |
|         | P1        | 8         | 78.4            | 47,313              | 126     | 82.9   | 52.9 | 62.0 |
|         | P2        | 1         | 82.0            | 50,513              | 82      | N/A    | N/A  | N/A  |
| Q       |           | 5         | 85.2            | 53,755              | 90      | 96.6   | 90.4 | 95.0 |
| R       |           | 4         | 101.5           | 71,348              | 117     | 91.4   | 84.8 | 87.0 |
| S       |           | 2         | 109.0           | 65,172              | 117     | 91.7   | 91.7 | 93.0 |
| T       |           | 3         | 66.7            | 42,833              | 83      | 86.1   | 82.5 | 80.0 |

*Cluster Averaged Shared Phamilies.
†Cluster Associated Phamilies.
‡Cluster Cohesion Index.

believe, defines an 'authentic' research experience. We note that in the SEA-PHAGES platform, substantial student effort is invested in arriving at high-quality genome annotations by close manual inspection followed by expert verification, a critical component of the detailed comparative analysis of phage gene content described here.

## Concluding comments

Bacteriophage genomics has progressed relatively slowly compared to that of other microbes in spite of their relatively small genome sizes. Here we have demonstrated that programmatically integrating the research and education missions at large scale provides an effective solution to expanding our knowledge of viral diversity, with a multitude of insights gained as a consequence of the scale of phage discovery. The nature of different genomic types, the variations of the diversity both within clusters and shared genome content among clusters, and the expanse of the mycobacteriophage population can be viewed at an unprecedented level of resolution. Our conclusions align well with comparative analyses of phages of *Enterobacteriacea* (*Grose and Casjens, 2014*) and *Bacillus* spp. (*Grose et al., 2014*) and we predict that these are general parameters of bacteriophage diversity, at least when sampling broadly across the environment. Both the rarefaction analysis described here and preliminary analysis of phamilies of all sequenced DNA phages illustrate how little of the global phage population has been genomically sampled. With a near endless supply of diverse viruses readily accessible for isolation and analyses, integrated research/education programs will continue to play substantial roles in defining the nature of the virosphere.

## Materials and methods

### Phages and genomes

In addition to extant GenBank sequence information, mycobacteriophages were isolated, sequenced, and annotated in the Phage Hunters Integrating Research and Education (PHIRE) or SEA-PHAGES programs. Phage genomes were shotgun sequenced using either 454, Ion Torrent, or Illumina platforms to at least 20-fold coverage. Shotgun reads were assembled de novo with Newbler

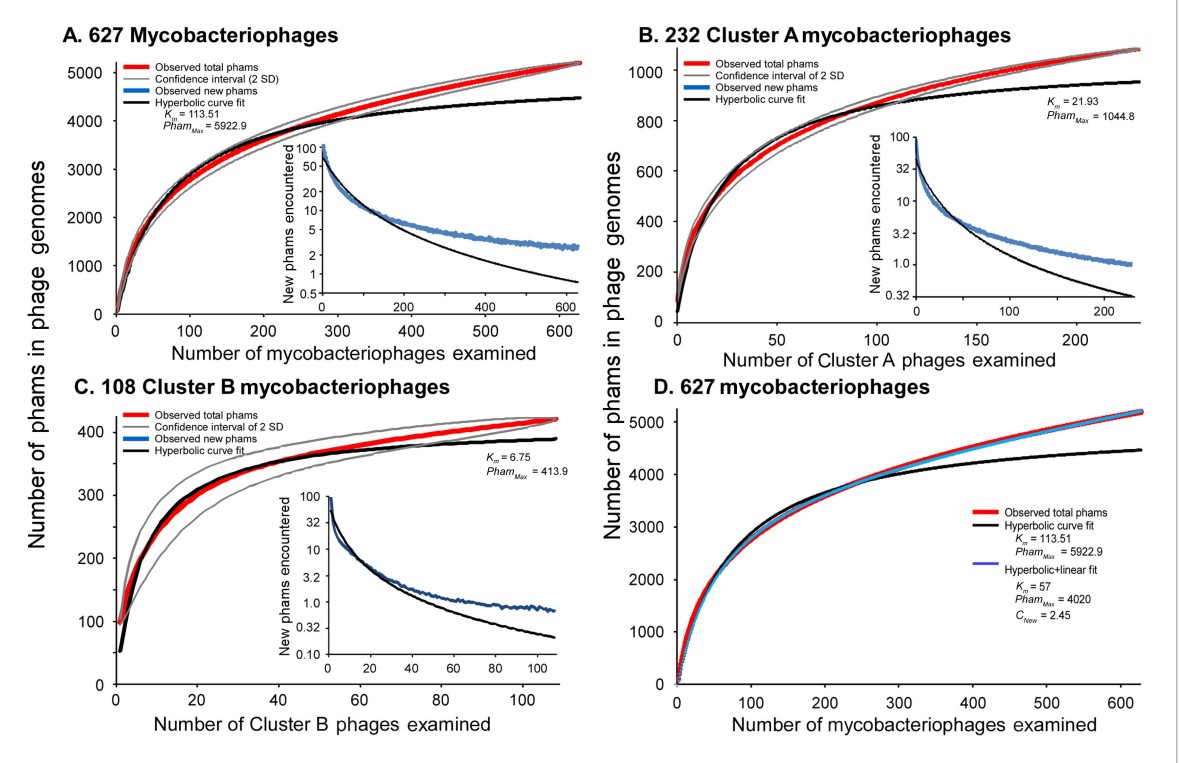

**Figure 7**. Rarefaction analysis of mycobacteriophage genomes. (**A**) The numbers of phamilies are reported for between 1 and 627 phage genomes sampled at random without replacement; the mean of 10,000 iterations is shown in red; gray lines indicate a confidence interval of two standard deviations. The black line shows a hyperbolic curve fit to the data from phage counts 1 to 314. The inset shows the number of new phams encountered upon the inclusion of each phage, with the mean number for the 10,000 iterations shown in blue and the predicted value from the hyperbolic curve shown in black. (**B**) Rarefaction analysis of 232 Cluster A phages. The total numbers of phamilies are reported for between 1 and 232 phages sampled at random without replacement from Cluster A; the mean of 10,000 iterations is shown in red; gray lines indicate a confidence interval of two standard deviations. The black line shows a hyperbolic curve fit to the data from phage counts 1 to 117. The inset shows the number of new phams encountered upon the inclusion of each phage, with the mean number for 10,000 iterations shown in blue and the predicted value from the hyperbolic curve shown in black. (**C**) Rarefaction analysis of 108 Cluster B phages; the hyperbolic curve was fit to the data from phage counts 1 to 54. (**D**) Fits of the hyperbolic (*Equation 1*) and hyperbolic with linear (*Equation 2*) models for phamily identification within genome samples.

The following source data is available for figure 7:

**Source data 1**. Datasets for determination of rarefaction curves.

versions 2.1 to 2.9. Assemblies were checked for low coverage or discrepant areas, and targeted Sanger reads were used to resolve weak areas and identify genome ends. All genome sequences are publically available at phagesDB.org or in GenBank. Nucleotide comparisons used BLASTN or Gepard (*Krumsiek et al., 2007*).

## Database construction

To create Phamerator database Mykobacteriophage_627, phamilies were constructed by first clustering the entire database of 69,574 genes using strict kClust parameters (70% clustering threshold and 0.25 alignment coverage of the longer sequence). This was followed by multiple sequence alignment of each preliminary cluster using Kalign (*Lassmann and Sonnhammer, 2005*). Consensus sequences were then extracted using HHmake and HHconsensus (*Remmert et al., 2012*). The resulting list of sequences was subjected to a second—and less strict—round of clustering via kClust (30% clustering threshold and 0.5 alignment coverage of the longer sequence) to obtain the final phamily assignments.

Network phylogeny constructions were made using the NeighborNet function with default parameters in SplitsTree (*Huson, 1998*; *Huson and Bryant, 2006*).

## Cluster diversity and isolation indices

Four parameters were used to evaluate cluster diversity. The first is the CLASP index that calculates the percentage of phamilies shared between two genomes, then averages across all possible pairs within a cluster or subcluster. Because the pairwise similarities are averaged, CLASP is relatively insensitive to either the overall size of the cluster, or the heterogeneity of its diversity (such as in Cluster C in which of the 45 genomes in total, 44 are in Cluster C1, and only one is in Cluster C2). CLASP robustness with respect to cluster size was demonstrated through a resampling analysis. For each cluster with more than 30 members, a random subset (of 5, 10, 20, or 30 genomes) was selected and CLASP was calculated. For each sample size, 20 iterations were performed with replacement. As expected, there is substantial deviation among the iterations, especially at smaller sizes. However, there is little change in the average CLASP values with different sample sizes (*Figure 4—figure supplement 1*), showing that cluster size is not a primary driver of diversity. The resampling analyses also suggest that while a greater number of genomes helps refine the CLASP value, there is still predictive power when only 10 genomes are compared. On average, the maximum and minimum iteration values at a sample size of 10 genomes were within 8% of the whole-cluster CLASP value. This implies that, for example, increasing Cluster D from 10 to 50 or 100 genomes may raise or lower its current CLASP value of 88.1, but that value is likely to remain between ~80 and ~96.

The second measure used is the CAP, which is calculated as the number of phamilies present in *all* genomes within a cluster divided by the average number of phamilies per genome. These cluster-conserved genes could correspond to core genes that define a particular phage group such as cluster or subcluster. However, for those clusters with sufficient diversity to detect such core genes, these values are low. For example, among the 66 Cluster F genomes, only five phamilies are present in all genomes. None are virion structural genes, one is a glycosyltransferase whose role is unknown, one is a putative regulator, and the others are small proteins of unknown function. For the Cluster A genomes, 11 phamilies are conserved, seven of which are virion structural proteins, three are involved in DNA metabolism (DNA Pol, Helicase, Rec-Like protein), and one is of unknown function.

The third parameter is the Cluster Phamily Variation (CPV) index, which is the proportion of phams that are not present in all members of the cluster. CAP and CPV are inversely related but imperfectly as CPV varies with cluster size even among similarly diverse clusters; a plot of CAP values against CPV values is shown in *Figure 6—figure supplement 2*.

The CCI is calculated as the average number of genes per genome as a percentage of the total number of phams in that cluster. Thus if all genomes in a cluster are identical (and if phamilies occur only once in a genome), CCI would be 100; the CCI for two sets of five randomly chosen genomes is ~2. CCI values correlate with cluster size, but similarly sized clusters as such G, J, and L, or E and K have substantially different CCI values (*Table 1*).

The CII is the percentage of phams present within a cluster that are not present in other mycobacteriophage genomes.

## Rarefaction analysis

Rarefaction analysis was performed by randomly selecting subsets (without replacement) of between 1 and 627 (all), 232 (Cluster A) or 108 (Cluster B) mycobacteriophages and determining the numbers of phamilies represented. This was repeated 10,000 times to generate a mean number of phamilies observed given a number of phage genomes selected. The means of the accumulated numbers of phams and the numbers of new phages identified are plotted as the function of the number of genomes selected at random. The observed numbers were fit to a hyperbolic function for 50% of the sample (i.e., 1 to 314, 116 or 54 genomes for all, Cluster A or Cluster B phages, respectively); Hanes-Woolf regression was used to estimate $Pham_{Max}$ and $K_m$ of the hyperbola:

$$N_{Phams} = \frac{Pham_{Max} \times N_{Genomes}}{K_m + N_{Genomes}}, \quad (1)$$

where $N_{Genomes}$ is the number of genomes sampled, $N_{Phams}$ is the number of total phams seen within those genomes, $Pham_{Max}$ is the total number of phams among all mycobacteriophage genomes, and $K_m$ is the number of genomes required to sample one half of $Pham_{Max}$.

The lack of fit of the observed data to the hyperbola—with the observed data reflecting infinite size—suggests that the overall population is dynamic. The lack of hyperbolic fit of the data does not result from outliers such as phages with highly deviant GC%, because removing these does not improve the fit. The fit is also not substantially improved by analysis of the two largest clusters, Cluster A and Cluster B (*Figure 7*), suggesting that the dynamic nature of the gene pool is not an artifact of examining independent phage clusters with separate gene pools.

To model this behavior, we modified *Equation 1* to include the introduction of novel phams via recombination with outside, non-mycobacteriophage genomes:

$$N_{Phams} = N_{Genomes} \times C_{Phage} + \frac{Pham_{Max} \times N_{Genomes}}{K_m + N_{Genomes}}, \tag{2}$$

where $C_{Phage}$ is the number of outside phams seen in each phage. The value of $C_{Phage}$ was estimated from *Figure 7B* and new values for $Pham_{Max}$ and $K_{Pham}$ were estimated by Hanes-Woolf regression following data normalization.

## Acknowledgements

Students, faculty, and their contributions to authorship are listed in the *Supplementary file 2*. We thank Aileen Beard, Gerald Henkel-Johnson, and Larry McGahey at the College of St. Scholastica, the Core Facility for Imaging, Cellular and Molecular Biology at Queens College, Jennifer Kelly and Towanda Kirksey-Stanton at Jacksonville State Univeristy, Susan Crump at Merrimack College and Dr Gregory Hendricks at the University of Massachusetts Medical School Electron Microscopy Imaging Facility, Melissa Cox at North Carolina State University and Valerie Lapham at the NCSU Center for Electron Microscopy, Dr Karen M Snetselaar at Saint Joseph's University, and Rick Ellingworth at the University of Wisconsin-River Falls for excellent technical assistance. We also thank Drs Winston Anderson and Broderick Eribo for their roles as consultants at Howard University, and Dr R Edelmann and the Miami University Center for Advanced Microscopy and Imaging for their support and assistance with electron microscopy. We also thank John Morrell, Alicia Brighton, Joshua Fisher, Michael Shelfo, Brigham Wright, Jessica Engle, Brian Early, Kyle Smith, Kyler Haskell, Tambi Issac, Bryce Lunt, David Payne II, Lissenya Argueta, Bryan Merrill, Adam Gardner, Hailey Meadows, Adam Hansen, and Marshall Sheide for contributions to phage isolation. This work was supported in part by the Howard Hughes Medical Institute SEA-PHAGES program, by the Howard Hughes Medical Institute through its Professorship grant to GFH, and by NIH grant GM51975 to GFH. Additional support was provided by the Department of Microbiology and Molecular Biology and the BYU College of Life Sciences; Cabrini College; NIH Grant No. P20 GM103408 to the College of Idaho; the National Science Foundation grant 0703449, the CUNY LSAMP program, the Office of the Provost, the Division of General Education, the Division of Mathematics and Natural Sciences, the Queens College UM/RE program and the Biology Department at Queens College; the Department of Biological Sciences at Lehigh University; the Center for Biotechnology and Biomedical Sciences of the Department of Biology in the School of Science and Engineering at Merrimack College; the NCSU Biotechnology Program and Department of Microbiology; a Davis Foundation grant, the Providence College Undergraduate Research Committee, RI-INBRE and NIGMS grant R15-GM094712 to Providence College; the Department of Biology, Saint Joseph's University; the Natural Science Department at the University of Houston-Downtown; the University of Maine Honors College, the University of Maine Department of Molecular and Biomedical Sciences, Maine-INBRE and NIH-INBRE Grant 8P20GM1003423-12; an Institutional Development Award (IDeA) from the National Institute of General Medical Sciences of the National Institutes of Health (P20GM0103423) to the University of Maine at Machias; the Howard Hughes Medical Institute, RISE and BRIC Programs, the Department of Biology, and the Offices of the Academic Dean and Chancellor at the University of Puerto Rico at Cayey; the University of Wisconsin River Falls Biology Department; the Gatton Academy of Science and Mathematics in Kentucky and the Western Kentucky University Bioinformatics and Information Science Center; Georgia College STEM Initiative and a Georgia College Faculty Research Grant; NSF Grant

REVISION DUE-1205059 and the Department of Natural Sciences at Del Mar College; Miami University Department of Microbiology and the College of Arts and Science Dean's office and National Science Foundation ABI award 1146960; the HHMI Scicomp Project (52007572) to Xavier University of Louisiana; the Doris Duke Foundation; the Gonzaga University Biology Department, NSF-TUES grant DUE-1245778 and HHMI Undergraduate Science Education grant to Gonzaga University; the School of Biological Sciences and the School of Molecular Biosciences at Washington State University; and the Benjamin Harris Memorial Fund through the Pittsburgh Foundation.

## Additional information

### Group author details

**Science Education Alliance Phage Hunters Advancing Genomics and Evolutionary Science**

Patrick Abbazia: Biology and Chemistry, Nyack College, Nyack, United States; Kristia Abernathy: Xavier University of Louisiana, New Orleans, United States; Andrew Abesamis: Biology, Loyola Marymount University, Los Angeles, United States; Syed Amaan Abidi: Biology, University of California San Diego, La Jolla, United States; Mamon Abrahim: Biological Sciences, University of Pittsburgh, Pittsburgh, United States; Colton Abrams: Environmental and Biological Science, University of Maine, Machias, Machias, United States; Alecia Achimovich: Biology, Gettysburg College, Gettysburg, United States; Brandon Ackerman: Biological Sciences and Geology, Queensboro Community College, Bayside, United States; Jonuelle Acosta: Purdue University, West Lafayette, United States; Luis A Actis: Microbiology, Miami University, Oxford, United States; Tamarah L Adair: Department of Biology, Baylor University, Waco, United States; Jaime Adame: Natural Sciences, Del Mar College, Corpus Christi, United States; Sandra D Adams: Montclair State University, Montclair, United States; Jefferson Adams: University of Maine, Honors College, Orono, United States; Kenyeda B Adams: Biology, Spelman College, Atlanta, United States; Rashidat F Adekunle: School of Science and Technology, Georgia Gwinnett College, Lawrenceville, United States; Christianah Ademuwagun: Biology, Howard College, Washington, DC, United States; Eric J Adjei-Danquah: Biology, Saint Joseph's University, Philadelphia, United States; Nancy Adkins: Biological Sciences, University of Pittsburgh, Pittsburgh, United States; Sheetal Agarwal: Montclair State University, Montclair, United States; Riddhima Agarwal: Biological Sciences, Carnegie Mellon University, Pittsburgh, United States; Geovar Agbayani: Biology, Gonzaga University, Spokane, United States; Robert Agee: Purdue University, West Lafayette, United States; Sahil Aggarwal: Virginia Commonwealth University, Richmond, United States; Temitayo Agoro: Morehouse College, Atlanta, United States; Carmen Aguirre: Biology, College of St. Scholastica, Duluth, United States; Rachael Ahler: Microbiology, Miami University, Oxford, United States; Salman Ahmad: Biology, Loyola Marymount University, Los Angeles, United States; Amiya Ahmed: Biology, University of Alabama Birmingham, Birmingham, United States; Michelle Ahn: Biology and Chemistry, Nyack College, Nyack, United States; Stephen Aiken: Biology, University of Wisconsin-River Falls, River Falls, United States; Kara Aittama: Biology, Carthage College, Kenosha, United States; Bisma Ahmed Ajaz: Biology, University of California San Diego, La Jolla, United States; Alexandra Akins: Biology, College of St. Scholastica, Duluth, United States; Bukola Akintayo: Biology, Howard College, Washington, DC, United States; Felix Akojie: Western Kentucky University, Bowling Green, United States; Zein Al-Atrache: Biological Sciences, University of Mary Washington, Fredericksburg, United States; Ola-Oluwakiti Alabi: Biology, Calvin College, Grand Rapids, United States; Olamide Alakija: Biology, University of Alabama Birmingham, Birmingham, United States; Nitheesha Alapati: Department of Biology, Baylor University, Waco, United States; Christian Alba: Biology, North Carolina Central University, Durham, United States; Patrick Albertolle: The Evergreen State College, Olympia, United States; Pedro Alejandro Ajsivinac: Purdue University, West Lafayette, United States; Cindy Alexander: Biology, Hope College, Holland, United States; Lisa M Alexander: Biological Sciences, Carnegie Mellon University, Pittsburgh, United States; Rush Alexander: Natural Sciences, University of Houston-Downtown, Houston, United States; Stephen Aley: Biology, University of Texas at El Paso, El Paso, United States; Andrea Alfonso: Natural Sciences, Del Mar College, Corpus Christi, United States; Rebecca F Alford: Biological Sciences, Carnegie Mellon University, Pittsburgh,

United States; Sarah Ali: Department of Biology, Baylor University, Waco, United States; Raul Alicea-Cabrera: Biology, University of Puerto Rico - Cayey, Cayey, United States; Malak Alkanani: Biology, Calvin College, Grand Rapids, United States; Dwa'a Alkhalaf: Montclair State University, Montclair, United States; Brandon J Allen: Department of Biology, Baylor University, Waco, United States; Elizabeth A Allen: Biology, College of William and Mary, Williamsburg, United States; Elizabeth Allen: Biology, Carthage College, Kenosha, United States; Venkata Alluri: Biology, University of Texas at El Paso, El Paso, United States; Fernanda Alonzo: Biology, University of Louisiana at Monroe, Monroe, United States; Erika Alvarado: Biology, Carthage College, Kenosha, United States; Dymaries Alvarado-Vega: Biology, University of Puerto Rico - Cayey, Cayey, United States; Amanda Alvelo-Aviles: Biology, University of Puerto Rico - Cayey, Cayey, United States; Maria Alvisi: Biology, Smith College, Northampton, United States; Kimberly Amick: Microbiology and Biotechnology, North Carolina State University, Raleigh, United States; Kimber M Amweg: Biology, College of Charleston, Charleston, United States; Kirk R Anders: Biology, Gonzaga University, Spokane, United States; Alexander G Anderson: Biology, Washington University in St. Louis, St. Louis, United States; Alison Anderson: Ohio State University, Columbus, United States; Kelly Anderson: Biology, University of Louisiana at Monroe, Monroe, United States; Michael Anderson: Ohio State University, Columbus, United States; Joseph Anderson: Virginia Commonwealth University, Richmond, United States; Kathryn M Anderson: Biology, Washington University in St. Louis, St. Louis, United States; Patrick Anderson: Biology, Gonzaga University, Spokane, United States; Sonya L Anderson: Biology, Washington University in St. Louis, St. Louis, United States; Joshua Andle: University of Maine, Honors College, Orono, United States; Nicole Anguiano: Biology, Loyola Marymount University, Los Angeles, United States; Nicolas Antis: Biology, University of Louisiana at Monroe, Monroe, United States; Abigail Antoine: Biology, Smith College, Northampton, United States; Tessa Anton: Biology, Gonzaga University, Spokane, United States; Ashley Anway: Biology, University of Wisconsin-River Falls, River Falls, United States; Callie Anyan: Biology, University of Louisiana at Monroe, Monroe, United States; Juan Apiz-Saab: Biology, University of Puerto Rico - Cayey, Cayey, United States; Javier Apodaca: Biology, University of Texas at El Paso, El Paso, United States; Robert Appleyard: The Evergreen State College, Olympia, United States; Saba Aqel: Ohio State University, Columbus, United States; Marianne Arakelyan: Department of Microbiology, Immunology, and Molecular Genetics, University of California, Los Angeles, Los Angeles, United States; Jobi Arceneaux: Biology, University of Louisiana at Monroe, Monroe, United States; Jordan Archer: Biology, College of St. Scholastica, Duluth, United States; Kathleen Archer: Biology, Trinity College, Hartford, United States; Nelish S Ardeshna: Biology, University of California San Diego, La Jolla, United States; Luke T Arduino: School of Science and Technology, Georgia Gwinnett College, Lawrenceville, United States; Manuel Ares Jr: University of California Santa Cruz, Santa Cruz, United States; Lissenya B Argueta: Microbiology and Molecular Biology, Brigham Young University, Provo, United States; Taylor Arhar: Biology, Loyola Marymount University, Los Angeles, United States; Jessica Arighi: Biology, Jacksonville State University, Jacksonville, United States; Abigail JS Armstrong: Biology, Calvin College, Grand Rapids, United States; Najealicka Armstrong: Biology, Howard College, Washington, DC, United States; Chanarion Arnold: Xavier University of Louisiana, New Orleans, United States; Eric Arnold: University of Maine, Honors College, Orono, United States; Kristin Arnold: Biology, Carthage College, Kenosha, United States; Rachel Aron: Biology, Illinois Wesleyan University, Bloomington, United States; Nikita Arora: Ohio State University, Columbus, United States; Catlin Arrington: Biology, Illinois Wesleyan University, Bloomington, United States; Lynda Asadourian: Biological Sciences, Lehigh University, Bethlehem, United States; Kathyrn Asalone: University of Maine, Honors College, Orono, United States; Anthony Ascolillo: Environmental and Biological Science, University of Maine, Machias, Machias, United States; Jessica Ashcraft: Biology, Ouachita Baptist University, Arkadelphia, United States; Brandon D Ashley: Honors Program, Florida Gulf Coast University, Fort Myers, United States; Carmela Asinas: Biology, Loyola Marymount University, Los Angeles, United States; Christopher C Asuzu: Biology, College of Charleston, Charleston, United States; Aliza Auces: Biology, College of Idaho, Caldwell, United States; Robin Audette: Biology, College of St. Scholastica, Duluth, United States; Matthew Aultman: Washington State University, Pullman, United States; Quenten Austin: Biology, Carthage College, Kenosha, United States; Nicanor Austriaco: Providence College, Providence, United States; Michelle Averkiou: University of Florida, Gainsville, United States; Taara Avery: Biology, Spelman College, Atlanta, United States; Izma Aviles: Biological Sciences, University of North Texas, Denton, United States; Anamaris Aviles-Rivera:

Biology, University of Puerto Rico - Cayey, Cayey, United States; Lauren Awdziejczyk: Biology, Illinois Wesleyan University, Bloomington, United States; Froogh Aziz: Montclair State University, Montclair, United States; Rahat Aziz: Biological Sciences, University of North Texas, Denton, United States; Grace Babbs: Western Kentucky University, Bowling Green, United States; Nikhil Babu: Washington State University, Pullman, United States; Stevie Bach: Biology, University of Louisiana at Monroe, Monroe, United States; Megan Bachman: Biology, College of St. Scholastica, Duluth, United States; Tasha D Baer: Honors Program, Florida Gulf Coast University, Fort Myers, United States; Joanna Bagienska: Biology, Smith College, Northampton, United States; Dadde Bah: School of Science and Technology, Georgia Gwinnett College, Lawrenceville, United States; Hana Baig: Biology, Howard College, Washington, DC, United States; Andrew Bailey: Department of Microbiology, Immunology, and Molecular Genetics, University of California, Los Angeles, Los Angeles, United States; Ryan Bailey: Biology, College of Charleston, Charleston, United States; Paul Baker: Ohio State University, Columbus, United States; Mitchell F Balish: Microbiology, Miami University, Oxford, United States; Sarah Ball: Center for Life Science Education, Ohio State University, Columbus, United States; Ellee Banaszak: Biology, Hope College, Holland, United States; Sophie Bandurski: Biology, Smith College, Northampton, United States; Debarko Banerji: Biology, University of Texas at El Paso, El Paso, United States; Laura Banken: Biology, College of St. Scholastica, Duluth, United States; Brittany Banks: Xavier University of Louisiana, New Orleans, United States; William J Banning: Biology, Hampden-Sydney College, Farmville, United States; Chen Bao: Biology, Washington University in St. Louis, St. Louis, United States; W Bradley Barbazuk: University of Florida, Gainsville, United States; Nastassia R Barber: Biological Sciences, Carnegie Mellon University, Pittsburgh, United States; Joshua R Barber: Division of Natural and Health Sciences, Seton Hill University, Greensburg, United States; Jessica Barber: Biology, College of Charleston, Charleston, United States; Rafi Bari: Ohio State University, Columbus, United States; Lucia Barker: Howard Hughes Medical Institute, Chevy Chase, United States; Alexis Barna: Biology, University of Wisconsin-River Falls, River Falls, United States; Emily Barner: Biology and Chemistry, Nyack College, Nyack, United States; Ryan Barnes: Biology, North Carolina Central University, Durham, United States; Brooke H Barnhart: Division of Natural and Health Sciences, Seton Hill University, Greensburg, United States; Stephanie N Barr: School of Science and Technology, Georgia Gwinnett College, Lawrenceville, United States; Alessandra L Barrera: School of Science and Technology, Georgia Gwinnett College, Lawrenceville, United States; Anne Barron: Biology, Calvin College, Grand Rapids, United States; William D Barshop: Biology, Washington University in St. Louis, St. Louis, United States; Nicole Bartels: Biology, Loyola Marymount University, Los Angeles, United States; Kristi Bartholomay: University of Colorado at Boulder, Boulder, United States; Azhar Bashir: Virginia Commonwealth University, Richmond, United States; Kimberly Bastille: Environmental and Biological Science, University of Maine, Machias, Machias, United States; Steven Bateh: University of Florida, Gainsville, United States; Tyler Bates: Biology, College of St. Scholastica, Duluth, United States; Neha Batra: Biology, University of Alabama Birmingham, Birmingham, United States; Megan Batty: Biology, Gonzaga University, Spokane, United States; Rebecca Baudin: Biology, University of Louisiana at Monroe, Monroe, United States; Victor Bauer: Biology, Gonzaga University, Spokane, United States; Cynthia Bauerle: Biology, Spelman College, Atlanta, United States; Ian M Bayles: Biological Sciences, Carnegie Mellon University, Pittsburgh, United States; Aaron Beach: Natural Sciences, Del Mar College, Corpus Christi, United States; Gwendolyn Beacham: University of Maine, Honors College, Orono, United States; Lorenzo Bean: University of Florida, Gainsville, United States; Shannon Beaty: Department of Microbiology, Immunology, and Molecular Genetics, University of California, Los Angeles, Los Angeles, United States; Torri Beaudoin: Biology, University of Louisiana at Monroe, Monroe, United States; Darius D Becker-Krail: Biology, College of Charleston, Charleston, United States; Madison Beckman: Biology, Jacksonville State University, Jacksonville, United States; Christina Beckwith: Washington State University, Pullman, United States; Blake Beehler: Biology, Illinois Wesleyan University, Bloomington, United States; Bethany Beekly: Biology, Gonzaga University, Spokane, United States; Alissa Behl: Biology, Carthage College, Kenosha, United States; Katherine Belfield: Biological Sciences, University of Mary Washington, Fredericksburg, United States; Campbell Belisle Haley: University of Maine, Honors College, Orono, United States; Abbey Bell: Biology, Calvin College, Grand Rapids, United States; Bianca Bell: University of California Santa Cruz, Santa Cruz, United States; Devyn Bell: Biology, Gonzaga University, Spokane, United States; Mecca Bell: Xavier University of Louisiana, New Orleans, United States; Trevene Bell: Biological Sciences, Lehigh

University, Bethlehem, United States; Adriano Bellotti: Microbiology and Biotechnology, North Carolina State University, Raleigh, United States; Ryan Benczik: Purdue University, West Lafayette, United States; Robert C Benjamin: Biological Sciences, University of North Texas, Denton, United States; Pilgrim Benjamin: Natural Sciences, University of Houston-Downtown, Houston, United States; Elizabeth Benner: Purdue University, West Lafayette, United States; Rebecca Benoit: Biology, Gonzaga University, Spokane, United States; Nicholas Bense: The Evergreen State College, Olympia, United States; Brandon Bensel: Biological Sciences, Lehigh University, Bethlehem, United States; Gabrielle Benson: Biology, Gettysburg College, Gettysburg, United States; Hannah Bergh: University of Colorado at Boulder, Boulder, United States; Rebecca E Berk: Biological Sciences, Carnegie Mellon University, Pittsburgh, United States; Charlotte Berkes: Biology, Merrimack College, North Andover, United States; Janella Bermudez: Biology, Gonzaga University, Spokane, United States; Joshua Bernal: Biological Sciences, University of North Texas, Denton, United States; Daniel Bernal: University of Florida, Gainsville, United States; Thomas J Bernardo: Biology, Saint Joseph's University, Philadelphia, United States; Anthony Bernicchi: Biology, Gonzaga University, Spokane, United States; Molly Berning: Providence College, Providence, United States; Jose Efrain Berrios-Lopez: Biology, University of Puerto Rico - Cayey, Cayey, United States; Luis Berrios-Pagan: Biology, University of Puerto Rico - Cayey, Cayey, United States; Johanna Berrios-Ruiz: Biology, University of Puerto Rico - Cayey, Cayey, United States; Hannah Berry: Biology, University of Louisiana at Monroe, Monroe, United States; Hannah Berry: Microbiology and Biotechnology, North Carolina State University, Raleigh, United States; Kara Beseler: Washington State University, Pullman, United States; Aaron A Best: Biology, Hope College, Holland, United States; Reba R Best: Biology, Culver-Stockton College, Canton, United States; Nicolette Bestul: Biology, University of Wisconsin-River Falls, River Falls, United States; Victoria Betancourt: University of California Santa Cruz, Santa Cruz, United States; Andres Betancourt-Torres: Biology, University of Puerto Rico - Cayey, Cayey, United States; Rudolf Beutner: Biology, University of Louisiana at Monroe, Monroe, United States; Yachana Bhakta: Biology, Gonzaga University, Spokane, United States; Nazia Bhatti: Biological Sciences and Geology, Queensboro Community College, Bayside, United States; Sonam B Bhimbra: Biology, College of Charleston, Charleston, United States; Swapan Bhuiyan: Biology, University of North Texas and University of Louisiana at Monroe, Monroe, United States; Tiffany Bibeau: The Evergreen State College, Olympia, United States; Mary Anthonette Binongcal: Biology, Gonzaga University, Spokane, United States; Monica Binsol: Montclair State University, Montclair, United States; Miles Black: Biology, Washington University in St. Louis, St. Louis, United States; William Blaine: Biology, Trinity College, Hartford, United States; Cole Blair: Western Kentucky University, Bowling Green, United States; Peter Blair: Biology, University of Alabama Birmingham, Birmingham, United States; Aaron Blake: Morehouse College, Atlanta, United States; Samantha Blake: Biology, Gonzaga University, Spokane, United States; Bradley Blankenship: Western Kentucky University, Bowling Green, United States; Michelle Blemker: Biology, Loyola Marymount University, Los Angeles, United States; Kathleen Blevins: Biological Sciences, University of Mary Washington, Fredericksburg, United States; Lawrence Blumer: Morehouse College, Atlanta, United States; Katherine Boas: Biology, Gettysburg College, Gettysburg, United States; Jon Lucas Boatwright: University of Florida, Gainsville, United States; Brittany H Bodnar: Biology, Saint Joseph's University, Philadelphia, United States; Molly Bogolin: Biology, Hope College, Holland, United States; Alan Bohn: Microbiology and Biotechnology, North Carolina State University, Raleigh, United States; Anthony Bohner: Biology, Illinois Wesleyan University, Bloomington, United States; Amy Bohner: Biology, Calvin College, Grand Rapids, United States; Cayla Boisseranc: Biology, Gonzaga University, Spokane, United States; Dave Bollivar: Biology, Illinois Wesleyan University, Bloomington, United States; Logan Bond: Biology, Ouachita Baptist University, Arkadelphia, United States; J Alfred Bonilla: Biology, University of Wisconsin-River Falls, River Falls, United States; James Bonner: Washington State University, Pullman, United States; Daniel Bonnette: Biology, University of Louisiana at Monroe, Monroe, United States; Ashley Boone: Biology, Gettysburg College, Gettysburg, United States; Kyle Boone: Biology, University of Texas at El Paso, El Paso, United States; Amanda Boozalis: Biology, Washington University in St. Louis, St. Louis, United States; Jaclyn Ann Boozalis: Biology, Washington University in St. Louis, St. Louis, United States; Elyse Borchik: Biology, Illinois Wesleyan University, Bloomington, United States; Brooke Borgert: University of Florida, Gainsville, United States; Kim M Borges: Arts and Sciences Division, University of Maine, Fort Kent, Fort Kent, United States; Denisse Borja: Biology, University of Texas at El Paso, El Paso, United States; Julia Boroday: Biological Sciences and Geology,

Queensboro Community College, Bayside, United States; Dajana Borova: Montclair State University, Montclair, United States; Mary Borque: Biology, University of Louisiana at Monroe, Monroe, United States; Valerie Bostrom: Biology, Washington University in St. Louis, St. Louis, United States; Mara Bottomley: Biology, Hope College, Holland, United States; James Bowen: Biological Sciences, Lehigh University, Bethlehem, United States; Ian N Boys: Department of Biology, Baylor University, Waco, United States; Kevin Bradley: Howard Hughes Medical Institute, Chevy Chase, United States; Kosi Bradley: Morehouse College, Atlanta, United States; Jace Bradshaw: Biology, Ouachita Baptist University, Arkadelphia, United States; Judd Bragg: Environmental and Biological Science, University of Maine, Machias, Machias, United States; Kaitlyn Brahm: Biology, Carthage College, Kenosha, United States; Veronica E Brandley: Biology, Saint Joseph's University, Philadelphia, United States; Andrew Brannan: Biology, Jacksonville State University, Jacksonville, United States; Clinton Branton: Biology, University of Louisiana at Monroe, Monroe, United States; Clayton Branton: Biology, University of Louisiana at Monroe, Monroe, United States; Caitlyn B Brashears: Department of Biology, Baylor University, Waco, United States; Sara Bratsch: Biology, University of Wisconsin-River Falls, River Falls, United States; Edward L Braun: University of Florida, Gainsville, United States; Mary A Braun: Biological Sciences, Carnegie Mellon University, Pittsburgh, United States; Gabriel Brautman: Microbiology and Biotechnology, North Carolina State University, Raleigh, United States; Donald P Breakwell: Microbiology and Molecular Biology, Brigham Young University, Provo, United States; Mackenzie Bredereck: Biology, Gonzaga University, Spokane, United States; Lisa Brehove: Biology, Loyola Marymount University, Los Angeles, United States; Caroline Breitenberger: Chemistry and Biochemistry, Ohio State University, Columbus, United States; Jason Breithaupt: Washington State University, Pullman, United States; Joseph Bretzmann: Purdue University, West Lafayette, United States; Levi Brewer: Biology, Jacksonville State University, Jacksonville, United States; Jerald S Bricker: Biology, Nebraska Wesleyan University, Lincoln, Nebraska, United States; Valerie C Briell: Department of Biology, Baylor University, Waco, United States; Alicia K Brighton: Microbiology and Molecular Biology, Brigham Young University, Provo, United States; Kirsten Brink: Biology, Calvin College, Grand Rapids, United States; Lauren Broadway: Biology, University of Louisiana at Monroe, Monroe, United States; John W Brooker: Biology, College of Charleston, Charleston, United States; Mia Broughton: Biology, Howard College, Washington, DC, United States; Abigail Brown: Biology, Illinois Wesleyan University, Bloomington, United States; Ariel Brown: Biology, North Carolina Central University, Durham, United States; Bryony Brown: Biological Sciences, University of Pittsburgh, Pittsburgh, United States; Emma Brown: Ohio State University, Columbus, United States; Gerald Brown: Morehouse College, Atlanta, United States; Heather Brown: Microbiology and Biotechnology, North Carolina State University, Raleigh, United States; Janaye Brown: School of Science and Technology, Georgia Gwinnett College, Lawrenceville, United States; Melissa Brown: Providence College, Providence, United States; Hilary A Brownstead: Biology, Washington University in St. Louis, St. Louis, United States; Claire Brownstone: Biology, Washington University in St. Louis, St. Louis, United States; Regina Bruce: Biology, Howard College, Washington, DC, United States; Amy Bruckbauer: Biology, Carthage College, Kenosha, United States; Laura Brumbaugh: Biology, Gettysburg College, Gettysburg, United States; Sarah Brusko: Virginia Commonwealth University, Richmond, United States; Anthony Brusnahan: Ohio State University, Columbus, United States; Christian Brutofsky: Montclair State University, Montclair, United States; Wesley Bryan: Biological Sciences, University of North Texas, Denton, United States; Hanna Bryant: University of California Santa Cruz, Santa Cruz, United States; Sarah Bryant: University of California Santa Cruz, Santa Cruz, United States; Ryann M Brzoska: Microbiology, Miami University, Oxford, United States; Harman Bual: Biology, Gonzaga University, Spokane, United States; Blake Buchanan: Biology, College of Charleston, Charleston, United States; Ciara Buechner: Biology, University of Wisconsin-River Falls, River Falls, United States; Daniel J Buhalo: Biology, Saint Joseph's University, Philadelphia, United States; Taylor Buhr: Biology, Nebraska Wesleyan University, Lincoln, Nebraska, United States; Duy Xuan Bui: Biology, University of California San Diego, La Jolla, United States; Mary Bulfin: Microbiology and Biotechnology, North Carolina State University, Raleigh, United States; Sarah Bunker: Biology, Trinity College, Hartford, United States; Mary Burak: Providence College, Providence, United States; Sarah Burdette: The Evergreen State College, Olympia, United States; Elizabeth Burger: Montclair State University, Montclair, United States; Aaron Burghgraef: Biology, Calvin College, Grand Rapids, United States; Kyle Burghgraef: Biology, Calvin College, Grand Rapids, United States; Kevin Burke: Biological Sciences, University of Pittsburgh, Pittsburgh, United

States; Victoria Burkhead: The Evergreen State College, Olympia, United States; Tate Burkholder: Biotechnology, James Madison University, Harrisonburg, United States; Andrew Burlingame: The Evergreen State College, Olympia, United States; Sandra H Burnett: Microbiology and Molecular Biology, Brigham Young University, Provo, United States; Angela Burr: Biological Sciences, University of North Texas, Denton, United States; Derek Burton: Biology, North Carolina Central University, Durham, United States; Tiffany Burton: Biology, Carthage College, Kenosha, United States; Mathew Bushey: Biology, College of St. Scholastica, Duluth, United States; Kristina Busser: Ohio State University, Columbus, United States; Nicholas Bussian: Biology, Carthage College, Kenosha, United States; Maude Bute: Biology, Howard College, Washington, DC, United States; Kristen A Butela: Division of Natural and Health Sciences, Seton Hill University, Greensburg, United States; Mandy Butler: Biology, University of California San Diego, La Jolla, United States; Dominique Bynum-Cooper: Biology, Howard College, Washington, DC, United States; Meghan Byrne: Biological Sciences, University of Pittsburgh, Pittsburgh, United States; Deanna Byrnes: Biology, Carthage College, Kenosha, United States; Cristina Cabrera-Mino: Biological Sciences, Carnegie Mellon University, Pittsburgh, United States; Carla Caceres-Velazquez: Biology, University of Puerto Rico - Cayey, Cayey, United States; Jacqueline Caelwarts: Biology, College of St. Scholastica, Duluth, United States; Grace Cain: Biology, University of Alabama Birmingham, Birmingham, United States; Mario Caldararo: Montclair State University, Montclair, United States; Darcie J Caldwell: Biology, Montana Tech of the University of Montana, Butte, United States; Tomika S Caldwell: Biology, College of Charleston, Charleston, United States; Czarina Calicdan: Biological Sciences and Geology, Queensboro Community College, Bayside, United States; Christopher M Caliva: Biology, Illinois Wesleyan University, Bloomington, United States; Caitlin J Callaghan: Biology, Saint Joseph's University, Philadelphia, United States; Brennan Calley: Biology, Gonzaga University, Spokane, United States; William Eamon Callison: Biology, Washington University in St. Louis, St. Louis, United States; Lee Calvert: Western Kentucky University, Bowling Green, United States; Javier Camacho: Biology, University of Texas at El Paso, El Paso, United States; Amanda Campbell: Biotechnology, Southern Maine Community College, South Portland, United States; Warren Campbell: Biology, Gettysburg College, Gettysburg, United States; Ian W Campbell: Biological Sciences, Carnegie Mellon University, Pittsburgh, United States; Joshua L Campbell: School of Science and Technology, Georgia Gwinnett College, Lawrenceville, United States; Laura Campbell: Biology, Howard College, Washington, DC, United States; Ross Campbell: Virginia Commonwealth University, Richmond, United States; Julie Anne Canter: Biology, Illinois Wesleyan University, Bloomington, United States; Nico Carbone: Montclair State University, Montclair, United States; Erick Cardona-Portalatin: Biology, University of Puerto Rico - Cayey, Cayey, United States; Andrew Cardwell: Western Kentucky University, Bowling Green, United States; Lydia Carlin: Biology and Chemistry, Nyack College, Nyack, United States; Emily Carlisi: Biology, Gettysburg College, Gettysburg, United States; Kristen Carlisle: Biology, Jacksonville State University, Jacksonville, United States; Alexander Carlson: Biology, Gonzaga University, Spokane, United States; Courtney Carlstrom: Purdue University, West Lafayette, United States; Elizabeth Carpenter: Purdue University, West Lafayette, United States; James Carpino: Biology, CUNY, Queens College, Queens, United States; Katherine Carr: Biology, Gonzaga University, Spokane, United States; Sophia Carroll: Biology, Smith College, Northampton, United States; Connor Carry: University of California Santa Cruz, Santa Cruz, United States; Susan Carson: Microbiology and Biotechnology, North Carolina State University, Raleigh, United States; Jennifer Carter: Biology, Illinois Wesleyan University, Bloomington, United States; Leah Carter: Microbiology, Miami University, Oxford, United States; Lucas Carter: The Evergreen State College, Olympia, United States; Morgan Carter: Microbiology and Biotechnology, North Carolina State University, Raleigh, United States; Steven M Caruso: Department of Biological Sciences, University of Maryland, Baltimore County, Baltimore, United States; Sarah Carzo: Science, Cabrini College, Radnor, United States; Danielle Cascione: Providence College, Providence, United States; Tarrin Casey: Xavier University of Louisiana, New Orleans, United States; Marliz Casiano-Real: Biology, University of Puerto Rico - Cayey, Cayey, United States; Tyler Caskin: Washington State University, Pullman, United States; Colleen Cassidy: Providence College, Providence, United States; James Cassoday: Biology, University of Texas at El Paso, El Paso, United States; Paulina Castillo: Biology, University of Texas at El Paso, El Paso, United States; Byron Castillo Silva: Biology, North Carolina Central University, Durham, United States; Anna Casto-Markosky: Biology, Calvin College, Grand Rapids, United States; Erinleigh Caughron: Biology, Gonzaga University, Spokane, United

States; Heather Caulkins: Biology, Washington University in St. Louis, St. Louis, United States; Mitchel D Cavallarin: Biology, Hampden-Sydney College, Farmville, United States; William Cavedon: Providence College, Providence, United States; Elymae Cedeno Garcia: Purdue University, West Lafayette, United States; Francisco Cerda: Biology, Loyola Marymount University, Los Angeles, United States; Nicco Cerda: Morehouse College, Atlanta, United States; Logan Cerkovnik: University of Colorado at Boulder, Boulder, United States; Jasiel Cervantes: Biology, University of Texas at El Paso, El Paso, United States; Juan Cervantes: Biology, College of Idaho, Caldwell, United States; James Cescon: Biology, Trinity College, Hartford, United States; Priya Chakrabarti: Biology, University of California San Diego, La Jolla, United States; Sanjeev Chalissery: Virginia Commonwealth University, Richmond, United States; Molly Chamberlin: Purdue University, West Lafayette, United States; Codee Champney: Biology, Illinois Wesleyan University, Bloomington, United States; Dong Woo Chang: Biology, Loyola Marymount University, Los Angeles, United States; Jee Yoon Chang: Biology, Washington University in St. Louis, St. Louis, United States; Michelle Chang: Biology, University of Alabama Birmingham, Birmingham, United States; Ana Chaparro: Biology, University of Texas at El Paso, El Paso, United States; Joshua Chappell: Microbiology and Biotechnology, North Carolina State University, Raleigh, United States; Reed Charlop: The Evergreen State College, Olympia, United States; Anna Chase: Biology, College of Idaho, Caldwell, United States; Wrik Chatterjee: Microbiology, Miami University, Oxford, United States; Kevin Chavez: Biological Sciences and Geology, Queensboro Community College, Bayside, United States; Vivian Chavez: Biology, University of Texas at El Paso, El Paso, United States; Alexandra Chen: Biology, University of California San Diego, La Jolla, United States; Annie Chen: Department of Microbiology, Immunology, and Molecular Genetics, University of California, Los Angeles, Los Angeles, United States; Xiaozhu Chen: Biology, Washington University in St. Louis, St. Louis, United States; Yue Ting Chen: Biological Sciences and Geology, Queensboro Community College, Bayside, United States; Xi Chen: Biological Sciences and Geology, Queensboro Community College, Bayside, United States; Jason Chia-Sheng Cheng: Biology, University of California San Diego, La Jolla, United States; Alison A Chesky: Division of Natural and Health Sciences, Seton Hill University, Greensburg, United States; Lashanda Cheston: Biology, North Carolina Central University, Durham, United States; Mahathee Meenakshi Chetlapalli: Biology, University of California San Diego, La Jolla, United States; Darren Chew: Biology and Chemistry, Nyack College, Nyack, United States; Behroz Khushrav Chhor: Biology, University of California San Diego, La Jolla, United States; Caitlyn M Chilinski: Biology, Montana Tech of the University of Montana, Butte, United States; Pawan Chitta: Virginia Commonwealth University, Richmond, United States; Ariel W Cho: Department of Biology, Baylor University, Waco, United States; Leela D Chockalingam: Biological Sciences, Carnegie Mellon University, Pittsburgh, United States; Jeremy Chou: Biology, University of California San Diego, La Jolla, United States; Tiffany Chow: Department of Microbiology, Immunology, and Molecular Genetics, University of California, Los Angeles, Los Angeles, United States; Jordan Church: Biological Sciences, University of North Texas, Denton, United States; Richard Churchill: Biology, Trinity College, Hartford, United States; Bryce Churilla: Biological Sciences, University of Pittsburgh, Pittsburgh, United States; Eric Ciardiello: Southern Connecticut State University, New Haven, United States; Bryan Ciccarello: Biology, Washington University in St. Louis, St. Louis, United States; Varun Cidambi: Biology, University of California San Diego, La Jolla, United States; Amy Cimo: Science, Cabrini College, Radnor, United States; Cassie Clark: Biology, University of Louisiana at Monroe, Monroe, United States; Matthew Clark: Biology, Gonzaga University, Spokane, United States; Zachary Clark: Biology, University of Texas at El Paso, El Paso, United States; Kari L Clase: Purdue University, West Lafayette, United States; Barbara Clement: Doane College, Crete, United States; Dylan Clevenger: University of California Santa Cruz, Santa Cruz, United States; Tiffany Clinton: Biology, Howard College, Washington, DC, United States; Benjamin J Cody: Department of Biology, Baylor University, Waco, United States; Rainna Coelho: Biological Sciences, University of North Texas, Denton, United States; Sarah Coffee: Biology, Smith College, Northampton, United States; Gayle Coggins: Ohio State University, Columbus, United States; Kali Coghlan: Biology, Gonzaga University, Spokane, United States; Karen Cohen: Biological Sciences, University of Pittsburgh, Pittsburgh, United States; Lianne B Cohen: Biological Sciences, Carnegie Mellon University, Pittsburgh, United States; Joanna Katherine Claire Coker: Biology, University of California San Diego, La Jolla, United States; Tiffany Colburn: Biology, University of Alabama Birmingham, Birmingham, United States; Allison Cole: Biology, Gettysburg College, Gettysburg, United States; Kris Cole: Biology, University of Wisconsin-

River Falls, River Falls, United States; Arlixer Coleman: Biology, Spelman College, Atlanta, United States; Maggie Colicchio: Biological Sciences, University of Pittsburgh, Pittsburgh, United States; Eric Collin: The Evergreen State College, Olympia, United States; Carol M Collins: Biology, Saint Joseph's University, Philadelphia, United States; Joseph M Collins: Biology, Saint Joseph's University, Philadelphia, United States; Justin Collins: Biology, Hope College, Holland, United States; Theresa Collins: Biological Sciences, Lehigh University, Bethlehem, United States; Kimberly Colombini: Biology, Gonzaga University, Spokane, United States; Jennifer Colquhoun: Biological Sciences, Lehigh University, Bethlehem, United States; Jessica E Colunga: Department of Biology, Baylor University, Waco, United States; Kevin Colvin: Ohio State University, Columbus, United States; Ronald Comeaux: Morehouse College, Atlanta, United States; Brian Conahan: Biology, Gonzaga University, Spokane, United States; Destinee Cone: Washington State University, Pullman, United States; Erik Cone: Biology, Gonzaga University, Spokane, United States; Robert Connolly: The Evergreen State College, Olympia, United States; Ashley Connors: Biology, Gonzaga University, Spokane, United States; Sandra Connors: Biotechnology, Southern Maine Community College, South Portland, United States; Paul Consiglio: Ohio State University, Columbus, United States; Kathrina Consing: Biology, Loyola Marymount University, Los Angeles, United States; Daniel Conti: Biology, Loyola Marymount University, Los Angeles, United States; Troy Coody: University of Colorado at Boulder, Boulder, United States; Stephanie Cook: Biology, Jacksonville State University, Jacksonville, United States; Jamika Cookson: University of Maine, Honors College, Orono, United States; Charles Coomer: Western Kentucky University, Bowling Green, United States; Crystal Cooper: Marine Science, Southern Maine Community College, South Portland, United States; Jacob Cooper: University of Colorado at Boulder, Boulder, United States; Cianna E Corbacio: Biology, Saint Joseph's University, Philadelphia, United States; Gabriel Cordero-Bernard: Biology, University of Puerto Rico - Cayey, Cayey, United States; Sarah Corley: Providence College, Providence, United States; Kathleen Cornely: Providence College, Providence, United States; Eduardo Correa: Biology, University of Puerto Rico - Cayey, Cayey, United States; Earl Cosby: Morehouse College, Atlanta, United States; Michael Costello: Biology, University of Texas at El Paso, El Paso, United States; Catherine Cota: Southern Connecticut State University, New Haven, United States; Idalid Cotto-Berríos: Biology, University of Puerto Rico - Cayey, Cayey, United States; Alexis Cotto-Rosario: Biology, University of Puerto Rico - Cayey, Cayey, United States; Mariele Courtois: Biology, Loyola Marymount University, Los Angeles, United States; Ashley Cox: Western Kentucky University, Bowling Green, United States; Michelle Cox: Purdue University, West Lafayette, United States; Madeline Cox: Purdue University, West Lafayette, United States; Elizabeth Craig: Biology, Trinity College, Hartford, United States; Lara Crawford: Biology, University of Louisiana at Monroe, Monroe, United States; Michael Crawford: Biological Sciences, University of Mary Washington, Fredericksburg, United States; Cheryl Creed: Biology, Smith College, Northampton, United States; Victor A Crespo-Vega: Biology, University of Puerto Rico - Cayey, Cayey, United States; Charlotte Cronenweth: Biology, Loyola Marymount University, Los Angeles, United States; Trevor Cross: Science, Cabrini College, Radnor, United States; Cody Crossley: The Evergreen State College, Olympia, United States; Luis Cruz-Garcia: Biology, University of Puerto Rico - Cayey, Cayey, United States; Daniel Cui Zhou: Biology, Washington University in St. Louis, St. Louis, United States; Ryan Cullen: Biology, College of St. Scholastica, Duluth, United States; Nicole Cullen: Providence College, Providence, United States; Andrew Cullett: Biology, Illinois Wesleyan University, Bloomington, United States; Samantha Culpepper: Biological Sciences, University of North Texas, Denton, United States; John Culver: Biology, Gonzaga University, Spokane, United States; Nico Cunanan: Biology, Gonzaga University, Spokane, United States; Christina Cunha: Biology, Loyola Marymount University, Los Angeles, United States; Richard Cunningham: Morehouse College, Atlanta, United States; Taylor Cunningham: Biology, College of St. Scholastica, Duluth, United States; Michael Cuoco: Biology, Trinity College, Hartford, United States; Chiara P Curcillo: Biology, Saint Joseph's University, Philadelphia, United States; Esmeralda Curiel: Biology, University of Texas at El Paso, El Paso, United States; James Curlin: Biology, Trinity College, Hartford, United States; Renee Curry: Biology and Chemistry, Nyack College, Nyack, United States; Matt Cusmiani: Science, Cabrini College, Radnor, United States; Christie L Cutting: Biological Sciences, Carnegie Mellon University, Pittsburgh, United States; Lauren Czaja: Biology, Illinois Wesleyan University, Bloomington, United States; Karolina W Czarnecki: Honors Program, Florida Gulf Coast University, Fort Myers, United States; Emila Czyszczon: Purdue University, West Lafayette, United States; Thomas D'Addario: Biology, Hope

College, Holland, United States; Satish Dahal: Biology, University of Louisiana at Monroe, Monroe, United States; Alysan Dahl: University of Colorado at Boulder, Boulder, United States; Katherine Daily: Biology, Loyola Marymount University, Los Angeles, United States; Tiffany Damiani: Biology, Trinity College, Hartford, United States; Kyle Dammann: Biology, University of Louisiana at Monroe, Monroe, United States; Jolene SP Damoiseaux: Department of Biology, Baylor University, Waco, United States; Duy Dang: Xavier University of Louisiana, New Orleans, United States; Samantha Danguilan: Biology, Smith College, Northampton, United States; Chuck Daniels: Microbiology, Ohio State University, Columbus, United States; Richard L Daniels: Biology, College of Idaho, Caldwell, United States; Benjamin Danner: Science, Cabrini College, Radnor, United States; Michael A Darcy: Biological Sciences, Carnegie Mellon University, Pittsburgh, United States; Marija Dargyte: University of California Santa Cruz, Santa Cruz, United States; Rachel Darko: Biology, Howard College, Washington, DC, United States; Adam Darwiche: Microbiology, Miami University, Oxford, United States; Aditya Das: Biological Sciences, Carnegie Mellon University, Pittsburgh, United States; Thressa DaSilva: Biology and Chemistry, Nyack College, Nyack, United States; Amina Dasin: Biological Sciences, University of North Texas, Denton, United States; Ekaterina Dasiuk: Biology, Gonzaga University, Spokane, United States; Cierra Dauenhauer: Biology, Gonzaga University, Spokane, United States; Sarahanne Davidson: Biology, University of Alabama Birmingham, Birmingham, United States; H Lane Davis: Biology, University of Louisiana at Monroe, Monroe, United States; Katie Davis: Biology, University of Louisiana at Monroe, Monroe, United States; Dillon Davis: Biology, University of Louisiana at Monroe, Monroe, United States; Jeremy S Davis: Biology, Washington University in St. Louis, St. Louis, United States; Kimberly Davis: University of California Santa Cruz, Santa Cruz, United States; Marshall Davis: Biology, Gonzaga University, Spokane, United States; William B Davis: Washington State University, Pullman, United States; Ariangela J Davis-Kozik: Biology, Calvin College, Grand Rapids, United States; Zazil-xa Davis-Vazquez: Biological Sciences and Geology, Queensboro Community College, Bayside, United States; Allie Day: Biology, Hope College, Holland, United States; Jelani Days: Biology, Culver-Stockton College, Canton, United States; Nydia De La Cruz: Biology, Loyola Marymount University, Los Angeles, United States; Jessica De La Luz: Biology, University of Texas at El Paso, El Paso, United States; Carla De Los Santos: University of California Santa Cruz, Santa Cruz, United States; Melissa De Mattos: University of California Santa Cruz, Santa Cruz, United States; Alexander J DeBernardo: Biology, Saint Joseph's University, Philadelphia, United States; LaJoyce Debro: Biology, Jacksonville State University, Jacksonville, United States; Samuel DeCero: Biology, Carthage College, Kenosha, United States; Sarah DeCou: Biology, Washington University in St. Louis, St. Louis, United States; Kimberly DeGlopper: Biology, Hope College, Holland, United States; Ari Dehn: Environmental and Biological Science, University of Maine, Machias, Machias, United States; Morgan Deihs: Microbiology, Miami University, Oxford, United States; Randall J DeJong: Biology, Calvin College, Grand Rapids, United States; Rafaelle Delaney: Morehouse College, Atlanta, United States; Alex Delenko: Biology, Gettysburg College, Gettysburg, United States; Veronique A Delesalle: Biology, Gettysburg College, Gettysburg, United States; Hilda Delgadillo: Biology, Loyola Marymount University, Los Angeles, United States; Zachary DeLong: Biology and Chemistry, Nyack College, Nyack, United States; Maggie DelPonte: Purdue University, West Lafayette, United States; Katherine E Deming: Biological Sciences, University of North Texas, Denton, United States; Cassondra Demming: Biology, College of St. Scholastica, Duluth, United States; Renee M Deneweth: Honors Program, Florida Gulf Coast University, Fort Myers, United States; Lisa Deng: Biology, Washington University in St. Louis, St. Louis, United States; Danielle M DeNigris: Biology, Saint Joseph's University, Philadelphia, United States; John J Dennehy: Biology, CUNY, Queens College, Queens, United States; Shannon Denny: Washington State University, Pullman, United States; Jonquil Dent: Biology, Howard College, Washington, DC, United States; Dee R Denver: Integrative Biology, Oregon State University, Corvallis, United States; Armelle DeRiso: Ohio State University, Columbus, United States; Payal P Desai: Biology, University of California San Diego, La Jolla, United States; Dana DeSantis: Biological Sciences, Lehigh University, Bethlehem, United States; Ria Deshpande: Biology, Smith College, Northampton, United States; Alana Deutsch: Biology, Washington University in St. Louis, St. Louis, United States; Veena Devaraju: Biological Sciences, University of North Texas, Denton, United States; Bethany DeVault: Biology and Chemistry, Nyack College, Nyack, United States; Sarah Devine: Biology, Smith College, Northampton, United States; Elizabeth B DeVore: Biology, Washington University in St. Louis, St. Louis, United States; Jodie DeVries: Biology, Calvin College, Grand Rapids,

United States; Brittney DeWald: Biology, Nebraska Wesleyan University, Lincoln, Nebraska, United States; Jaskirat Dhanoa: Department of Microbiology, Immunology, and Molecular Genetics, University of California, Los Angeles, Los Angeles, United States; Molly Diamond: Washington State University, Pullman, United States; Crystal Diaz: Biology, Illinois Wesleyan University, Bloomington, United States; Felix Diaz-Medero: Biology, University of Puerto Rico - Cayey, Cayey, United States; Mike DiCandia: Biology, Gettysburg College, Gettysburg, United States; Leon Dickson Jr: Biology, Howard College, Washington, DC, United States; Lauren Dieleman: Biology, Culver-Stockton College, Canton, United States; Joshua T Dimmick: Biology, Hampden-Sydney College, Farmville, United States; Sasha DiNitto: Biology, Trinity College, Hartford, United States; Luke E Diorio-Toth: Biological Sciences, Carnegie Mellon University, Pittsburgh, United States; Anthony Disteso: Montclair State University, Montclair, United States; Nikita Divekar: University of California Santa Cruz, Santa Cruz, United States; Michael DiVito: Biology, Merrimack College, North Andover, United States; Ravi Dixit: Microbiology and Biotechnology, North Carolina State University, Raleigh, United States; Andrea Doak: University of Colorado at Boulder, Boulder, United States; Joanne Dobbins: Western Kentucky University, Bowling Green, United States; Pamela Dockstader: Biology, College of Idaho, Caldwell, United States; Shea Dolan: Southern Connecticut State University, New Haven, United States; Ebony Domingo: Ohio State University, Columbus, United States; Bianca Dominguez: University of Florida, Gainsville, United States; Evaristo Dominguez-Rodriguez: Biology, University of Puerto Rico - Cayey, Cayey, United States; Julie Donna: Microbiology, Miami University, Oxford, United States; John Paul Donohue: University of California Santa Cruz, Santa Cruz, United States; Melindy Dorcin: Biology, Trinity College, Hartford, United States; Emilio Doring: Biology, University of Texas at El Paso, El Paso, United States; Robert Dorit: Biology, Smith College, Northampton, United States; Stanna Dorn: Biology, Hope College, Holland, United States; Viviana Doros-Bonciu: School of Science and Technology, Georgia Gwinnett College, Lawrenceville, United States; Amrit Dosanjh: Department of Microbiology, Immunology, and Molecular Genetics, University of California, Los Angeles, Los Angeles, United States; Meredith Doughty: Western Kentucky University, Bowling Green, United States; Jasmine Douglas: Xavier University of Louisiana, New Orleans, United States; Erin Doyle: Doane College, Crete, United States; Matthew Doyle: Science, Cabrini College, Radnor, United States; Nicolette Driscoll: Biology and Medicine, Brown University, Providence, United States; Karlee Driver: Western Kentucky University, Bowling Green, United States; Bayless E Drum: Department of Biology, Baylor University, Waco, United States; Alan Yicong Du: Biology, University of California San Diego, La Jolla, United States; Heloise Dubois: Providence College, Providence, United States; Madison Duckworth: Biology, University of Alabama Birmingham, Birmingham, United States; Emily Duex: Biology, Carthage College, Kenosha, United States; Mary Duff: Biology, College of Charleston, Charleston, United States; Jackie Duffy: Biology, Culver-Stockton College, Canton, United States; Jacob Dums: Biology, University of Wisconsin-River Falls, River Falls, United States; David Dunbar: Science, Cabrini College, Radnor, United States; Courtney Dunkerley: Biology, University of Alabama Birmingham, Birmingham, United States; Azaline Dunlap-Smith: Environmental and Biological Science, University of Maine, Machias, Machias, United States; Matthew Dunn: Microbiology, Miami University, Oxford, United States; Matthew Dunworth: Biology, Gettysburg College, Gettysburg, United States; Quoc-viet Duong: Biology, University of Louisiana at Monroe, Monroe, United States; Stephen Duong: The Evergreen State College, Olympia, United States; Ben Duplechain: Biology, University of Louisiana at Monroe, Monroe, United States; Mackenzie Durham: Washington State University, Pullman, United States; Ryan J Durham: Department of Biology, Baylor University, Waco, United States; Mieke Dykhouse: Biology, Calvin College, Grand Rapids, United States; Maciej Dzikowski: Biological Sciences and Geology, Queensboro Community College, Bayside, United States; Keith Earley: University of Colorado at Boulder, Boulder, United States; Brian J Early: Microbiology and Molecular Biology, Brigham Young University, Provo, United States; Nicole A Ebalo: Biology, Washington University in St. Louis, St. Louis, United States; Annelle Eben: Biology, Calvin College, Grand Rapids, United States; Erich Eberts: Biology, Loyola Marymount University, Los Angeles, United States; Eric Edewaard: Biology, Calvin College, Grand Rapids, United States; Nicholas P Edgington: Biology, Southern Connecticut State University, New Haven, United States; Jessica Edwards: Biology, Illinois Wesleyan University, Bloomington, United States; Maria Eguiluz: Biology, Hope College, Holland, United States; Bernadette M Eichman: Biology, Saint Joseph's University, Philadelphia, United States; Rachel Ekdahl: Biology, College of Charleston, Charleston, United States; Ashuvinee Elangovan: University of Colorado at Boulder, Boulder, United States;

Sarah CR Elgin: Biology, Washington University in St. Louis, St. Louis, United States; Shelby A Ellis: Honors Program, Florida Gulf Coast University, Fort Myers, United States; Catherine E Elorette: Biology, Saint Joseph's University, Philadelphia, United States; Moustafa ElSayed: Biology, University of Louisiana at Monroe, Monroe, United States; Shannon Ely: Montclair State University, Montclair, United States; Abby Emanuel: Biology, Ouachita Baptist University, Arkadelphia, United States; Nicholas Emard: Southern Connecticut State University, New Haven, United States; Jan A Enabore: Biology, College of Charleston, Charleston, United States; Pauline Encarnacion: Montclair State University, Montclair, United States; Nicole Enciso: Biology, Loyola Marymount University, Los Angeles, United States; Rachel Ende: Biology, Illinois Wesleyan University, Bloomington, United States; Abby Engelkes: Biology, Ouachita Baptist University, Arkadelphia, United States; Angela Engelsen: Biology, University of Louisiana at Monroe, Monroe, United States; Jessica M Engle: Microbiology and Molecular Biology, Brigham Young University, Provo, United States; Belle V English: Biological Sciences, Carnegie Mellon University, Pittsburgh, United States; Sandy Enriquez: Biological Sciences and Geology, Queensboro Community College, Bayside, United States; Elizabeth Ensink: Biology, Hope College, Holland, United States; Marcella L Erb: Biology, University of California San Diego, La Jolla, United States; Gereltuya Erdenejargal: University of Colorado at Boulder, Boulder, United States; Jessica Erlich: Biology, Washington University in St. Louis, St. Louis, United States; Dana Escareno-Linger: Division of Natural and Health Sciences, Seton Hill University, Greensburg, United States; Dulce Escobar: University of California Santa Cruz, Santa Cruz, United States; Joshua Esguerra: Biological Sciences and Geology, Queensboro Community College, Bayside, United States; Militza Espada-Ramos: Biology, University of Puerto Rico - Cayey, Cayey, United States; Kathryn Esposito: Biology, Loyola Marymount University, Los Angeles, United States; Lauren A Esposito: Biology, CUNY, Queens College, Queens, United States; Maria Esquinca: Biology, University of Texas at El Paso, El Paso, United States; Paige Estave: Xavier University of Louisiana, New Orleans, United States; Amanda Estes: Biotechnology, Southern Maine Community College, South Portland, United States; Crystal Estrada: Biology, University of California San Diego, La Jolla, United States; Yesenia Estrada-Rivera: Biology, University of Puerto Rico - Cayey, Cayey, United States; Ann-Scott Ettinger: Biology, Gonzaga University, Spokane, United States; Nicole Deanne Evangelista: Biology, University of California San Diego, La Jolla, United States; Jared Evans: Ohio State University, Columbus, United States; Mikala Evans: Biology, Washington University in St. Louis, St. Louis, United States; Tom Everding: Biology, Calvin College, Grand Rapids, United States; Mitchell Eyerman: Microbiology, Miami University, Oxford, United States; Daniel R Ezzo: Biology, Saint Joseph's University, Philadelphia, United States; Deborah Fadoju: Biology, Howard College, Washington, DC, United States; Mohammed Fahad: Biology, Howard College, Washington, DC, United States; J Grant Fahey: Biology, Calvin College, Grand Rapids, United States; Michael Falahat: Biology, University of Alabama Birmingham, Birmingham, United States; Emily Falch: Biology, University of Wisconsin-River Falls, River Falls, United States; Alexandra L Falk: Biological Sciences, Carnegie Mellon University, Pittsburgh, United States; Yiwen Fang: Biology, Loyola Marymount University, Los Angeles, United States; Michael Fangman: Biology, Gonzaga University, Spokane, United States; Jennifer Farina: Science, Cabrini College, Radnor, United States; Charles Newton Farmer: Biology, University of California San Diego, La Jolla, United States; Amal Farooq: Biological Sciences, University of North Texas, Denton, United States; Summer Farooq: University of California Santa Cruz, Santa Cruz, United States; Kanhai Farrakhan: Biology, Howard College, Washington, DC, United States; Elias Farran: Biology, University of Texas at El Paso, El Paso, United States; Joseph Farrell: ISBT, LaSalle University, Philadelphia, United States; Tolulope Fasoranti: Biological Sciences, University of Pittsburgh, Pittsburgh, United States; Mokunfope Fatukasi: Purdue University, West Lafayette, United States; Jonathan Faughn: Western Kentucky University, Bowling Green, United States; Emilio Feal: University of California Santa Cruz, Santa Cruz, United States; Cameron Feathers: Biological Sciences, Lehigh University, Bethlehem, United States; Melissa Feeney: Biological Sciences, Lehigh University, Bethlehem, United States; Joel Feldhake: Biology, Calvin College, Grand Rapids, United States; Zachery Feldker: Biology, Carthage College, Kenosha, United States; Juan Feliciano-Figueroa: Biology, University of Puerto Rico - Cayey, Cayey, United States; Celia Feng: University of Colorado at Boulder, Boulder, United States; Chelsea L Ferguson: Division of Natural and Health Sciences, Seton Hill University, Greensburg, United States; Jacquelyn R Ferguson: Biological Sciences, Carnegie Mellon University, Pittsburgh, United States; Asia Fernandes: Xavier University of Louisiana, New Orleans, United States; Alyka Glor Fernandez: University of Florida,

Gainsville, United States; Mariceli Fernandez-Martinez: Pedagogy, University of Puerto Rico - Cayey, Cayey, United States; Pilar Fernandez-Rodriguez: Biology, University of Puerto Rico - Cayey, Cayey, United States; Michelle Fernando: Providence College, Providence, United States; Aisha Ferrazares: Biology and Medicine, Brown University, Providence, United States; Gregory J Ferroni: Biology, Saint Joseph's University, Philadelphia, United States; Kyra Feuer: Biological Sciences, Lehigh University, Bethlehem, United States; Alex Fields: Biology, University of Louisiana at Monroe, Monroe, United States; Rachel Fieman: Biological Sciences, Lehigh University, Bethlehem, United States; Laura Z Filliger: Biological Sciences, Carnegie Mellon University, Pittsburgh, United States; Christy Fillman: University of Colorado at Boulder, Boulder, United States; Jared Filut: Ohio State University, Columbus, United States; Ann M Findley: Biology, University of Louisiana at Monroe, Monroe, United States; Adam W Fine: Biology, Montana Tech of the University of Montana, Butte, United States; Joseph Fiorenza: Montclair State University, Montclair, United States; Marlie Fisher: University of Colorado at Boulder, Boulder, United States; Joshua NB Fisher: Microbiology and Molecular Biology, Brigham Young University, Provo, United States; Jodi Fitzgerald: University of Florida, Gainsville, United States; Nicholas M Flaherty: Biology, Merrimack College, North Andover, United States; Brandon Flatgard: Biology, Gonzaga University, Spokane, United States; Taylor Fleet: Biology, Calvin College, Grand Rapids, United States; Robert Fleming: Biology, Trinity College, Hartford, United States; Alexandru Florea: School of Science and Technology, Georgia Gwinnett College, Lawrenceville, United States; Desirey Flores: Natural Sciences, Del Mar College, Corpus Christi, United States; Izamar Flores Castillo: Biological Sciences, University of North Texas, Denton, United States; Shikira Flounory: Xavier University of Louisiana, New Orleans, United States; Caroline E Flowers: Biological Sciences, Carnegie Mellon University, Pittsburgh, United States; Matthew Flowers: Morehouse College, Atlanta, United States; Kelsey Focht: Biology, Gonzaga University, Spokane, United States; Rose Fogliano: Biology, Gettysburg College, Gettysburg, United States; Chase Fong: Department of Microbiology, Immunology, and Molecular Genetics, University of California, Los Angeles, Los Angeles, United States; Lindsey M Fong: Biological Sciences, Carnegie Mellon University, Pittsburgh, United States; Amy Fontenot: Biology, University of Louisiana at Monroe, Monroe, United States; Lauren Ford: Biology, University of Louisiana at Monroe, Monroe, United States; Jacquelyn Ford: Xavier University of Louisiana, New Orleans, United States; Berencia Fore: Biology, Howard College, Washington, DC, United States; Rebecca Foreman: Biology, Washington University in St. Louis, St. Louis, United States; Kathryn M Forman: Biology, Illinois Wesleyan University, Bloomington, United States; Steven Forrester: Science, Cabrini College, Radnor, United States; Katherine S Forsyth: Biological Sciences, Carnegie Mellon University, Pittsburgh, United States; Mark H Forsyth: Biology, College of William and Mary, Williamsburg, United States; Gloria Foster: University of Maine, Honors College, Orono, United States; Lisa Anne Foster: Biology, Trinity College, Hartford, United States; Deitrick Fowler: Morehouse College, Atlanta, United States; Kristen Fowler: Xavier University of Louisiana, New Orleans, United States; Courtney Fox: Biology, University of Wisconsin-River Falls, River Falls, United States; Tyler M Fox: Biological Sciences, Carnegie Mellon University, Pittsburgh, United States; Janey Foxe: Washington State University, Pullman, United States; Ethan Fram: Biology, University of California San Diego, La Jolla, United States; Sarah Francisco: Biology, Gettysburg College, Gettysburg, United States; Rene D Francolini: Biological Sciences, Carnegie Mellon University, Pittsburgh, United States; Samantha Frangos: Biological Sciences, Lehigh University, Bethlehem, United States; Shanah Frankel: Ohio State University, Columbus, United States; Regina Frawley: The Evergreen State College, Olympia, United States; Ryan Frazier: Providence College, Providence, United States; Christina M Freeman: Biology, Saint Joseph's University, Philadelphia, United States; Carlyn Freeman: Biology, Gonzaga University, Spokane, United States; Vanessa Freitas: Biology, Gonzaga University, Spokane, United States; Stanislav Fridland: University of California Santa Cruz, Santa Cruz, United States; Iddo Friedberg: Microbiology, Miami University, Oxford, United States; Shawn Friel: Science, Cabrini College, Radnor, United States; Gabriel Michael Frischer: Biology, University of California San Diego, La Jolla, United States; Chadley D Froes: Biology, College of Charleston, Charleston, United States; Julia Froud: University of California Santa Cruz, Santa Cruz, United States; Aubree Frownfelter: Biology, Calvin College, Grand Rapids, United States; Megan Fruchte: Microbiology and Biotechnology, North Carolina State University, Raleigh, United States; Katherine Fu: Biology, Loyola Marymount University, Los Angeles, United States; Kayci Fudge: Biology, Gonzaga University, Spokane, United States; Ana Lucia Fuentes: Biology, Loyola Marymount University, Los Angeles, United States;

Chelsea Fulmore: Biology, North Carolina Central University, Durham, United States; Ho Yee Joyce Fung: Biology, Washington University in St. Louis, St. Louis, United States; Kaitlin Fusco: Biology, Loyola Marymount University, Los Angeles, United States; Deanna Fyffe: Microbiology, Miami University, Oxford, United States; Jamal Gaddis: Biology, Washington University in St. Louis, St. Louis, United States; Christopher Gager: Biology, Hope College, Holland, United States; Eliot Gagne: University of Maine, Honors College, Orono, United States; Miranda Gagnon: Biology, Merrimack College, North Andover, United States; Jasmine Gajeton: Department of Microbiology, Immunology, and Molecular Genetics, University of California, Los Angeles, Los Angeles, United States; Maria E Galassi: Biology, Saint Joseph's University, Philadelphia, United States; Ruth Galatowitsch: University of Florida, Gainesville, United States; Chanah Gallagher: University of Colorado at Boulder, Boulder, United States; Jordan Gallardo: Biology, University of Alabama Birmingham, Birmingham, United States; Isaura Gallegos: Washington State University, Pullman, United States; Jenna Galletta: Ohio State University, Columbus, United States; Tyler Galvelis: Biological Sciences, University of Pittsburgh, Pittsburgh, United States; Aakash Y Gandhi: Biology, Washington University in St. Louis, St. Louis, United States; Ryan Gandy: Purdue University, West Lafayette, United States; Danielle Gannon: Science, Cabrini College, Radnor, United States; Alejandra Garcia: Biology, University of Texas at El Paso, El Paso, United States; Carlos Garcia: Biological Sciences and Geology, Queensboro Community College, Bayside, United States; Karla Garcia: Biology, University of Texas at El Paso, El Paso, United States; Oscar Garcia: Biology, University of Texas at El Paso, El Paso, United States; Samuel Garcia: Biology, University of Texas at El Paso, El Paso, United States; Karla Garcia-Delgado: Biology, University of Puerto Rico - Cayey, Cayey, United States; Hernan Garcia-Ruiz: Molecular and Cell Biology Program, Oregon State University, Corvallis, United States; Adam V Gardner: Microbiology and Molecular Biology, Brigham Young University, Provo, United States; Jeremy Garner: Morehouse College, Atlanta, United States; Jacqualyn Garrett: The Evergreen State College, Olympia, United States; Logan Garthe: Biology, Illinois Wesleyan University, Bloomington, United States; Samantha M Gatt: Honors Program, Florida Gulf Coast University, Fort Myers, United States; Brianna Gaytan: Biology, Loyola Marymount University, Los Angeles, United States; Abraham Gebreselassie: Biology, Loyola Marymount University, Los Angeles, United States; Kaitlyn Geffen: Washington State University, Pullman, United States; Sarah Geiger: Microbiology, Miami University, Oxford, United States; Katelyn Geleynse: Biology, Calvin College, Grand Rapids, United States; Ethan Gelke: Biology, Illinois Wesleyan University, Bloomington, United States; William Gendron: Biology, Loyola Marymount University, Los Angeles, United States; Jessica Genkil: Biological Sciences, University of Pittsburgh, Pittsburgh, United States; Cody Gensen: Biology, University of Wisconsin-River Falls, River Falls, United States; Katia George: Biology, Smith College, Northampton, United States; Kara Geraci: Biology, Gonzaga University, Spokane, United States; Shaunt Gharabegian: Biology, Loyola Marymount University, Los Angeles, United States; Kamalini Ghosh: Biological Sciences, University of Pittsburgh, Pittsburgh, United States; Bryan C Gibbon: Department of Biology, Baylor University, Waco, United States; Zane Gibbs: Biological Sciences, University of North Texas, Denton, United States; Allison Gibson: Biology, Illinois Wesleyan University, Bloomington, United States; Katherine Giddens: Biology, Trinity College, Hartford, United States; Dometria Gilbert: Biology, Howard College, Washington, DC, United States; Neil Gilbert: Biology, Calvin College, Grand Rapids, United States; Claire Gillette: University of Colorado at Boulder, Boulder, United States; Brooke Gillispie: The Evergreen State College, Olympia, United States; Sinead Gilmore: Biology, James Madison University, Harrisonburg, United States; Meghana Ginjpalli: University of California Santa Cruz, Santa Cruz, United States; Chris R Gissendanner: Biology, University of Louisiana at Monroe, Monroe, United States; Jennifer Giulietti: Providence College, Providence, United States; Felipe Giuste: Biology, Washington University in St. Louis, St. Louis, United States; John Givler: Biology, Ouachita Baptist University, Arkadelphia, United States; Mitchell Go: Washington State University, Pullman, United States; Grayland W Godfrey: Biology, Hampden-Sydney College, Farmville, United States; Erich Goebel: Microbiology, Miami University, Oxford, United States; Eric A Goethe: Department of Biology, Baylor University, Waco, United States; Amber Goins: Natural Sciences, Del Mar College, Corpus Christi, United States; Urszula P Golebiewska: Biological Sciences and Geology, Queensboro Community College, Bayside, United States; Pawel Golyski: Biology and Medicine, Brown University, Providence, United States; Norma Gomez-Fuentes: Biology, University of Puerto Rico - Cayey, Cayey, United States; Germarie Gomez-Garcia: Biology, University of Puerto Rico - Cayey, Cayey, United States; Jessmarie Gonzales:

Biological Sciences, University of North Texas, Denton, United States; Alfredo Gonzalez: Providence College, Providence, United States; Gabriela Gonzalez: Department of Microbiology, Immunology, and Molecular Genetics, University of California, Los Angeles, Los Angeles, United States; Bridget Gonzalez: Montclair State University, Montclair, United States; Elizabeth D Gonzalez: ISBT, LaSalle University, Philadelphia, United States; Joshua Gonzalez: Ohio State University, Columbus, United States; Stephanie Gonzalez: Biology, University of Texas at El Paso, El Paso, United States; Yvonne Gonzalez: Biology, University of Texas at El Paso, El Paso, United States; Joshua Gonzalez-Berrios: Biology, University of Puerto Rico - Cayey, Cayey, United States; Leanabel Gonzalez-Menendez: Biology, University of Puerto Rico - Cayey, Cayey, United States; Karla Gonzalez-Pagan: Biology, University of Puerto Rico - Cayey, Cayey, United States; Benjamin Goodwin: Biology, Gonzaga University, Spokane, United States; Emma Goodwin: Department of Microbiology, Immunology, and Molecular Genetics, University of California, Los Angeles, Los Angeles, United States; Matthew Goodwin: Biology, College of Idaho, Caldwell, United States; Sean Goralski: Providence College, Providence, United States; Sonja Gorman: Biological Sciences, Lehigh University, Bethlehem, United States; Alexander Goss: Biology, Culver-Stockton College, Canton, United States; Maya Gotsat-senko: Ohio State University, Columbus, United States; Emily Gough: Ohio State University, Columbus, United States; Jayalakshmi A Govindan: Biological Sciences, Lehigh University, Bethlehem, United States; Hannah M Gowaty: Division of Natural and Health Sciences, Seton Hill University, Greensburg, United States; Jamie Gradishar: Biology, Illinois Wesleyan University, Bloomington, United States; Neshaun Grady: Biology, Howard College, Washington, DC, United States; Hannah Graff: Western Kentucky University, Bowling Green, United States; Allison Graine: Biological Sciences, University of Pittsburgh, Pittsburgh, United States; Jordan M Grainger: Biology, Washington University in St. Louis, St. Louis, United States; Melanie Grajales: Biological Sciences, Lehigh University, Bethlehem, United States; Jose Grajeda: Biology, University of Texas at El Paso, El Paso, United States; Marcela Grajeda: Biology, University of Texas at El Paso, El Paso, United States; Brittany Grandaw: Biology, University of Wisconsin-River Falls, River Falls, United States; John Grant: Biology, Carthage College, Kenosha, United States; Deborah Grant: Biological Sciences and Geology, Queensboro Community College, Bayside, United States; Racheal Granville: Biology, Nebraska Wesleyan University, Lincoln, Nebraska, United States; Kaitlynn Graven: Biology, University of Wisconsin-River Falls, River Falls, United States; Austin Graves: Microbiology and Biotechnology, North Carolina State University, Raleigh, United States; Andrea Graves: Biological Sciences, University of North Texas, Denton, United States; Abby Grawe: Biology, Culver-Stockton College, Canton, United States; Angela Gray: School of Science and Technology, Georgia Gwinnett College, Lawrenceville, United States; Ken Gray: The Evergreen State College, Olympia, United States; Kevin Gray: University of California Santa Cruz, Santa Cruz, United States; Veronica C Gray: Biology, College of William and Mary, Williamsburg, United States; Eric Greene: University of Colorado at Boulder, Boulder, United States; Kyla L Greenfield: Biology, Spelman College, Atlanta, United States; Stephanie Gregory: Ohio State University, Columbus, United States; Janelle Grendler: Environmental and Biological Science, University of Maine, Machias, Machias, United States; Jacob Gries: Biology, Gonzaga University, Spokane, United States; Phyllis Griffard: Natural Sciences, University of Houston-Downtown, Houston, United States; Andrew Griffin: Biology, Jacksonville State University, Jacksonville, United States; Alishia K Griffin: Department of Biology, Baylor University, Waco, United States; Maura Griffith: Biology, Trinity College, Hartford, United States; Phoebe Grijalva: Biology, University of Texas at El Paso, El Paso, United States; Wendy Grillo: Biology, North Carolina Central University, Durham, United States; Melvin Grimes: Biology, University of Louisiana at Monroe, Monroe, United States; Kimberly Grome: Biology, Gonzaga University, Spokane, United States; Julianne H Grose: Microbiology and Molecular Biology, Brigham Young University, Provo, United States; Oleg Gross: University of Maine, Honors College, Orono, United States; Adam Groth: Biology, University of Wisconsin-River Falls, River Falls, United States; Halle Grove: Biological Sciences, University of Pittsburgh, Pittsburgh, United States; Clayton Gruber: Microbiology and Biotechnology, North Carolina State University, Raleigh, United States; Katherine Zhen H Guan: University of California Santa Cruz, Santa Cruz, United States; Hebe Guardiola-Diaz: Biology, Trinity College, Hartford, United States; Brittany Guay: Marine Science, Southern Maine Community College, South Portland, United States; Genevieve Guerra: Biology, Loyola Marymount University, Los Angeles, United States; Stephanie L Guerra: Biological Sciences, Carnegie Mellon University, Pittsburgh, United States; Stuart W Guertin: Biological Sciences, Carnegie Mellon

University, Pittsburgh, United States; Nicole Guevara: Biology, University of Texas at El Paso, El Paso, United States; Ayele Gugssa: Biology, Howard College, Washington, DC, United States; Nancy Guild: University of Colorado at Boulder, Boulder, United States; Nathaniel Guilford: Biology, Gonzaga University, Spokane, United States; Desire Guillory: Xavier University of Louisiana, New Orleans, United States; Quentin Guillory: Xavier University of Louisiana, New Orleans, United States; Bridget Guiza: Biology, University of California San Diego, La Jolla, United States; Shan Gulamani: Biology, University of Alabama Birmingham, Birmingham, United States; Naomi D Gunawardena: Biological Sciences, Carnegie Mellon University, Pittsburgh, United States; Jinny Guo: Ohio State University, Columbus, United States; Stella Guo: Biology, Washington University in St. Louis, St. Louis, United States; Auroni Gupta: Biology, University of California San Diego, La Jolla, United States; Swati Gupta: Biology, CUNY, Queens College, Queens, United States; Eric L Gustafson: Biology, Washington University in St. Louis, St. Louis, United States; Nicholas Guthrie: Biology, Howard College, Washington, DC, United States; Alexandra Gutierrez: Biology, University of Texas at El Paso, El Paso, United States; Jesus Gutierrez: Biology, University of Texas at El Paso, El Paso, United States; Omar Gutierrez Ruiz: Biology, University of Texas at El Paso, El Paso, United States; Ivan Guzman: Department of Microbiology, Immunology, and Molecular Genetics, University of California, Los Angeles, Los Angeles, United States; Soo Jung Ha: Purdue University, West Lafayette, United States; Brandon Haake: Biology, Culver-Stockton College, Canton, United States; Jake Hackel: Biology and Chemistry, Nyack College, Nyack, United States; Alexander Hadik: Biology and Medicine, Brown University, Providence, United States; Kaylee Hagen: Biology, College of St. Scholastica, Duluth, United States; Ritika Halder: Department of Biology, Baylor University, Waco, United States; Richard H Hale: Biological Sciences, University of North Texas, Denton, United States; Emily Hall: Biology, College of St. Scholastica, Duluth, United States; Jeremy Hall: Western Kentucky University, Bowling Green, United States; Michelle Hall: Biology, Culver-Stockton College, Canton, United States; Andrew Halleran: Biology, College of William and Mary, Williamsburg, United States; Mitchell Hallman: Biology, Washington University in St. Louis, St. Louis, United States; Alexander Hallwachs: Biology, Washington University in St. Louis, St. Louis, United States; Catherine Halpern: Virginia Commonwealth University, Richmond, United States; Peter Hamar: The Evergreen State College, Olympia, United States; Jameel Hamdan: Biology, University of Texas at El Paso, El Paso, United States; Ariel Hamil: Department of Microbiology, Immunology, and Molecular Genetics, University of California, Los Angeles, Los Angeles, United States; Elizabeth Hamilton: Biology, Culver-Stockton College, Canton, United States; Kaitlin Hamilton: Biological Sciences, Carnegie Mellon University, Pittsburgh, United States; Taylor Hammock: Biology, College of Charleston, Charleston, United States; Latanya P Hammonds-Odie: School of Science and Technology, Georgia Gwinnett College, Lawrenceville, United States; Maxwell Hampton: Biology, College of Idaho, Caldwell, United States; Stephanie A Haney: Biology, College of Charleston, Charleston, United States; Adam W Hansen: Microbiology and Molecular Biology, Brigham Young University, Provo, United States; Brent Harbaugh: Ohio State University, Columbus, United States; Emily Hardison: University of California Santa Cruz, Santa Cruz, United States; Charles Hardnett: Biology, Spelman College, Atlanta, United States; Shyla L Hardwick: Biology, Spelman College, Atlanta, United States; Raven Hardy: Biology, Spelman College, Atlanta, United States; Victoria Hare: University of Colorado at Boulder, Boulder, United States; Raven Hargrove: Biology, University of Louisiana at Monroe, Monroe, United States; Nyema Harmon: Purdue University, West Lafayette, United States; Jeremy Harmson: Biology, University of Louisiana at Monroe, Monroe, United States; Jesse Haro: Biology, Loyola Marymount University, Los Angeles, United States; Jourdan Harper: Biology, University of Texas at El Paso, El Paso, United States; Ravyne Harper: Biology, North Carolina Central University, Durham, United States; Donyelle Harrigan: Biology and Chemistry, Nyack College, Nyack, United States; Alex Harris: Biology, Calvin College, Grand Rapids, United States; Celina Harris: Biology, Gettysburg College, Gettysburg, United States; Jamie Harris: Biology, College of William and Mary, Williamsburg, United States; Katrina Harris: University of Maine, Honors College, Orono, United States; Jenna N Harrison: Biology, Saint Joseph's University, Philadelphia, United States; Jon Harrison: Biological Sciences, Lehigh University, Bethlehem, United States; Melinda Harrison: Science, Cabrini College, Radnor, United States; Andrew Hart: University of Maine, Honors College, Orono, United States; Matthew Hartman: Biology, University of Texas at El Paso, El Paso, United States; Grant A Hartzog: University of California Santa Cruz, Santa Cruz, United States; Robert Harvell: Biology, Jacksonville State University, Jacksonville, United States; Jayla Harvey: Biology,

Howard College, Washington, DC, United States; Samuel E Harvey: Biology, College of William and Mary, Williamsburg, United States; Vivian Harvey: Biology, College of William and Mary, Williamsburg, United States; Patrick Hashiguchi: Biology, Gonzaga University, Spokane, United States; Hina Hashmi: University of Maine, Honors College, Orono, United States; Selena Hasircoglu: Biological Sciences, University of Pittsburgh, Pittsburgh, United States; Kyler J Haskell: Microbiology and Molecular Biology, Brigham Young University, Provo, United States; Rachel Hastert: University of California Santa Cruz, Santa Cruz, United States; Paige Hasty: Biology, Illinois Wesleyan University, Bloomington, United States; J Rob Hatherill: Natural Sciences, Del Mar College, Corpus Christi, United States; Sarah Hausmann: Microbiology, Miami University, Oxford, United States; Tyler Hawk: Morehouse College, Atlanta, United States; Christina B Hawkins: Biology, College of Charleston, Charleston, United States; Harry Hawthorne: Biology, University of Louisiana at Monroe, Monroe, United States; Samantha Hawtrey: Western Kentucky University, Bowling Green, United States; Kendra Hayden: Biology, Gettysburg College, Gettysburg, United States; Joseph Haydock: Biology, Gonzaga University, Spokane, United States; Mallorie Hayes: Biology, University of Louisiana at Monroe, Monroe, United States; Michael Hayes: School of Science and Technology, Georgia Gwinnett College, Lawrenceville, United States; Camilla Haynes: Biology, Howard College, Washington, DC, United States; Michaela Haynie: Southern Connecticut State University, New Haven, United States; Sarah Hays: Biology, University of Louisiana at Monroe, Monroe, United States; Diana He: Biology, Washington University in St. Louis, St. Louis, United States; Zezhong He: Biology, Washington University in St. Louis, St. Louis, United States; Kevin He: Biology, University of California San Diego, La Jolla, United States; Siping He: Biological Sciences, Carnegie Mellon University, Pittsburgh, United States; Kaitlin E Healy: Biological Sciences, Carnegie Mellon University, Pittsburgh, United States; Stacey Heaver: Biology, Gettysburg College, Gettysburg, United States; Emily Heckman: Biological Sciences, Lehigh University, Bethlehem, United States; Eric Hederstedt: Biology, Hope College, Holland, United States; Morgan Hefner: Environmental and Biological Science, University of Maine, Machias, Machias, United States; Erin M Hegarty: Department of Biology, Baylor University, Waco, United States; Neal Hegarty: Ohio State University, Columbus, United States; Cally Hein: Biology, University of Wisconsin-River Falls, River Falls, United States; Bryce Henderson: Washington State University, Pullman, United States; Melissa S Henderson: Department of Biology, Baylor University, Waco, United States; Hope Hendricks: Microbiology and Biotechnology, North Carolina State University, Raleigh, United States; Blake Henley: Biology, Gonzaga University, Spokane, United States; Colleen Henry: Microbiology, Miami University, Oxford, United States; Lanese Henry: Biology and Chemistry, Nyack College, Nyack, United States; Robert Henry: Biology, Hope College, Holland, United States; Shana-kay Henry-Grant: Biological Sciences and Geology, Queensboro Community College, Bayside, United States; Ellen Hensle: Biology, Gonzaga University, Spokane, United States; Ryan M Hensleigh: Chemistry, Montana Tech of the University of Montana, Butte, United States; Vanessa Hernandez: Biology, University of Texas at El Paso, El Paso, United States; Emma Herold: Biology and Medicine, Brown University, Providence, United States; April Hester: Biology, North Carolina Central University, Durham, United States; Sara A Heyn: Department of Biology, Baylor University, Waco, United States; Henry Higby: Biology, Calvin College, Grand Rapids, United States; Charles Highfield: Biology, Jacksonville State University, Jacksonville, United States; Gabriela Hilario: Montclair State University, Montclair, United States; Heather Hill: Microbiology and Biotechnology, North Carolina State University, Raleigh, United States; Rose Zabel Hill: Biology, University of California San Diego, La Jolla, United States; Sarah Hillson: University of Colorado at Boulder, Boulder, United States; Brenda E Hinojoza: Biology, University of Texas at El Paso, El Paso, United States; Daniel Hinson: Biology, College of Charleston, Charleston, United States; Traci-Lynn Hirai: Biology, Loyola Marymount University, Los Angeles, United States; Aspen Hirsch: Biology, Gonzaga University, Spokane, United States; Hana Hoang: Biology, College of Idaho, Caldwell, United States; Malayna Hocker: Biology, Smith College, Northampton, United States; Kimberly Hodgson: Biology, Hope College, Holland, United States; Krista Hoevemeyer: Biology, Hope College, Holland, United States; Taylor Hoff: University of Colorado at Boulder, Boulder, United States; Hilary Hoffman: University of California Santa Cruz, Santa Cruz, United States; Gina Hogan: Biology, University of Louisiana at Monroe, Monroe, United States; Sara E Hoge: Biological Sciences, Carnegie Mellon University, Pittsburgh, United States; Ryan Holden: Biology, Illinois Wesleyan University, Bloomington, United States; Alexis Holder: Biology, Gonzaga University, Spokane, United States; Cameron Holder: Biology, Hope College, Holland,

United States; Courtney Hollingsworth: Biology, Howard College, Washington, DC, United States; Johnathon Hollis: Morehouse College, Atlanta, United States; Gail Hollowell: Biology, North Carolina Central University, Durham, United States; Georgeanna M Holmes: School of Science and Technology, Georgia Gwinnett College, Lawrenceville, United States; Le'nica Holmes: School of Science and Technology, Georgia Gwinnett College, Lawrenceville, United States; Bethany Holtz: Biology, Gettysburg College, Gettysburg, United States; Nathan Holz: University of California Santa Cruz, Santa Cruz, United States; David Homan: University of California Santa Cruz, Santa Cruz, United States; Joris Hoogendoorn: Microbiology, Miami University, Oxford, United States; Stacy Hooker: Biology, Calvin College, Grand Rapids, United States; James Horsfall: University of California Santa Cruz, Santa Cruz, United States; David Daniel Horstman: Biology, University of California San Diego, La Jolla, United States; Ellen Hostert: Environmental and Biological Science, University of Maine, Machias, Machias, United States; Katherine Hotze: Biology, Carthage College, Kenosha, United States; Laken C Houser: Division of Natural and Health Sciences, Seton Hill University, Greensburg, United States; Catherine M Howard: Department of Biology, Baylor University, Waco, United States; Lauren A Howell: Biology, College of Charleston, Charleston, United States; Sunnie Hsiung: Biology, Washington University in St. Louis, St. Louis, United States; G Jason Huang: Biology, Washington University in St. Louis, St. Louis, United States; Tina Huang: University of California Santa Cruz, Santa Cruz, United States; Vincent J Huang: Biology, Washington University in St. Louis, St. Louis, United States; Jasmine Hubley: The Evergreen State College, Olympia, United States; Erica Hufford: Biology, University of Louisiana at Monroe, Monroe, United States; Raya Hughes: Biology, University of Louisiana at Monroe, Monroe, United States; Lee E Hughes: Biological Sciences, University of North Texas, Denton, United States; Emily Huizenga: Biology, Calvin College, Grand Rapids, United States; Nathaniel Hunnewell: Purdue University, West Lafayette, United States; Kellie Hunnicutt: Biology, Howard College, Washington, DC, United States; Gregory Hunt: Biology, Gonzaga University, Spokane, United States; Taylor Hunt: Providence College, Providence, United States; Kessler Hurd: Biology, Howard College, Washington, DC, United States; Mackenzie Hurlbert: Southern Connecticut State University, New Haven, United States; Kayla Hurst: Microbiology and Biotechnology, North Carolina State University, Raleigh, United States; Arturo Husein: Biological Sciences, University of North Texas, Denton, United States; Keith W Hutchison: Molecular and Biomedical Sciences, University of Maine, Honors College, Orono, United States; Sydney Hutton: Biology, Gonzaga University, Spokane, United States; Huyen Huynh: University of Colorado at Boulder, Boulder, United States; Stephanie Huynh: Biology, Smith College, Northampton, United States; Vicky Hwang: Biology, University of California San Diego, La Jolla, United States; Joshua Hynes: Western Kentucky University, Bowling Green, United States; Chikodi Ibe: University of Florida, Gainsville, United States; Aubrey Ibele: Biology, Gonzaga University, Spokane, United States; Catherine Ibrahim: Montclair State University, Montclair, United States; Amanda Icazatti-Burtell: Biology, University of Puerto Rico - Cayey, Cayey, United States; Torri Igou: University of California Santa Cruz, Santa Cruz, United States; Jasmine Ikejiani: School of Science and Technology, Georgia Gwinnett College, Lawrenceville, United States; Mary Illback: Biology, Gonzaga University, Spokane, United States; Emily Illingworth: University of Maine, Honors College, Orono, United States; Ariel Imler: Biology, College of Charleston, Charleston, United States; Joshua Imperial: Department of Microbiology, Immunology, and Molecular Genetics, University of California, Los Angeles, Los Angeles, United States; Emily J Interrante: Department of Biology, Baylor University, Waco, United States; John Ion: Biology, College of St. Scholastica, Duluth, United States; Andra Ionescu: Biology, University of California San Diego, La Jolla, United States; Khristina Ipapo: Department of Microbiology, Immunology, and Molecular Genetics, University of California, Los Angeles, Los Angeles, United States; Shubha K Ireland: Xavier University of Louisiana, New Orleans, United States; Camille E Irwin: Biology, Saint Joseph's University, Philadelphia, United States; Sharon Isern: Biological Sciences, Florida Gulf Coast University, Fort Myers, United States; Akira Issac: Biology, Howard College, Washington, DC, United States; Tambi F Issac: Microbiology and Molecular Biology, Brigham Young University, Provo, United States; Kristina Ivanova: University of Florida, Gainsville, United States; Varun Iyengar: Biological Sciences, University of Pittsburgh, Pittsburgh, United States; Osai Ize-Iyamu: Biology, Howard College, Washington, DC, United States; Antonio Jackson: Morehouse College, Atlanta, United States; Charity Jackson: Western Kentucky University, Bowling Green, United States; Stephen Jackson: Biology, University of Louisiana at Monroe, Monroe, United States; Jelissa Jackson: Biology, Howard College, Washington, DC, United

States; Naveen Jain: Biology, Washington University in St. Louis, St. Louis, United States; Marilyn Jalal: Biological Sciences, University of North Texas, Denton, United States; Aleksandar Jamborcic: Biology, University of California San Diego, La Jolla, United States; Brett James: Biology, Ouachita Baptist University, Arkadelphia, United States; Wendy Jamison: Physical and Life sciences, Chadron State College, Chadron, United States; Alicia Jancevski: Providence College, Providence, United States; Elaina Jandourek: Biology, Carthage College, Kenosha, United States; Emily Jankowski: Biology, Gettysburg College, Gettysburg, United States; Gary Janssen: Microbiology, Miami University, Oxford, United States; Jonathan W Jarvik: Biological Sciences, Carnegie Mellon University, Pittsburgh, United States; Paul G Jasinto: Biological Sciences, Carnegie Mellon University, Pittsburgh, United States; Rahul Jaswaney: Biology, Washington University in St. Louis, St. Louis, United States; Rohit Jaswaney: Biology, Washington University in St. Louis, St. Louis, United States; Kishore L Jayakumar: Biological Sciences, Carnegie Mellon University, Pittsburgh, United States; Alec Jeffers: Microbiology, Miami University, Oxford, United States; Myleka Jefferson: Biology, Howard College, Washington, DC, United States; Sharese Jefferson: Biology, College of William and Mary, Williamsburg, United States; Tori Jefferson: Microbiology and Biotechnology, North Carolina State University, Raleigh, United States; Sabrina Jen: Biology, Loyola Marymount University, Los Angeles, United States; Timothy Jen: Biology, Calvin College, Grand Rapids, United States; Adam C Jenkins: Biology, College of Charleston, Charleston, United States; Meagan M Jenkins: Honors Program, Florida Gulf Coast University, Fort Myers, United States; Valerie R Jenkins: Biology, Saint Joseph's University, Philadelphia, United States; Stephen Jensen: Biology, College of St. Scholastica, Duluth, United States; Chandler Jensen-Cody: Biology, Gonzaga University, Spokane, United States; Ji-il Jeon: Science, Cabrini College, Radnor, United States; Huiyi Jiang: University of California Santa Cruz, Santa Cruz, United States; Wenxuan Lilith Jiang: Department of Microbiology, Immunology, and Molecular Genetics, University of California, Los Angeles, Los Angeles, United States; Xuexia Jiang: Biological Sciences, Carnegie Mellon University, Pittsburgh, United States; Yi Jiang: Biological Sciences and Geology, Queensboro Community College, Bayside, United States; Allison A Johnson: Center for the Study of Biological Complexity, Virginia Commonwealth University, Richmond, United States; Anna Johnson: Biology, College of St. Scholastica, Duluth, United States; Ashley Marie Johnson: Biology, Washington University in St. Louis, St. Louis, United States; Brett Johnson: Biology, Jacksonville State University, Jacksonville, United States; Cody Johnson: The Evergreen State College, Olympia, United States; Christopher Johnson: Morehouse College, Atlanta, United States; Jessica Johnson: Xavier University of Louisiana, New Orleans, United States; Joseph E Johnson: Biology, Saint Joseph's University, Philadelphia, United States; Ember Johnson: The Evergreen State College, Olympia, United States; Benjamin K Johnson: Biology, Calvin College, Grand Rapids, United States; Katie Johnson: Biology, Culver-Stockton College, Canton, United States; Laura Johnson: Biology, Hope College, Holland, United States; Mark Johnson: Western Kentucky University, Bowling Green, United States; Nicholas Johnson: Biology, Nebraska Wesleyan University, Lincoln, Nebraska, United States; Rakiyah Johnson: Biology, Loyola Marymount University, Los Angeles, United States; Rachel Johnson: Biology, Culver-Stockton College, Canton, United States; Alexander Johnson: Biology, University of Alabama Birmingham, Birmingham, United States; Emily Johnston: Biological Sciences, University of North Texas, Denton, United States; Rachel Johnston: Biology, University of Louisiana at Monroe, Monroe, United States; Kevin Johnston: Biology, Gonzaga University, Spokane, United States; Netherland Joiner: Biology, Illinois Wesleyan University, Bloomington, United States; Alana Jones: Biology, Howard College, Washington, DC, United States; George Jones: Western Kentucky University, Bowling Green, United States; Isabel Jones: Washington State University, Pullman, United States; Jackson Jones: Biology, Gonzaga University, Spokane, United States; Nancy L Jones: ISBT, LaSalle University, Philadelphia, United States; J Derek Jones: Biology, University of Louisiana at Monroe, Monroe, United States; Keara Jones: Providence College, Providence, United States; Latoya Jones: Western Kentucky University, Bowling Green, United States; Mackenzie Jones: Western Kentucky University, Bowling Green, United States; Margaret Jones: Biology, Gonzaga University, Spokane, United States; Morgan Jones: Biological Sciences, University of North Texas, Denton, United States; Laura Joo: Biological Sciences and Geology, Queensboro Community College, Bayside, United States; Taylor Jordan: Biology, Gonzaga University, Spokane, United States; Caelan Jordan-Ferrer: Biology, University of Puerto Rico - Cayey, Cayey, United States; Michelle Juarez: Biological Sciences, Lehigh University, Bethlehem, United States; Billy Judd: The Evergreen State College, Olympia, United States; Mark Juel: Biology, Calvin College, Grand Rapids, United

States; Ryan Juhring: Science, Cabrini College, Radnor, United States; Harsha Jujjavarapu: Biology, Washington University in St. Louis, St. Louis, United States; John F Julian: Biology, Saint Joseph's University, Philadelphia, United States; Nickolas W Julian: Biology, Saint Joseph's University, Philadelphia, United States; Erin Jung: Biological Sciences, Carnegie Mellon University, Pittsburgh, United States; Jennifer Jung: Biological Sciences, University of North Texas, Denton, United States; Peter Jung: Biology, Trinity College, Hartford, United States; Jesus Jurado: Biology, University of Texas at El Paso, El Paso, United States; Ziomara Jurado: Biology, Nebraska Wesleyan University, Lincoln, Nebraska, United States; Nicholas Justus: Ohio State University, Columbus, United States; Kyle Kaczynski: Biology, Calvin College, Grand Rapids, United States; Day Nahm Kagy: Biology, University of California San Diego, La Jolla, United States; Maria Kakonikos: Biological Sciences and Geology, Queensboro Community College, Bayside, United States; Cody Kamstra: The Evergreen State College, Olympia, United States; Thalia Kanani-Hendijani: Biological Sciences, University of North Texas, Denton, United States; Karen Kantor: Virginia Commonwealth University, Richmond, United States; Aaron James Kappe: Biology, University of California San Diego, La Jolla, United States; Alexandra Karacozoff: Biology, Loyola Marymount University, Los Angeles, United States; Ann Karam: Biology, Gonzaga University, Spokane, United States; Lakshmi Karamsetty: University of Colorado at Boulder, Boulder, United States; Semir Karic: University of Florida, Gainsville, United States; Srilakshmi Karuturi: Virginia Commonwealth University, Richmond, United States; Surabhi Kasera: Biology, Washington University in St. Louis, St. Louis, United States; Woderyelesh Kassa: Montclair State University, Montclair, United States; Manuel Kassardjian: Biology, University of Louisiana at Monroe, Monroe, United States; Tomas Kasza: University of California Santa Cruz, Santa Cruz, United States; Dylan Katon: Environmental and Biological Science, University of Maine, Machias, Machias, United States; Balpreet Kaur: Ohio State University, Columbus, United States; Kevin Kaurich: Biology, James Madison University, Harrisonburg, United States; Mousa Kawwa: University of Florida, Gainsville, United States; Katelynn Kazane: Department of Microbiology, Immunology, and Molecular Genetics, University of California, Los Angeles, Los Angeles, United States; Michelle Keag: University of Colorado at Boulder, Boulder, United States; Sean Kearney: Purdue University, West Lafayette, United States; Alex Kearns: Western Kentucky University, Bowling Green, United States; Michael G Kearse: Biological Sciences, Lehigh University, Bethlehem, United States; Gabrielle Keeler: University of California Santa Cruz, Santa Cruz, United States; Nigel Keen: ISBT, LaSalle University, Philadelphia, United States; Ayleen Keeton: Ohio State University, Columbus, United States; Meghan Keleher: Biology, Trinity College, Hartford, United States; Chandler Keller: Washington State University, Pullman, United States; Paige Keller: School of Science and Technology, Georgia Gwinnett College, Lawrenceville, United States; Aubrey Kelley: Xavier University of Louisiana, New Orleans, United States; Brianna Kelley: Biology, University of Alabama Birmingham, Birmingham, United States; Tess Kelley: Biology, Illinois Wesleyan University, Bloomington, United States; Bobbi Kelling: Biology, University of Wisconsin-River Falls, River Falls, United States; Evan Kelly: Microbiology and Biotechnology, North Carolina State University, Raleigh, United States; Karla Kelly: The Evergreen State College, Olympia, United States; Jessica Kelsey: Department of Biological Sciences, University of Maryland, Baltimore County, Baltimore, United States; Margaret A Kenna: Biological Sciences, Lehigh University, Bethlehem, United States; Emma Kennedy: Biology, Loyola Marymount University, Los Angeles, United States; Erin Kennedy: Biological Sciences, Lehigh University, Bethlehem, United States; Kendall Kennedy: Biology, North Carolina Central University, Durham, United States; Lacey Kennedy: Biology, University of Alabama Birmingham, Birmingham, United States; Emily J Kenney: Biology, Washington University in St. Louis, St. Louis, United States; Samuel Kerk: Biology, Calvin College, Grand Rapids, United States; Matthew Kern: Biology and Chemistry, Nyack College, Nyack, United States; McKenzie Kerr: Biology, Loyola Marymount University, Los Angeles, United States; Mark Kerry: Biology, University of Louisiana at Monroe, Monroe, United States; Anna C Kesaris: Biology, Saint Joseph's University, Philadelphia, United States; Evan Kesinger: Biology, Culver-Stockton College, Canton, United States; Stecey B Kessel: School of Science and Technology, Georgia Gwinnett College, Lawrenceville, United States; Yousif Kettoola: Department of Microbiology, Immunology, and Molecular Genetics, University of California, Los Angeles, Los Angeles, United States; Joshua Keyes: University of Colorado at Boulder, Boulder, United States; Zenab Khan: Montclair State University, Montclair, United States; Sher Adam Khan: Biology, University of California San Diego, La Jolla, United States; Rohan Khazanchi: Biology, Washington University in St. Louis, St. Louis, United States; Elizaveta Khenner: Western Kentucky

University, Bowling Green, United States; Farzana Kibria: Biology, Smith College, Northampton, United States; Joshua Kiehl: Biology, Gettysburg College, Gettysburg, United States; Michael Kiflezghi: Virginia Commonwealth University, Richmond, United States; Albert Kiladjian: Biology, Trinity College, Hartford, United States; Chad Killen: Biology, Gettysburg College, Gettysburg, United States; Bradley W Killingsworth: Department of Biology, Baylor University, Waco, United States; Lauren Killough: Biology, Illinois Wesleyan University, Bloomington, United States; Myles S Killpatrick: Biological Sciences, Carnegie Mellon University, Pittsburgh, United States; Bryan Kim: Biological Sciences, Lehigh University, Bethlehem, United States; Justine S Kim: Biological Sciences, Carnegie Mellon University, Pittsburgh, United States; Deborah Kim: Biology and Chemistry, Nyack College, Nyack, United States; Nelly Kim: ISBT, LaSalle University, Philadelphia, United States; Young Kim: Biology, Trinity College, Hartford, United States; Jonathan Kindberg: Virginia Commonwealth University, Richmond, United States; Meghan Kinder: Biology, College of Idaho, Caldwell, United States; Allyson King: Western Kentucky University, Bowling Green, United States; Katie King: Biology, Ouachita Baptist University, Arkadelphia, United States; Tierra King: Biology, University of Louisiana at Monroe, Monroe, United States; Rodney A King: Western Kentucky University, Bowling Green, United States; Christina King Smith: Biology, Saint Joseph's University, Philadelphia, United States; Britni Kiosse: Southern Connecticut State University, New Haven, United States; Jonathan Kirk: Biology, Carthage College, Kenosha, United States; Ericka Kirkpatrick: University of Florida, Gainsville, United States; Joe Kirsch: The Evergreen State College, Olympia, United States; Taylor Kiskamp: Virginia Commonwealth University, Richmond, United States; Evan Kittaka: Ohio State University, Columbus, United States; Amy Kivela: Biology, Trinity College, Hartford, United States; Beth Klein: Biology, Carthage College, Kenosha, United States; Rachel Klein: Ohio State University, Columbus, United States; Holly Klepek: Microbiology, Miami University, Oxford, United States; Christine Kline: Biology and Chemistry, Nyack College, Nyack, United States; Josh Klinger: Purdue University, West Lafayette, United States; Linus Klingler: Natural Sciences, University of Houston-Downtown, Houston, United States; Tyler Klobucher: Biology, College of St. Scholastica, Duluth, United States; Karen Klyczek: Biology, University of Wisconsin-River Falls, River Falls, United States; Rachel Knapp: Biological Sciences, University of North Texas, Denton, United States; Anna Knight: Microbiology and Biotechnology, North Carolina State University, Raleigh, United States; Jacob Knol: Biology, Hope College, Holland, United States; Joshua Knopf: Biology, Trinity College, Hartford, United States; Andrew Knutson: Biology, University of Wisconsin-River Falls, River Falls, United States; Kevin Ko: Biology, Washington University in St. Louis, St. Louis, United States; Tae Wuk Ko: Department of Microbiology, Immunology, and Molecular Genetics, University of California, Los Angeles, Los Angeles, United States; Staci Koberstein: Biology, Gonzaga University, Spokane, United States; Ann P Koga: Biology, College of Idaho, Caldwell, United States; Anna Kogler: Biology, Washington University in St. Louis, St. Louis, United States; William Kohlway IV: Microbiology and Biotechnology, North Carolina State University, Raleigh, United States; Kristen Kokkonen: University of Colorado at Boulder, Boulder, United States; Christopher R Kolar: Department of Biology, Baylor University, Waco, United States; Ljuvica Kolich: Biology, College of Idaho, Caldwell, United States; Meghan Konda: Biological Sciences, University of North Texas, Denton, United States; Brett Konzek: Biology, Gonzaga University, Spokane, United States; Joseph Koon: Biology, Ouachita Baptist University, Arkadelphia, United States; Timothy Kopper: University of Colorado at Boulder, Boulder, United States; Christopher A Korey: Biology, College of Charleston, Charleston, United States; Kristen Kornsey: Biology, Gettysburg College, Gettysburg, United States; Michal Kozdronkiewicz: Biology, Illinois Wesleyan University, Bloomington, United States; Aleksey Kozlov: Biology, Gonzaga University, Spokane, United States; Marie-Morella Kponou: Biology, University of Wisconsin-River Falls, River Falls, United States; Allison Kraft: Biology, University of Wisconsin-River Falls, River Falls, United States; Lauren Kraft: Biological Sciences, Lehigh University, Bethlehem, United States; Zachary Kramer: Biological Sciences, University of Pittsburgh, Pittsburgh, United States; Arcadia Kratkiewicz: Biology, Smith College, Northampton, United States; Taylor Kratochvil: Biology, Gonzaga University, Spokane, United States; Aurora Kraus: Biology, Gonzaga University, Spokane, United States; Amy Krause: Biology, Washington University in St. Louis, St. Louis, United States; Jessica Kraut: Biology, Illinois Wesleyan University, Bloomington, United States; Amanda Kreger: Purdue University, West Lafayette, United States; Megan Kreitzman: Biology, University of Wisconsin-River Falls, River Falls, United States; Olivia Krejcarek: Biology, College of St. Scholastica, Duluth, United States; Jacqueline Kremer: University of California Santa Cruz, Santa Cruz, United

States; Laura Krings: Biology, Carthage College, Kenosha, United States; Grace Mahony Kroner: Biology, Washington University in St. Louis, St. Louis, United States; Greg P Krukonis: Biology, Gettysburg College, Gettysburg, United States; Alex Kryger: Biology, Loyola Marymount University, Los Angeles, United States; Lawrence Chihyung Ku: Biology, University of California San Diego, La Jolla, United States; James L Kuhn: Department of Biology, Baylor University, Waco, United States; Logan Kuhn: Biology, Ouachita Baptist University, Arkadelphia, United States; Gary Kuleck: Biology, Loyola Marymount University, Los Angeles, United States; Emil Kurian: Biology, University of Alabama Birmingham, Birmingham, United States; Meghan Kurz: Biology, University of Louisiana at Monroe, Monroe, United States; Thomas Kwak: Department of Microbiology, Immunology, and Molecular Genetics, University of California, Los Angeles, Los Angeles, United States; Samantha Kwok: University of Maine, Honors College, Orono, United States; Kaitlin Gee-Mei Kwong: Biology, University of California San Diego, La Jolla, United States; Shelby Labe: Biology, Trinity College, Hartford, United States; Fabiani Laboy-De-Jesus: Biology, University of Puerto Rico - Cayey, Cayey, United States; Holly LaFerriere: Biology, Culver-Stockton College, Canton, United States; Maryanne LaFollette: University of Maine, Honors College, Orono, United States; Caitlin Lahey: Purdue University, West Lafayette, United States; Lindsay A Lahoda: Biology, Saint Joseph's University, Philadelphia, United States; Peggy Lai: Biological Sciences, Lehigh University, Bethlehem, United States; Gilbert Laim: Department of Microbiology, Immunology, and Molecular Genetics, University of California, Los Angeles, Los Angeles, United States; Christian Laing: Biology, Wilkes University, Wilkes-Barre, United States; Kelly Laird: Biology, Gonzaga University, Spokane, United States; Conner Lajoie: University of Maine, Honors College, Orono, United States; Zachery Lake: Purdue University, West Lafayette, United States; Dieter Ka Yeung Lam: Biology, University of California San Diego, La Jolla, United States; Yuwan Lam: Biology, Trinity College, Hartford, United States; Nicole Lamas: University of Florida, Gainsville, United States; Evan Lambert: Biological Sciences, Lehigh University, Bethlehem, United States; Nicholas Lambrecht: Biology, College of St. Scholastica, Duluth, United States; Jessica Lambright: Biology, College of Idaho, Caldwell, United States; Anne Lamsa: Biology, University of California San Diego, La Jolla, United States; Ignacio Landaverde: Biology, Gettysburg College, Gettysburg, United States; Joseph F Lang: Honors Program, Florida Gulf Coast University, Fort Myers, United States; Jennifer Lang: Biology, Illinois Wesleyan University, Bloomington, United States; Jessica Lang: Biology, Calvin College, Grand Rapids, United States; Tyler Langenbrunner: The Evergreen State College, Olympia, United States; Allison Langone: Biology, Merrimack College, North Andover, United States; Elizabeth Lanum: Biology, Carthage College, Kenosha, United States; Jonathan Lapin: Biological Sciences, University of Pittsburgh, Pittsburgh, United States; Anthony Lapsansky: Biology, Gonzaga University, Spokane, United States; Erin Lapsansky: Biology, Gonzaga University, Spokane, United States; Kristina Lardner: Natural Sciences, Del Mar College, Corpus Christi, United States; Lisa Lark: Biological Sciences, Lehigh University, Bethlehem, United States; Tori Larsen: University of Florida, Gainsville, United States; Victoria Larsen: Biology, Howard College, Washington, DC, United States; Emily Larson: Washington State University, Pullman, United States; Victoria Larson: Biology, Trinity College, Hartford, United States; Julie Lau: Montclair State University, Montclair, United States; Charlotte Laven: Biology, Loyola Marymount University, Los Angeles, United States; Joseph Lawhead: Washington State University, Pullman, United States; Zachary Lawrence: Biology, University of Louisiana at Monroe, Monroe, United States; Stephen Layfield: Biology, University of Alabama Birmingham, Birmingham, United States; Manuel Lazos: Biology, University of Texas at El Paso, El Paso, United States; Derek Le: Department of Microbiology, Immunology, and Molecular Genetics, University of California, Los Angeles, Los Angeles, United States; Mimi Le: Biology and Medicine, Brown University, Providence, United States; John Leatherman: University of Florida, Gainsville, United States; Eric Lebel: Providence College, Providence, United States; Janine M LeBlanc-Straceski: Biology, Merrimack College, North Andover, United States; Dave Lee: Biology, Gonzaga University, Spokane, United States; HeaGie Lee: University of Colorado at Boulder, Boulder, United States; Ling-Ling Y Lee: Biological Sciences, Carnegie Mellon University, Pittsburgh, United States; Paul Lee: Biology, University of Alabama Birmingham, Birmingham, United States; Stacey Lee: Biological Sciences, Carnegie Mellon University, Pittsburgh, United States; Taewoo Lee: Department of Biology, Baylor University, Waco, United States; Lee Ying-Chiang Jeffrey: Biology, Washington University in St. Louis, St. Louis, United States; Julia Y Lee-Soety: Biology, Saint Joseph's University, Philadelphia, United States; Laquisha Leemow: Biology, North Carolina Central University, Durham, United States; Taylor

Leet: Western Kentucky University, Bowling Green, United States; Julie H Lench: Biology, College of Charleston, Charleston, United States; Donicia Lenori: Biology, Jacksonville State University, Jacksonville, United States; Toinette Lenori: Biology, Jacksonville State University, Jacksonville, United States; Justina Leo: Department of Microbiology, Immunology, and Molecular Genetics, University of California, Los Angeles, Los Angeles, United States; Brittany Leonard: Biology, Hope College, Holland, United States; Dean M Leonard: Department of Biology, Baylor University, Waco, United States; Jonathan Leonard: Biology, Gettysburg College, Gettysburg, United States; Felecia Leslie: Biology, North Carolina Central University, Durham, United States; Samantha Leslie: Microbiology, Miami University, Oxford, United States; Aden Lessiak: Biology, Gettysburg College, Gettysburg, United States; Robert J Lester: Biology, Montana Tech of the University of Montana, Butte, United States; Nobel LeTendre: Biology, Gonzaga University, Spokane, United States; Robin Levy: Biology, Spelman College, Atlanta, United States; Azlin Lewis: Western Kentucky University, Bowling Green, United States; Demi Lewis: Biology, Howard College, Washington, DC, United States; Jennifer Lewis: Biological Sciences, Lehigh University, Bethlehem, United States; Lynn O Lewis: Biological Sciences, University of Mary Washington, Fredericksburg, United States; Nakia Lewis: Biology, Jacksonville State University, Jacksonville, United States; Elizabeth Lewis Roberts: Biology, Southern Connecticut State University, New Haven, United States; Amber Li: Department of Microbiology, Immunology, and Molecular Genetics, University of California, Los Angeles, Los Angeles, United States; Diana Li: Biology, University of California San Diego, La Jolla, United States; Frances Li: University of Colorado at Boulder, Boulder, United States; Ardy Li: Ohio State University, Columbus, United States; Yi Li: Purdue University, West Lafayette, United States; Amy J Li: Biological Sciences, Carnegie Mellon University, Pittsburgh, United States; Vicky Li: Biology, Washington University in St. Louis, St. Louis, United States; Wei Li: Master of Physician Assistant Studies, Indiana Universiy, Indianapolis, United States; Wendy F Li: Biological Sciences, Carnegie Mellon University, Pittsburgh, United States; Benjamin Lichtenfels: Providence College, Providence, United States; Lauren Liddy: Montclair State University, Montclair, United States; Gretchen Lidicker: Biology, College of Charleston, Charleston, United States; Mitchell Lienemann: University of California Santa Cruz, Santa Cruz, United States; Morgan Light: Microbiology, Miami University, Oxford, United States; Jonathan Lin: Biology, Calvin College, Grand Rapids, United States; Tiffany Lin: Biology, Washington University in St. Louis, St. Louis, United States; Sawyer E Linke: Biology, Montana Tech of the University of Montana, Butte, United States; Amanda Links: Western Kentucky University, Bowling Green, United States; Leah C Liston: Purdue University, West Lafayette, United States; Megan Little: University of Maine, Honors College, Orono, United States; Kunyao Liu: Biology, Washington University in St. Louis, St. Louis, United States; Mingchun Liu: Biology, University of Alabama Birmingham, Birmingham, United States; Lisa Lizarraga: The Evergreen State College, Olympia, United States; Manuel Llano: Biology, University of Texas at El Paso, El Paso, United States; Maria Llanos: University of Colorado at Boulder, Boulder, United States; Ivan Llavona-Cartagena: Biology, University of Puerto Rico - Cayey, Cayey, United States; Ashley Lloyd: Ohio State University, Columbus, United States; Roxanne Lockhart: Biology, University of Alabama Birmingham, Birmingham, United States; Paola Lockwood: Biology, Loyola Marymount University, Los Angeles, United States; Kathryn E Loesser-Casey: Biological Sciences, University of Mary Washington, Fredericksburg, United States; Kathleen R Logan: Biology, Saint Joseph's University, Philadelphia, United States; Lebron Logan: Morehouse College, Atlanta, United States; Alexandra Lombard: Ohio State University, Columbus, United States; Laricca London: Microbiology, Howard College, Washington, DC, United States; Shawn C London: Biology, Saint Joseph's University, Philadelphia, United States; Angelina Londono-Joshi: Pathology, University of Alabama Birmingham, Birmingham, United States; Christopher B Long: Biology, College of Charleston, Charleston, United States; Michael Longmire: Biology, University of Alabama Birmingham, Birmingham, United States; Luther Loose: Biological Sciences, University of Pittsburgh, Pittsburgh, United States; Carlos Lopez: Biology, University of Texas at El Paso, El Paso, United States; A Javier Lopez: Biological Sciences, Carnegie Mellon University, Pittsburgh, United States; Juan Lopez: Natural Sciences, Del Mar College, Corpus Christi, United States; Naomy Lopez-Cuevas: Biology, University of Puerto Rico - Cayey, Cayey, United States; Todd Lorenz: Department of Microbiology, Immunology, and Molecular Genetics, University of California, Los Angeles, Los Angeles, United States; Brian Lott: University of California Santa Cruz, Santa Cruz, United States; Kevin K Lou: Biology, Washington University in St. Louis, St. Louis, United States; Lize Loubser: Biology, Hope College, Holland, United States; Dustin Lovas:

Biology, University of Louisiana at Monroe, Monroe, United States; Emily Love: Biology, Gettysburg College, Gettysburg, United States; Ryan Loviza: Virginia Commonwealth University, Richmond, United States; Abigail Lowery: University of Colorado at Boulder, Boulder, United States; Katelyn Lowery: Biology, University of Wisconsin-River Falls, River Falls, United States; Paola Lozada: Biological Sciences and Geology, Queensboro Community College, Bayside, United States; Alexander J Lu: Biology, Washington University in St. Louis, St. Louis, United States; Jacky W Lu: Biology, University of California San Diego, La Jolla, United States; Alexis Lubbers: Biology, University of Texas at El Paso, El Paso, United States; Jenna M Luczka: Division of Natural and Health Sciences, Seton Hill University, Greensburg, United States; Catherine M Ludwig: Biology, Washington University in St. Louis, St. Louis, United States; Sean Joseph Lund: Biology, University of California San Diego, La Jolla, United States; Emily Lundberg: Department of Microbiology, Immunology, and Molecular Genetics, University of California, Los Angeles, Los Angeles, United States; David Lunderberg: Biology, Hope College, Holland, United States; Bryce L Lunt: Microbiology and Molecular Biology, Brigham Young University, Provo, United States; Yan Luo: Purdue University, West Lafayette, United States; Lena N Lupey: Biology, Saint Joseph's University, Philadelphia, United States; Danielle Lutyk: Biology Department, Georgia State University, Milledgeville, United States; Brittany Lynch: Biology, North Carolina Central University, Durham, United States; Tatiana Lyons: Health Sciences, James Madison University, Harrisonburg, United States; Lena Ma: Biological Sciences, Lehigh University, Bethlehem, United States; Angela Maccarrone: Biology, Gonzaga University, Spokane, United States; Ana L Maccarrone: ISBT, LaSalle University, Philadelphia, United States; Margaret MacGibeny: Biology, Saint Joseph's University, Philadelphia, United States; Roger Machin-Rivera: Biology, University of Puerto Rico - Cayey, Cayey, United States; Maxwell Machurick: Biology, Carthage College, Kenosha, United States; Heyward B Mack: Biology, College of Charleston, Charleston, United States; Stephanie Mack: Biological Sciences, Lehigh University, Bethlehem, United States; Anne MacKenzie: Biology, Gonzaga University, Spokane, United States; Lara Madison: Physical and Life Sciences, Chadron State College, Chadron, United States; Tanya Maestas: Biology, University of Texas at El Paso, El Paso, United States; John Kevin Magarian: Biology, University of California San Diego, La Jolla, United States; Lauren Magee: Biology, Loyola Marymount University, Los Angeles, United States; Catherine M Mageeney: Biological Sciences, Lehigh University and Cabrini College, Bethlehem, United States; Samantha Magier: Biological Sciences, Lehigh University, Bethlehem, United States; Michael Majekodunmi: Montclair State University, Montclair, United States; Juan Maldonado-Rodriguez: Biology, University of Puerto Rico - Cayey, Cayey, United States; Natalia Maldonado-Vazquez: Biology, University of Puerto Rico - Cayey, Cayey, United States; Sarah Mam: Biology and Chemistry, Nyack College, Nyack, United States; Richa D Manglorkar: Department of Biology, Baylor University, Waco, United States; Coreen Manley: Biological Sciences, University of North Texas, Denton, United States; Janet Mansaray: Biology, Howard College, Washington, DC, United States; Diane Marcillo: Montclair State University, Montclair, United States; Joseph E Marcus: Biology, Washington University in St. Louis, St. Louis, United States; Matthew Mardis: Biology, University of Alabama Birmingham, Birmingham, United States; Cody J Marfizo: Honors Program, Florida Gulf Coast University, Fort Myers, United States; Alyssa Marian: Biology, Nebraska Wesleyan University, Lincoln, Nebraska, United States; Emil Maric: Biology, Illinois Wesleyan University, Bloomington, United States; Diane Marin: Biological Sciences and Geology, Queensboro Community College, Bayside, United States; Joseph Marin: Natural Sciences, Del Mar College, Corpus Christi, United States; Raven Mark: Biology, Howard College, Washington, DC, United States; Bryan Marone: The Evergreen State College, Olympia, United States; Anahi Marquez: Biology, University of Texas at El Paso, El Paso, United States; Nicholas Marquis: Marine Science, Southern Maine Community College, South Portland, United States; Paige Marsolek: Biology, College of St. Scholastica, Duluth, United States; Jordyn Mart: Biology, Culver-Stockton College, Canton, United States; Tehja Martin: Biology, Howard College, Washington, DC, United States; Alexander Martinez: University of Colorado at Boulder, Boulder, United States; Katherine Martinez: Ohio State University, Columbus, United States; Matthew Martinez: Biology, University of Texas at El Paso, El Paso, United States; Yanilda Martinez-Vega: Virginia Commonwealth University, Richmond, United States; Ryan Martinie: Biology, Calvin College, Grand Rapids, United States; Olivia A Martino: Biology, Saint Joseph's University, Philadelphia, United States; Victoria L Martino: Biology, Saint Joseph's University, Philadelphia, United States; Olumide A Martins: Biological Sciences, Carnegie Mellon University, Pittsburgh, United States; Jutta Y Marzillier:

Biological Sciences, Lehigh University, Bethlehem, United States; John Mason: Purdue University, West Lafayette, United States; Siiri M Mason: Division of Natural and Health Sciences, Seton Hill University, Greensburg, United States; Benjamin Massat: Biology, Carthage College, Kenosha, United States; Jorge Mata: The Evergreen State College, Olympia, United States; James Matern: Biology, Gonzaga University, Spokane, United States; Reny M Mathew: Department of Biology, Baylor University, Waco, United States; Oliver Mathrani: Biology, Gonzaga University, Spokane, United States; Hedieh Matinrad: Biology, University of California San Diego, La Jolla, United States; Andrea Matthew: Biology, Hope College, Holland, United States; Danielle Mauch: Biology, Loyola Marymount University, Los Angeles, United States; Kimberly Maurer: Biology, Gonzaga University, Spokane, United States; Jadae A Maxton: Biology, Saint Joseph's University, Philadelphia, United States; Jessica Maya: Biology, University of Alabama Birmingham, Birmingham, United States; Christine Mayer: Microbiology and Biotechnology, North Carolina State University, Raleigh, United States; Eric Mayer: Microbiology and Biotechnology, North Carolina State University, Raleigh, United States; Luke Maynard: Biology, Trinity College, Hartford, United States; Rebecca Catherine Mazahreh: Biology, University of California San Diego, La Jolla, United States; Katherine McArthur: Biology, Trinity College, Hartford, United States; Jim Mcburney-lin: Biology, University of California San Diego, La Jolla, United States; Kirk McCall: Xavier University of Louisiana, New OrleansUnited States; Christina McChesney: Microbiology and Biotechnology, North Carolina State University, Raleigh, United States; Casey A McConnell: Biology, Montana Tech of the University of Montana, Butte, United States; Shanika McCray: Biology, University of Louisiana at Monroe, Monroe, United States; Kyle McCurdy: Microbiology, Miami University, Oxford, United States; Taashaylaray McDuffie: Biology, Howard College, Washington, DC, United States; Reshard McElrath: Morehouse College, Atlanta, United States; Kyren McGary: Morehouse College, Atlanta, United States; Earyn McGee: Biology, Howard College, Washington, DC, United States; Breann McGee: Purdue University, West Lafayette, United States; Hayley McGinnis: The Evergreen State College, Olympia, United States; Erik McGuire: Biology, University of Alabama Birmingham, Birmingham, United States; Dustin McHugh: Environmental and Biological Science, University of Maine, Machias, Machias, United States; Kristina McInnes: Biology, Nebraska Wesleyan University, Lincoln, Nebraska, United States; Emily McKee: Biology, College of Charleston, Charleston, United States; Melanie McKell: Microbiology, Miami University, Oxford, United States; Angela L McKinney: Biology, Nebraska Wesleyan University, Lincoln, Nebraska, United States; Kelly E McKinnon: School of Science and Technology, Georgia Gwinnett College, Lawrenceville, United States; Lisa McLellan: Biology, Hope College, Holland, United States; Connor W McMahon: Honors Program, Florida Gulf Coast University, Fort Myers, United States; Sean McMahon: Biology, University of Alabama Birmingham, Birmingham, United States; Thomas McNamara: Biology, Trinity College, Hartford, United States; Lauren McNulty: Ohio State University, Columbus, United States; Abigail McPherson: Biological Sciences, University of Pittsburgh, Pittsburgh, United States; Erin E McPherson: Biology, College of Charleston, Charleston, United States; Briana McRae: Biology, North Carolina Central University, Durham, United States; Jacqueline Measer: Biology, Loyola Marymount University, Los Angeles, United States; Andrew J Medenbach: Biological Sciences, Carnegie Mellon University, Pittsburgh, United States; Stephanie Medina-Delgado: Biology, University of Puerto Rico - Cayey, Cayey, United States; Theodore Medling: Biology, Loyola Marymount University, Los Angeles, United States; Anna M Meese: Department of Biology, Baylor University, Waco, United States; Joseph Meggs: Biology, Jacksonville State University, Jacksonville, United States; Haley Mehalik: Ohio State University, Columbus, United States; Christopher Meier: Department of Biological Sciences, Genetics Course - Univeristy of Pittsburgh, Pittsburgh, United States; Trevor Meindertsma: Biology, Calvin College, Grand Rapids, United States; Riley Meister: Biology, Gonzaga University, Spokane, United States; Nisa Melendez-Rodriguez: Biology, University of Puerto Rico - Cayey, Cayey, United States; Brett Mellbye: Molecular and Cell Biology Program, Oregon State University, Corvallis, United States; Kimberly Menendez: Natural Sciences, University of Houston-Downtown, Houston, United States; Monica Menkis: University of Florida, Gainsville, United States; Theresa Menna: Biology, Gettysburg College, Gettysburg, United States; Akshay Mentreddy: Biology, University of Alabama Birmingham, Birmingham, United States; Seth Menzer: Biology, Hope College, Holland, United States; Alvin Mercado: Montclair State University, Montclair, United States; Raysean Mercer: Biological Sciences and Geology, Queensboro Community College, Bayside, United States; Paul Merlau: Biological Sciences, University of North Texas, Denton, United States; Bryan D Merrill: Microbiology and

Molecular Biology, Brigham Young University, Provo, United States; Brenden Merriman: Biology, Hope College, Holland, United States; Robert Merritt: Biology, Smith College, Northampton, United States; Nairobi Mesa: Montclair State University, Montclair, United States; Adriana Messyasz: Montclair State University, Montclair, United States; Maeva Metz: Microbiology, Miami University, Oxford, United States; Eli Metzler-Prieb: Purdue University, West Lafayette, United States; Danielle Meyer: Biology, Hope College, Holland, United States; Jordan Meyer: Department of Microbiology, Immunology, and Molecular Genetics, University of California, Los Angeles, Los Angeles, United States; Malcolm Mezue: Xavier University of Louisiana, New Orleans, United States; Scott F Michael: Biological Sciences, Florida Gulf Coast University, Fort Myers, United States; Steven Micheletti: Washington State University, Pullman, United States; Nicole Michmerhuizen: Biology, Calvin College, Grand Rapids, United States; Michael Miera: Biology, Gonzaga University, Spokane, United States; Olivia Miess: Biology, Gonzaga University, Spokane, United States; John Mike: Biology, College of St. Scholastica, Duluth, United States; Devon Miles: Microbiology and Biotechnology, North Carolina State University, Raleigh, United States; Adela Miller: University of Colorado at Boulder, Boulder, United States; Barrett Miller: Biological Sciences, Lehigh University, Bethlehem, United States; Brenda Miller: Biology, Illinois Wesleyan University, Bloomington, United States; Dustin Miller: The Evergreen State College, Olympia, United States; Eric S Miller: Microbiology and Biotechnology, North Carolina State University, Raleigh, United States; Lindsay Miller: Biological Sciences, Lehigh University, Bethlehem, United States; Brittany Miller: Biology, University of Louisiana at Monroe, Monroe, United States; Savannah Miller: Biology, Gettysburg College, Gettysburg, United States; Mistie Miller: Biology, Culver-Stockton College, Canton, United States; Meredith Millman: Biology, Smith College, Northampton, United States; Shelby Mills: Biology, Gonzaga University, Spokane, United States; Taylor Mills: University of Colorado at Boulder, Boulder, United States; Mariah Minder: Biology, Gonzaga University, Spokane, United States; Thomas Minkiewitz: Washington State University, Pullman, United States; Briaunna Minor: Xavier University of Louisiana, New Orleans, United States; Jessica Minor: Xavier University of Louisiana, New Orleans, United States; Crystal Miranda-Mendoza: Biology, University of Puerto Rico - Cayey, Cayey, United States; Lauren Misel: Microbiology, Miami University, Oxford, United States; Brianna Mishler: Biology, Gonzaga University, Spokane, United States; Nikita Mishra: Biological Sciences, Carnegie Mellon University, Pittsburgh, United States; Neil Mistry: Virginia Commonwealth University, Richmond, United States; Andreas Mitchell: Biology, Washington University in St. Louis, St. Louis, United States; Casey Mitchell: Biology, College of St. Scholastica, Duluth, United States; Dan Mize: Microbiology, Miami University, Oxford, United States; Jordan Moberg Parker: Department of Microbiology, Immunology, and Molecular Genetics, University of California, Los Angeles, Los Angeles, United States; Samantha Moffat: Biology, Hope College, Holland, United States; Kim Mogen: Biology, University of Wisconsin-River Falls, River Falls, United States; Jay Mohan: Biology, Washington University in St. Louis, St. Louis, United States; Tal Mohn: Biology, University of Wisconsin-River Falls, River Falls, United States; Zaffiro Molina-Rivera: Biology, University of Puerto Rico - Cayey, Cayey, United States; Molly Mollica: Ohio State University, Columbus, United States; Sally D Molloy: Molecular and Biomedical Sciences, University of Maine, Honors College, Orono, United States; Kirsten Monsen-Collar: Montclair State University, Montclair, United States; Erin Montgomery: Biology, College of St. Scholastica, Duluth, United States; Greg Montgomery: Biology, College of Idaho, Caldwell, United States; Denise Monti: Biology, University of Alabama Birmingham, Birmingham, United States; Steven Montiel-Melgar: Biological Sciences and Geology, Queensboro Community College, Bayside, United States; Nina Montoya: Biology, Gonzaga University, Spokane, United States; Austin L Moody: Biology, Montana Tech of the University of Montana, Butte, United States; Joshua Moore: Morehouse College, Atlanta, United States; Justin Moore: The Evergreen State College, Olympia, United States; Tiffany Moore: Ohio State University, Columbus, United States; Christina Moore: Microbiology, Miami University, Oxford, United States; Jose Morales: University of California Santa Cruz, Santa Cruz, United States; Marjorie Morales: Biological Sciences and Geology, Queensboro Community College, Bayside, United States; Abimelec Morales-Rivera: Biology, University of Puerto Rico - Cayey, Cayey, United States; Deborah Moran: Science, Cabrini College, Radnor, United States; Lauren Mordukhaev: Biology, CUNY, Queens College, Queens, United States; Robert Morefield: Marine Science, Southern Maine Community College, South Portland, United States; Benji Morehead: Biology, University of Louisiana at Monroe, Monroe, United States; Jonathan Morena: Ohio State University, Columbus, United States; Cameron Morford: Biology, Culver-Stockton College, Canton, United States;

Alexandra Morgan: University of Colorado at Boulder, Boulder, United States; Brandon Morgan: Biology, University of Louisiana at Monroe, Monroe, United States; Miranda Nicole Morgan: Biology, University of California San Diego, La Jolla, United States; Gabrielle Morgan: Biology, University of Louisiana at Monroe, Monroe, United States; Yolanda Morgan: Biology and Chemistry, Nyack College, Nyack, United States; John D Morrell: Microbiology and Molecular Biology, Brigham Young University, Provo, United States; Carlye Morris: Biology, University of Wisconsin-River Falls, River Falls, United States; Lailonny Y Morris: Biological Sciences, Carnegie Mellon University, Pittsburgh, United States; Tessa Morris: Biology, Loyola Marymount University, Los Angeles, United States; Tara Morton: Purdue University, West Lafayette, United States; Jordan Mosier: Biological Sciences, University of North Texas, Denton, United States; Allegra Mosley: Biology, Howard College, Washington, DC, United States; Jonah Moss: University of Colorado at Boulder, Boulder, United States; Sarah Mosteller: Ohio State University, Columbus, United States; Maroutcha Mouawad: Biology, University of Louisiana at Monroe, Monroe, United States; Christopher Mulhern: Biology, Trinity College, Hartford, United States; Stefanie E Mundhenk: Department of Biology, Baylor University, Waco, United States; Trapper Munn: Biology, University of Louisiana at Monroe, Monroe, United States; David Munoz: Biology, University of Texas at El Paso, El Paso, United States; Miranda J Munoz: Biological Sciences, Carnegie Mellon University, Pittsburgh, United States; Ernesto Munoz-Pena: Biology, University of Puerto Rico - Cayey, Cayey, United States; Chris Murdock: Biology, Jacksonville State University, Jacksonville, United States; Meghan M Muretta: Biology, Saint Joseph's University, Philadelphia, United States; Alexandra Murphy: Biology, Gonzaga University, Spokane, United States; Ashley Murphy: Biology, College of William and Mary, Williamsburg, United States; Brendan Murphy: Providence College, Providence, United States; Jason Murphy: Biology, Illinois Wesleyan University, Bloomington, United States; Zachery Murphy: Ohio State University, Columbus, United States; Cara Murphy: Microbiology, Miami University, Oxford, United States; Ryan Murphy: Biology, College of Charleston, Charleston, United States; Jessica Murray: Biology, College of William and Mary, Williamsburg, United States; Rachel Murray: Biology, Carthage College, Kenosha, United States; Brandon C Mus: Biology, Montana Tech of the University of Montana, Butte, United States; Munia M Mustafa: Biology, Illinois Wesleyan University, Bloomington, United States; Justin Muste: Biology, Washington University in St. Louis, St. Louis, United States; Rebecca Muthalaly: Biology, University of Alabama Birmingham, Birmingham, United States; Mallory E Myers: Department of Biology, Baylor University, Waco, United States; Samantha Midori Nadeau: Biology, University of California San Diego, La Jolla, United States; Colby Nance: Biology, University of Alabama Birmingham, Birmingham, United States; Joshua Napial: Biology, Loyola Marymount University, Los Angeles, United States; Alberto Napuli: The Evergreen State College, Olympia, United States; Ryan Narbutas: Department of Microbiology, Immunology, and Molecular Genetics, University of California, Los Angeles, Los Angeles, United States; Courtney Navarro: Biology, University of Texas at El Paso, El Paso, United States; Mercedes Navarro-Ohara: Biology, University of Texas at El Paso, El Paso, United States; Saba Nawaz: Biology, Howard College, Washington, DC, United States; Ann Nduati: Montclair State University, Montclair, United States; Kevin Ndukwe: Montclair State University, Montclair, United States; Kelli Neal: Biology, University of Louisiana at Monroe, Monroe, United States; Robert Neblett: Biology, Howard College, Washington, DC, United States; Mekala Kavya Neelakantan: Biology, University of California San Diego, La Jolla, United States; Om Neelay: Biology, Gonzaga University, Spokane, United States; Fathom Neft: University of California Santa Cruz, Santa Cruz, United States; Nicholas Negretti: Washington State University, Pullman, United States; Casey Neisius: Biology, University of Wisconsin-River Falls, River Falls, United States; James J Neitzel: The Evergreen State College, Olympia, United States; Brock Nelson: Biology, Gonzaga University, Spokane, United States; Matthew Nelson: University of Colorado at Boulder, Boulder, United States; Peter R Nelson: Arts and Sciences Division, University of Maine, Fort Kent, Fort Kent, United States; Samantha Nelson: University of Colorado at Boulder, Boulder, United States; Victoria A Nelson: Department of Biology, Baylor University, Waco, United States; Rahmi Nemri: Biology, Gonzaga University, Spokane, United States; Julie Nessler: Biology, Ouachita Baptist University, Arkadelphia, United States; Paul Nestor: Microbiology, Miami University, Oxford, United States; Elizabeth K Neumann: Department of Biology, Baylor University, Waco, United States; Ann Neumeyer: Biology, University of Wisconsin-River Falls, River Falls, United States; Jordan Newhof: Biology, Calvin College, Grand Rapids, United States; Alysha Newsom: Biology, Carthage College, Kenosha, United States; Octavius Newsome: Morehouse College, Atlanta, United States; Carmen

Ng: Department of Microbiology, Immunology, and Molecular Genetics, University of California, Los Angeles, Los Angeles, United States; Aundrea Nguyen: School of Science and Technology, Georgia Gwinnett College, Lawrenceville, United States; Minh Nguyen: Biology, Howard College, Washington, DC, United States; Emilie Nguyen: Biology, University of California San Diego, La Jolla, United States; Emily Dao Anh Nguyen: Biology, University of California San Diego, La Jolla, United States; Josephina Nguyen: Department of Microbiology, Immunology, and Molecular Genetics, University of California, Los Angeles, Los Angeles, United States; Katrina Nguyen: Biology, University of California San Diego, La Jolla, United States; Katrina Nguyen: Biology, University of California San Diego, La Jolla, United States; Lida K Nguyen: School of Science and Technology, Georgia Gwinnett College, Lawrenceville, United States; Christine Nguyen: Biology, University of Alabama Birmingham, Birmingham, United States; Thai Nguyen: Biology, University of Louisiana at Monroe, Monroe, United States; Catherine Nicholas: University of Colorado at Boulder, Boulder, United States; Conor P Nichols: ISBT, LaSalle University, Philadelphia, United States; Janet Nickels: University of California Santa Cruz, Santa Cruz, United States; Rachel E Nicoletto: Biology, Saint Joseph's University, Philadelphia, United States; Natalie Nigg: Biology, Gonzaga University, Spokane, United States; Yana A Nikolayeva: School of Science and Technology, Georgia Gwinnett College, Lawrenceville, United States; Ryan Nintzel: Biology, Gonzaga University, Spokane, United States; Emily Noble: Biology, University of Louisiana at Monroe, Monroe, United States; Ron Noble: Purdue University, West Lafayette, United States; Emily Nolting: Biology, University of Wisconsin-River Falls, River Falls, United States; Poochit Nonejuie: Biology, University of California San Diego, La Jolla, United States; Heather Nootbar: Ohio State University, Columbus, United States; Megan North: Biological Sciences, University of Pittsburgh, Pittsburgh, United States; Gina Notaro: Biological Sciences, Lehigh University, Bethlehem, United States; Adam Novajovsky: Biology, Gonzaga University, Spokane, United States; Peter Novick: Biology, CUNY, Queens College, Queens, United States; Daniel Andre Novoa: Biological Sciences and Geology, Queensboro Community College, Bayside, United States; Ian Noyes: Biology, Calvin College, Grand Rapids, United States; Amy Nusbaum: Washington State University, Pullman, United States; Brenna Nye: Biology, University of Alabama Birmingham, Birmingham, United States; James O'Brien: Providence College, Providence, United States; Heather O'Connell: Biology, Hope College, Holland, United States; Delaney O'Malley: Washington State University, Pullman, United States; Lauren P O'Neil: Biological Sciences, Carnegie Mellon University, Pittsburgh, United States; Kayla O'Sullivan: Biology, Gonzaga University, Spokane, United States; Stacey A Oakes: Biology, Saint Joseph's University, Philadelphia, United States; Nadeem Obaydou: Montclair State University, Montclair, United States; Kasey A Ober: ISBT, LaSalle University, Philadelphia, United States; Trevor Obrinsky: Biology, Gonzaga University, Spokane, United States; Natalia Ocasio-Ramos: Biology, University of Puerto Rico - Cayey, Cayey, United States; Raul Ochoa: Biology, University of Texas at El Paso, El Paso, United States; Richard Ode: Montclair State University, Montclair, United States; Nwamara Ogbonna: Biology, Howard College, Washington, DC, United States; Marcus Oglesby: Morehouse College, Atlanta, United States; Kasopefoluwa Oguntuyo: Biology, University of Alabama Birmingham, Birmingham, United States; David Ohnstad: Biology, College of St. Scholastica, Duluth, United States; Kristina Ok: Biology, University of Louisiana at Monroe, Monroe, United States; Oreoluwa M Olaniyan: Department of Biology, Baylor University, Waco, United States; Margot Oliver: Biology, Calvin College, Grand Rapids, United States; Olivia Oliver: University of Florida, Gainsville, United States; Helena Olivieri: Biology, Loyola Marymount University, Los Angeles, United States; Alishe Olmo-Colon: Biology, University of Puerto Rico - Cayey, Cayey, United States; Heather Olney: Biological Sciences, University of North Texas, Denton, United States; Morenike Olu: Biology, Howard College, Washington, DC, United States; Uche Onoh: Biological Sciences, University of North Texas, Denton, United States; Judy Oranika: Biology, Howard College, Washington, DC, United States; Kathryn Orban: Biology, Loyola Marymount University, Los Angeles, United States; Kristina A Orbe: Biology, Saint Joseph's University, Philadelphia, United States; Javier Ordonez: Biology, University of Texas at El Paso, El Paso, United States; Alexander S Orenstein: Biological Sciences, Carnegie Mellon University, Pittsburgh, United States; Iris Orion: Biology, Gonzaga University, Spokane, United States; Chelsea M Orlando: Department of Biology, Baylor University, Waco, United States; Ryan Orloski: Biology, Illinois Wesleyan University, Bloomington, United States; Beu Oropeza: Biology, University of Texas at El Paso, El Paso, United States; Andrew M Orr: Department of Biology, Baylor University, Waco, United States; Kiara Ortiz-Camacho: Biology,

University of Puerto Rico - Cayey, Cayey, United States; Eric Ortiz-Lopez: Biology, University of Puerto Rico - Cayey, Cayey, United States; Luis Ortiz-Munoz: Biology, University of Puerto Rico - Cayey, Cayey, United States; Erin Osborne: Ohio State University, Columbus, United States; Gina Osburn: Biology, Culver-Stockton College, Canton, United States; Brandon P Ossont: Biology, Saint Joseph's University, Philadelphia, United States; Elijah Ostalza: School of Science and Technology, Georgia Gwinnett College, Lawrenceville, United States; Larissa Osterbaan: Biology, Calvin College, Grand Rapids, United States; Taylor Oswald: Biology, Gonzaga University, Spokane, United States; Hidalisse Otero-Ruiz: Biology, University of Puerto Rico - Cayey, Cayey, United States; Nathaly Otero-Ruiz: Biology, University of Puerto Rico - Cayey, Cayey, United States; Cassandra Ott: Biological Sciences, University of Pittsburgh, Pittsburgh, United States; Nakeya Owens: Western Kentucky University, Bowling Green, United States; Sarah Owens: Biology, Gonzaga University, Spokane, United States; Oreoluwa Oyetan: Biology, Howard College, Washington, DC, United States; Oluwaseun Oyewole: Biology Department, Georgia State University, Milledgeville, United States; Jose Pabon-Lopez: Biology, University of Puerto Rico - Cayey, Cayey, United States; Joanna Padolina: Virginia Commonwealth University, Richmond, United States; Shallee Page: Environmental and Biological Science, University of Maine, Machias, Machias, United States; Josue Paico: Biology, University of Texas at El Paso, El Paso, United States; Jillian Pailin: Biology, Howard College, Washington, DC, United States; Ramya Palaniappan: University of Colorado at Boulder, Boulder, United States; Jamie L Palmer: Biology, Saint Joseph's University, Philadelphia, United States; Caressie Palomino: University of Colorado at Boulder, Boulder, United States; Haoyu Pan: Biological Sciences, University of Pittsburgh, Pittsburgh, United States; Marianne Pan: Biological Sciences, Carnegie Mellon University, Pittsburgh, United States; Lianette M Pappaterra: Biology, Saint Joseph's University, Philadelphia, United States; Mili Parikh: Biology, University of California San Diego, La Jolla, United States; Min Jin Park: Biological Sciences and Geology, Queensboro Community College, Bayside, United States; Minyoung Park: Biological Sciences, University of Pittsburgh, Pittsburgh, United States; Peter J Park: Biology and Chemistry, Nyack College, Nyack, United States; Jacob Parker: Biology, Gonzaga University, Spokane, United States; Lindsay Parnell: Biology, Spelman College, Atlanta, United States; Laura E Parrella: Biological Sciences, Carnegie Mellon University, Pittsburgh, United States; Prasanna Parthasarathy: Western Kentucky University, Bowling Green, United States; Natasha Pascal: Microbiology, Miami University, Oxford, United States; Valerie Paschalis: Montclair State University, Montclair, United States; Jacob Pascual: Biology, Loyola Marymount University, Los Angeles, United States; Anjali P Patel: Department of Biology, Baylor University, Waco, United States; Mira Patel: Biology, University of Alabama Birmingham, Birmingham, United States; Krupa Patel: Biology, Gettysburg College, Gettysburg, United States; Nikita Patel: Montclair State University, Montclair, United States; Shivani Patel: Montclair State University, Montclair, United States; Pooja M Patel: Biology, College of Charleston, Charleston, United States; Pooja B Patel: Biology, Saint Joseph's University, Philadelphia, United States; Pooja Patel: Biology, Loyola Marymount University, Los Angeles, United States; Rikita Patel: Biology, University of Alabama Birmingham, Birmingham, United States; Ritesh Patel: Biology, University of Alabama Birmingham, Birmingham, United States; Ruchik Patel: Biology, Washington University in St. Louis, St. Louis, United States; Shivum Patel: Biology, Illinois Wesleyan University, Bloomington, United States; Sarah Patno: Biology, Loyola Marymount University, Los Angeles, United States; Briawn Patrick: Biology, North Carolina Central University, Durham, United States; Emma Patschorke: Biology, Carthage College, Kenosha, United States; Shejuti Paul: Biology, University of Alabama Birmingham, Birmingham, United States; David E Payne II: Microbiology and Molecular Biology, Brigham Young University, Provo, United States; Yogitha Pazhani: University of Colorado at Boulder, Boulder, United States; Jacelyn Peabody: Biology, Carthage College, Kenosha, United States; Bria Peacock: Biology, Howard College, Washington, DC, United States; Katelyn C Peasley: Department of Biology, Baylor University, Waco, United States; Eric N Pederson: Biological Sciences, Carnegie Mellon University, Pittsburgh, United States; Marisa L Pedulla: Biology, Montana Tech of the University of Montana, Butte, United States; Luke Peebles: Biology, Montana Tech of the University of Montana, Butte, United States; Amber Peek-Simpson: Biology, Jacksonville State University, Jacksonville, United States; Alexandra Peister: Morehouse College, Atlanta, United States; Fei Peng: Biology, Smith College, Northampton, United States; Katelyn Pennington: Biology, Gonzaga University, Spokane, United States; Laura Penrod: Biological Sciences, University of North Texas, Denton, United States; Kathleen Peoples: University of Colorado at Boulder, Boulder, United

States; Gretchen Peppers: Xavier University of Louisiana, New Orleans, United States; Luigi Peracchi: University of California Santa Cruz, Santa Cruz, United States; Natalia Perecki: Biology and Medicine, Brown University, Providence, United States; Ariane N Pereira: Biology, College of Charleston, Charleston, United States; Paola Pereira-Rivera: Biology, University of Puerto Rico - Cayey, Cayey, United States; Varahenage R Perera: Biology, University of California San Diego, La Jolla, United States; Dahiana Perez: School of Science and Technology, Georgia Gwinnett College, Lawrenceville, United States; Ricardo Perez: University of Florida, Gainsville, United States; Eliezer Perez-Colomba: Biology, University of Puerto Rico - Cayey, Cayey, United States; Gabriel Perez-Lopez: Biology, University of Puerto Rico - Cayey, Cayey, United States; Joseph Perez-Otero: Biology, University of Puerto Rico - Cayey, Cayey, United States; Abbey Perl: Biological Sciences, Lehigh University, Bethlehem, United States; Christina Perri: Providence College, Providence, United States; Benjamin M Perrin: Biology, College of Charleston, Charleston, United States; Christine Elise Peters: Biology, University of California San Diego, La Jolla, United States; Caroline Petersen: Biological Sciences, Lehigh University, Bethlehem, United States; Christian Peterson: Biology, Gonzaga University, Spokane, United States; Kraig Peterson: Biology, College of St. Scholastica, Duluth, United States; Thalia Peterson Galvez: The Evergreen State College, Olympia, United States; Mitchell Petredis: Biology, Loyola Marymount University, Los Angeles, United States; David Peyton: Western Kentucky University, Bowling Green, United States; Brandon Pezley: Biology, Culver-Stockton College, Canton, United States; Patrick Pfaffle: Biology, Carthage College, Kenosha, United States; Shawna Pfeif: Biology, Gonzaga University, Spokane, United States; Rachel E Pferdehirt: Biological Sciences, Carnegie Mellon University, Pittsburgh, United States; Elizabeth Pflugradt: University of Maine, Honors College, Orono, United States; Anthony Pham: Xavier University of Louisiana, New Orleans, United States; Danny Pham: University of Maine, Honors College, Orono, United States; Michael Pham: Biology, University of California San Diego, La Jolla, United States; Mai-Trinh Pham: Biology, College of Charleston, Charleston, United States; Alex Phillips: University of California Santa Cruz, Santa Cruz, United States; Kaysi Phillips: Western Kentucky University, Bowling Green, United States; Melissa Phillips: Science, Cabrini College, Radnor, United States; Olivia Phillips: Microbiology and Biotechnology, North Carolina State University, Raleigh, United States; Connie Phung: Purdue University, West Lafayette, United States; Carlie Pickrel: Biology, Nebraska Wesleyan University, Lincoln, Nebraska, United States; Anya Pierson: Biology, Gettysburg College, Gettysburg, United States; Britta Pihl: Biology, Gonzaga University, Spokane, United States; Nathan Pihl: Biology, Loyola Marymount University, Los Angeles, United States; Jessamine Pilcher: Purdue University, West Lafayette, United States; Indiren Pillay: Biology Department, Georgia State University, Milledgeville, United States; Robert Pinches: Providence College, Providence, United States; Megan Pineda: Biological Sciences, University of North Texas, Denton, United States; Jonathan Pinney: Biology, University of Texas at El Paso, El Paso, United States; Matthew Piotrowiak: Biology, Illinois Wesleyan University, Bloomington, United States; Yasaman Camelia Pirahanchi: Biology, University of California San Diego, La Jolla, United States; Brianna Piro: Biology, Illinois Wesleyan University, Bloomington, United States; Kaknika Pisith: The Evergreen State College, Olympia, United States; Andrew Pita: Biology, Loyola Marymount University, Los Angeles, United States; Lauren Pittak: Microbiology, Miami University, Oxford, United States; Thomas J Pittman: Biology, College of Charleston, Charleston, United States; Amanda Pitts: Biology, Jacksonville State University, Jacksonville, United States; Sonja Pizzini: Biological Sciences, University of North Texas, Denton, United States; Marie Pizzorno: Department of Biology, Bucknell University, Lewisburg, United States; Anna Plantinga: Biology, Calvin College, Grand Rapids, United States; Andryus Planutis: Biology, University of California San Diego, La Jolla, United States; Melanie Plastini: University of California Santa Cruz, Santa Cruz, United States; Ruth Plymale: Biology, Ouachita Baptist University, Arkadelphia, United States; Adam Poff: Biology, Gettysburg College, Gettysburg, United States; Joe Pogliano: Biology, University of California San Diego, La Jolla, United States; Kit Pogliano: Biology, University of California San Diego, La Jolla, United States; Zeeshan Polani: Biology, North Carolina Central University, Durham, United States; Rachel Polet: Biology, Calvin College, Grand Rapids, United States; Galen Polise: Biology, Trinity College, Hartford, United States; Elliot Pollack: Biology, Trinity College, Hartford, United States; Tatianna Pollak: Biology, College of St. Scholastica, Duluth, United States; Aja Pollard: Biology, Howard College, Washington, DC, United States; Georgia Pollard: University of California Santa Cruz, Santa Cruz, United States; Mariela Ponce-Torres: Biology, University of Puerto Rico - Cayey, Cayey, United States; Sita Pongthunyaviriya: University of

Colorado at Boulder, Boulder, United States; Amanda Porter: Biology, Hope College, Holland, United States; Chritiane J Porter: Biology, College of Charleston, Charleston, United States; Lindsey Porter: Western Kentucky University, Bowling Green, United States; Michelle Portillo: Biology, University of Texas at El Paso, El Paso, United States; Abigail J Porzucek: Biology, Saint Joseph's University, Philadelphia, United States; Lindsay M Poss: Biological Sciences, Carnegie Mellon University, Pittsburgh, United States; Matthew Postolowski: Ohio State University, Columbus, United States; Joseph Potesta: Science, Cabrini College, Radnor, United States; Michael Potter: The Evergreen State College, Olympia, United States; Destinee Powell: Biology, North Carolina Central University, Durham, United States; Elijah Powell: Morehouse College, Atlanta, United States; Emily Powell: Biology, Jacksonville State University, Jacksonville, United States; Jordan Powell: Xavier University of Louisiana, New Orleans, United States; Taylor R Power: Honors Program, Florida Gulf Coast University, Fort Myers, United States; Marianne Poxleitner: Biology, Gonzaga University, Spokane, United States; Riemer Praamsma: Biology, Calvin College, Grand Rapids, United States; Ashley Prasad: Biology, CUNY, Queens College, Queens, United States; Smrithi Prem: Biological Sciences, Lehigh University, Bethlehem, United States; Anthony Preston: Biology, Merrimack College, North Andover, United States; Jordan Price: Morehouse College, Atlanta, United States; Kirsten Price: Biology, Gonzaga University, Spokane, United States; Kimberly Prince: Biological Sciences, University of North Texas, Denton, United States; Arturo Principe-Cartagena: Biology, University of Puerto Rico - Cayey, Cayey, United States; Amber Prins: Biology, Hope College, Holland, United States; Stephenie Prinsen: Biology, College of St. Scholastica, Duluth, United States; Sara Privatt: Washington State University, Pullman, United States; Daysi Proano: Biological Sciences and Geology, Queensboro Community College, Bayside, United States; Atticus M Proctor: Biology, Montana Tech of the University of Montana, Butte, United States; Lora Prosser: Washington State University, Pullman, United States; Ashley Prout: Biological Sciences, University of Pittsburgh, Pittsburgh, United States; Brooke Pruitt: Biology, University of Alabama Birmingham, Birmingham, United States; Alyssa Prush: Montclair State University, Montclair, United States; Seamus Pugh: Microbiology, Miami University, Oxford, United States; Sandra Pumper: Biology, University of Wisconsin-River Falls, River Falls, United States; Taylor Purks: Biology, Howard College, Washington, DC, United States; Kendyl B Pyfer: Biology, Saint Joseph's University, Philadelphia, United States; Zhijie Qi: Biology, Washington University in St. Louis, St. Louis, United States; Hong Qin: Biology, Spelman College, Atlanta, United States; Michael Qiu: Biology, University of California San Diego, La Jolla, United States; Conrad Quade: University of Colorado at Boulder, Boulder, United States; Ariana Quattlebaum: Biology, College of Charleston, Charleston, United States; Ashley Queen: Microbiology, Howard College, Washington, DC, United States; Thaina Quiles: Biology, North Carolina Central University, Durham, United States; Alexandra Quinn: Biology, College of St. Scholastica, Duluth, United States; Christopher Quintanal-Segarra: Biology, University of Puerto Rico - Cayey, Cayey, United States; Hector Quintero-Alvarez: Biology, University of Puerto Rico - Cayey, Cayey, United States; Hannah S Rabinowitz: Biology, Washington University in St. Louis, St. Louis, United States; Hiram Rabri-Diaz: Biology, University of Puerto Rico - Cayey, Cayey, United States; Aleksandar Radakovic: Biology, College of St. Scholastica, Duluth, United States; Brendan Radel: Microbiology, Miami University, Oxford, United States; Casey Radice: Biological Sciences, University of North Texas, Denton, United States; Nicholas J Radigan: Biology, Saint Joseph's University, Philadelphia, United States; Jennifer Radke: Biology, College of Charleston, Charleston, United States; Paige Radtke: University of Colorado at Boulder, Boulder, United States; Miranda Raevsky: Biology, Culver-Stockton College, Canton, United States; Haley Rafferty: Biology, Trinity College, Hartford, United States; Prartana Ramachandran: Biology, Illinois Wesleyan University, Bloomington, United States; Eloy Ramirez: Biology, University of Texas at El Paso, El Paso, United States; Juan Ramirez: Washington State University, Pullman, United States; Valeria Ramirez: University of Florida, Gainsville, United States; Jose Ramirez-Diaz: Biology, University of Puerto Rico - Cayey, Cayey, United States; Mauro Ramos: Biological Sciences and Geology, Queensboro Community College, Bayside, United States; Kareliz Ramos-Aponte: Biology, University of Puerto Rico - Cayey, Cayey, United States; Laura Ramos-Flores: Biology, University of Puerto Rico - Cayey, Cayey, United States; Taylor Randell: Biology, Gettysburg College, Gettysburg, United States; Alex Randoph: The Evergreen State College, Olympia, United States; Alexa Raney: Natural Sciences, Del Mar College, Corpus Christi, United States; Surabhi Rao: Biology, University of Alabama Birmingham, Birmingham, United States; Dustin Raper: Biology, North Carolina Central University, Durham, United States;

Rehnuma Rashid: Montclair State University, Montclair, United States; Amir Rastegari: Biology, University of Texas at El Paso, El Paso, United States; Suchita Rastogi: Biology, Washington University in St. Louis, St. Louis, United States; Rebecca Ratusnik: Biology, Smith College, Northampton, United States; Luke Raudaskoski: University of California Santa Cruz, Santa Cruz, United States; Tiffany Ravelomanantsoa: Biology, Gettysburg College, Gettysburg, United States; Rawnok Rayeka: Biological Sciences and Geology, Queensboro Community College, Bayside, United States; John R Raymond: Biology, College of Charleston, Charleston, United States; Ulugbek Razakov: Biological Sciences and Geology, Queensboro Community College, Bayside, United States; David C Ream: Microbiology, Miami University, Oxford, United States; Krisanavane Reddi: Department of Microbiology, Immunology, and Molecular Genetics, University of California, Los Angeles, Los Angeles, United States; Sherry Reddix: Xavier University of Louisiana, New Orleans, United States; Kayla Reed: Biology, University of Wisconsin-River Falls, River Falls, United States; Abigayle Reed: Biology, University of Louisiana at Monroe, Monroe, United States; Jeremiah Reenders: Biology, Calvin College, Grand Rapids, United States; Lukas Rees: Biology, Washington University in St. Louis, St. Louis, United States; Brooke Reese: Biology, Gonzaga University, Spokane, United States; Jordan Rego: Providence College, Providence, United States; Amanda Reicherter: Biological Sciences, Lehigh University, Bethlehem, United States; Margaret Reilly: Ohio State University, Columbus, United States; Meagan Reinecke: Biology, Carthage College, Kenosha, United States; Emily Reinhart: Ohio State University, Columbus, United States; Mickey J Reiss: Biological Sciences, Carnegie Mellon University, Pittsburgh, United States; Leah Relko: Microbiology, Miami University, Oxford, United States; Daniel Renner: Virginia Commonwealth University, Richmond, United States; Piper Replogle: University of Colorado at Boulder, Boulder, United States; Tessa Retzlaff: Biology, Chadron State College, Chadron, United States; Daniel Reyes: Biology, University of Texas at El Paso, El Paso, United States; Angel Reyes-Aponte: Biology, University of Puerto Rico - Cayey, Cayey, United States; Adriana Reyes-DeLeon: Biology, University of Puerto Rico - Cayey, Cayey, United States; Nathan Reyna: Biology, Ouachita Baptist University, Arkadelphia, United States; Brady Reys: Microbiology, Miami University, Oxford, United States; Cortney Rhoades: The Evergreen State College, Olympia, United States; Christopher Rhodes: Biology, Loyola Marymount University, Los Angeles, United States; Christopher Rhodes: Biology, Loyola Marymount University, Los Angeles, United States; Emily Rhude: Biology, Calvin College, Grand Rapids, United States; Corwin N Rhyan: Biology, Washington University in St. Louis, St. Louis, United States; Ricci-tam Chiara Jeun-Ning Elena: Biology, University of California San Diego, La Jolla, United States; Jeremy Rice: Washington State University, Pullman, United States; Benjamin James Rich: Biology, University of California San Diego, La Jolla, United States; Erin Morris Richard: Biology, College of Charleston, Charleston, United States; Mitchell Richards: Biology, Ouachita Baptist University, Arkadelphia, United States; Megan Richters: Biology, University of Louisiana at Monroe, Monroe, United States; Jenna Rickus: Purdue University, West Lafayette, United States; Kendall Riddelle: Biology, Gonzaga University, Spokane, United States; Tanya Riddick: ISBT, LaSalle University, Philadelphia, United States; Benjamin C Riley: Department of Biology, Baylor University, Waco, United States; Claire A Rinehart: Western Kentucky University, Bowling Green, United States; Sarah Ring: University of Florida, Gainsville, United States; Alyssa Riordan: Biology, University of Louisiana at Monroe, Monroe, United States; Luis Rivera-Arce: Biology, University of Puerto Rico - Cayey, Cayey, United States; Yoliveliz Rivera-Bernier: Biology, University of Puerto Rico - Cayey, Cayey, United States; Myrielis Rivera-Burgos: Biology, University of Puerto Rico - Cayey, Cayey, United States; Denise Rivera-Cordero: Biology, University of Puerto Rico - Cayey, Cayey, United States; Asdrubal Rivera-Dones: Biology, University of Puerto Rico - Cayey, Cayey, United States; Shalimar Rivera-Gonzalez: Biology, University of Puerto Rico - Cayey, Cayey, United States; Marlene Rivera-Martinez: Biology, University of Puerto Rico - Cayey, Cayey, United States; Yamil Rivera-Ortiz: Biology, University of Puerto Rico - Cayey, Cayey, United States; Carlivette Rivera-Rivera: Biology, University of Puerto Rico - Cayey, Cayey, United States; Alejandra Rivera-Rodriguez: Biology, University of Puerto Rico - Cayey, Cayey, United States; Miriam Rivera-Rolon: Biology, University of Puerto Rico - Cayey, Cayey, United States; Angelica Rivera-Rosa: Biology, University of Puerto Rico - Cayey, Cayey, United States; Melanie Rivera-Rosario: Biology, University of Puerto Rico - Cayey, Cayey, United States; Rivera-San Antonio Jennifer: Biology, University of Puerto Rico - Cayey, Cayey, United States; Antonella Rivezzi: Montclair State University, Montclair, United States; Erin Rizzo: Biology, University of Louisiana at Monroe, Monroe, United States; Haleigh Roach: University of Maine, Honors College, Orono, United

States; Darcy Roberson: Biological Sciences, University of North Texas, Denton, United States; Megan Robertson: Biological Sciences, Lehigh University, Bethlehem, United States; Molly Robertson: Natural Sciences, Del Mar College, Corpus Christi, United States; Laura Robey: Biology, University of Wisconsin-River Falls, River Falls, United States; Courtney Robinson: Biology, Howard College, Washington, DC, United States; Danielle Robinson: Biology, University of Alabama Birmingham, Birmingham, United States; Jana Robinson: Biology, University of Louisiana at Monroe, Monroe, United States; Kayla Robinson: Biological Sciences, University of North Texas, Denton, United States; Tamara S Robinson: Division of Natural and Health Sciences, Seton Hill University, Greensburg, United States; Joseph Robinson: Biology, Gettysburg College, Gettysburg, United States; Sarah Rock: Biology, Loyola Marymount University, Los Angeles, United States; Kathy Rodogiannis: Biology, Trinity College, Hartford, United States; Christina Rodriguez: Biology, University of Texas at El Paso, El Paso, United States; Jasmin Rodriguez: Biological Sciences, University of North Texas, Denton, United States; Katherine M Rodriguez: Department of Biology, Baylor University, Waco, United States; Lauren Rodriguez: University of California Santa Cruz, Santa Cruz, United States; Neudalis Rodriguez-Burgos: Biology, University of Puerto Rico - Cayey, Cayey, United States; Frances Rodriguez-Diaz: Biology, University of Puerto Rico - Cayey, Cayey, United States; Ashley Rodriguez-Gonzalez: Biology, University of Puerto Rico - Cayey, Cayey, United States; Claribel Rodriguez-Lopez: Biology, University of Puerto Rico - Cayey, Cayey, United States; Stephen Rogers: Providence College, Providence, United States; Thomas Rogers: Biology, College of St. Scholastica, Duluth, United States; Kelvin J Rojas: Biological Sciences, Carnegie Mellon University, Pittsburgh, United States; Lucero Rojas Gallegos: Biology, University of Texas at El Paso, El Paso, United States; David Rokhinson: Purdue University, West Lafayette, United States; Lauren E Rollman: Biology, Saint Joseph's University, Philadelphia, United States; Alfredo Roman: Biology, University of Texas at El Paso, El Paso, United States; Will Romero: Biology, University of Louisiana at Monroe, Monroe, United States; Kimberly A Rosales: Honors Program, Florida Gulf Coast University, Fort Myers, United States; German Rosas-Acosta: Biology, University of Texas at El Paso, El Paso, United States; Elizabeth L Rosenthal: Department of Biology, Baylor University, Waco, United States; Meredith K Rosenthal: Department of Biology, Baylor University, Waco, United States; Joseph Ross: Xavier University of Louisiana, New Orleans, United States; Jessica Rossi: University of Florida, Gainsville, United States; Lindsay Roth: Biology, Smith College, Northampton, United States; Kasey Roush: Ohio State University, Columbus, United States; Aislinn Rowan: Biological Sciences, Lehigh University, Bethlehem, United States; Ansley T Royal: School of Science and Technology, Georgia Gwinnett College, Lawrenceville, United States; Michael R Rubin: Biology, University of Puerto Rico - Cayey, Cayey, United States; Jeffrey Rubin: University of California Santa Cruz, Santa Cruz, United States; Jessica Ruby: University of California Santa Cruz, Santa Cruz, United States; Leanna Rucker: Xavier University of Louisiana, New Orleans, United States; Maxwell Sung Ruckstuhl: Biology, University of California San Diego, La Jolla, United States; Xiang Rui: Ohio State University, Columbus, United States; Michael Ruiz: Montclair State University, Montclair, United States; Sarafina Rush: Biology, Loyola Marymount University, Los Angeles, United States; Jeanette Russell: Washington State University, Pullman, United States; Alfredo Ruvalcaba: University of California Santa Cruz, Santa Cruz, United States; Joseph Ryan: Biology, Gonzaga University, Spokane, United States; Nathan Ryan: Biology, Gonzaga University, Spokane, United States; Ryan Amelia-Frances M: ISBT, LaSalle University, Philadelphia, United States; Jessica L Sabo: University of Florida, Gainsville, United States; Daniel Sabzghabaei: Biological Sciences, University of North Texas, Denton, United States; Lindsey Sacco: Biology, Gonzaga University, Spokane, United States; Quang V Sack: Biological Sciences, Carnegie Mellon University, Pittsburgh, United States; Rachna Sadana: Natural Sciences, University of Houston-Downtown, Houston, United States; Margaret S Saha: Biology, College of William and Mary, Williamsburg, United States; Katrina Sahawneh: Biology, University of Alabama Birmingham, Birmingham, United States; Kenny Salamea: Montclair State University, Montclair, United States; Samaher M Saleh: ISBT, LaSalle University, Philadelphia, United States; Mary Salim: Montclair State University, Montclair, United States; Angelica Salinas: Biology, University of Louisiana at Monroe, Monroe, United States; Brianna Salverda: Biology, Jacksonville State University, Jacksonville, United States; Andria Sammon: Biology, College of St. Scholastica, Duluth, United States; Jorrel Sampana: Biology, Loyola Marymount University, Los Angeles, United States; Shaterrica Sampson: Biology, University of Louisiana at Monroe, Monroe, United States; Jessica Sanchez: Biology, University of Texas at El Paso, El Paso, United States; Marieli Sanchez-Collazo:

Biology, University of Puerto Rico - Cayey, Cayey, United States; Erin Sanders: Department of Microbiology, Immunology, and Molecular Genetics, University of California, Los Angeles, Los Angeles, United States; James Sandoz: Department of Biological Sciences, University of Maryland, Baltimore County, Baltimore, United States; Eric Sanford: The Evergreen State College, Olympia, United States; Michael Santana: Biological Sciences, Lehigh University, Bethlehem, United States; Alexander Santiago: Biology, Loyola Marymount University, Los Angeles, United States; Jan Clement Santiago: Biology, Loyola Marymount University, Los Angeles, United States; David Santiago-Ochoa: Biology, University of Puerto Rico - Cayey, Cayey, United States; Jose Santiago-Vazquez: Biology, University of Puerto Rico - Cayey, Cayey, United States; Natanael Santillana: Biology, University of Texas at El Paso, El Paso, United States; Eddy Santos: Biology and Chemistry, Nyack College, Nyack, United States; Christy Saquin: Biological Sciences, University of North Texas, Denton, United States; Heba Sarhan: Xavier University of Louisiana, New Orleans, United States; Richard Sater: Microbiology and Biotechnology, North Carolina State University, Raleigh, United States; Lori Saunders: Biology, Smith College, Northampton, United States; Erin Sauve: Biology, Loyola Marymount University, Los Angeles, United States; Judith G Savitskaya: Biological Sciences, Carnegie Mellon University, Pittsburgh, United States; Lauren E Sawyer: Department of Biology, Baylor University, Waco, United States; Aatif Sayeed: Biology, Washington University in St. Louis, St. Louis, United States; Heidi Sayre: Western Kentucky University, Bowling Green, United States; Tyler Scaff: Western Kentucky University, Bowling Green, United States; Allysan Scatterday: Biology, College of Charleston, Charleston, United States; Claire Schaar: Biology, Hope College, Holland, United States; Amy Schade: Biological Sciences, University of North Texas, Denton, United States; Claire Schafer: Biological Sciences, University of Pittsburgh, Pittsburgh, United States; Lauren B Schellenberger: Biology, Culver-Stockton College, Canton, United States; Anne Scherer: Biology, College of St. Scholastica, Duluth, United States; Alexander Schierbeek: Biology, Calvin College, Grand Rapids, United States; Isaac B Schiller: Biology, Saint Joseph's University, Philadelphia, United States; Catherine Schilling: Biology, University of Louisiana at Monroe, Monroe, United States; Jessica Schipper: University of Colorado at Boulder, Boulder, United States; Jennifer Schlegel: Biological Sciences, Lehigh University, Bethlehem, United States; Elizabeth Schleh: Biology, Calvin College, Grand Rapids, United States; Joshua Schmidt: Biology, College of Charleston, Charleston, United States; Theresa Schmidt: Ohio State University, Columbus, United States; Carson Schneider: Biology, Gonzaga University, Spokane, United States; Seth Schneider: Washington State University, Pullman, United States; Sydney Schneider: Washington State University, Pullman, United States; Christine E Schnitzler: Integrative Biology, Oregon State University, Corvallis, United States; Morgan B Schoer: Biology, Washington University in St. Louis, St. Louis, United States; Leo R Scholl: Biological Sciences, Carnegie Mellon University, Pittsburgh, United States; Amanda Schorr: Biology, University of Louisiana at Monroe, Monroe, United States; Sarah Schrader: Western Kentucky University, Bowling Green, United States; Stephanie Schramm: Purdue University, West Lafayette, United States; Ariel L Schroeder: Division of Natural and Health Sciences, Seton Hill University, Greensburg, United States; Katherine Schroeder: Ohio State University, Columbus, United States; Allison Schroeder: Ohio State University, Columbus, United States; Monica Schroll: Biology, Gonzaga University, Spokane, United States; Karyssa Schrouder: Biology, Calvin College, Grand Rapids, United States; Jacob Schrull: University of Florida, Gainsville, United States; Kaitlyn Schuberth: Washington State University, Pullman, United States; Olivia Schuele: Biology, Gonzaga University, Spokane, United States; Thomas Schulte: Ohio State University, Columbus, United States; Ellen Schultz: Biology, University of Wisconsin-River Falls, River Falls, United States; Lisa Schultz: Biology, Calvin College, Grand Rapids, United States; Michael B Schultz: Biology, Washington University in St. Louis, St. Louis, United States; Morgan Schultz: Biology, Nebraska Wesleyan University, Lincoln, Nebraska, United States; Megan Schulz: Biology, Gonzaga University, Spokane, United States; AJ Schumacher: Washington State University, Pullman, United States; Victoria Schwartz: Providence College, Providence, United States; Amanda Schwarz: Biological Sciences, University of Pittsburgh, Pittsburgh, United States; Shelby Scola: Providence College, Providence, United States; Amanda Scott: Biology, University of Louisiana at Monroe, Monroe, United States; Taylor D Scott: Department of Biology, Baylor University, Waco, United States; Vincent Scuttaro: Biology and Chemistry, Nyack College, Nyack, United States; Corey D Seacrist: Biology, College of Charleston, Charleston, United States; Sabe Sears: Biology, Culver-Stockton College, Canton, United States; Amanda Seaton: Western Kentucky University, Bowling Green, United States; Hailey N Seaver:

Microbiology and Molecular Biology, Brigham Young University, Provo, United States; Ethan Sebasco: Montclair State University, Montclair, United States; M Esa Seegulam: Biology, Culver-Stockton College, Canton, United States; J Bradley Segal: Biology, University of California San Diego, La Jolla, United States; Gabriel C Segarra: Biology, College of Charleston, Charleston, United States; Ana Segura Lerma: Biology, University of Texas at El Paso, El Paso, United States; Reuben Seidl: Biology, Gonzaga University, Spokane, United States; Robert Semler: Science, Cabrini College, Radnor, United States; Sageanne Senneff: University of California Santa Cruz, Santa Cruz, United States; Jiwon Seo: Biological Sciences and Geology, Queensboro Community College, Bayside, United States; Bijan Sepheri: Biological Sciences, Carnegie Mellon University, Pittsburgh, United States; Erica Sewell: University of Maine, Honors College, Orono, United States; Amy Shafer: Biology, Gonzaga University, Spokane, United States; Rachel A Shaffer: Biological Sciences, Carnegie Mellon University, Pittsburgh, United States; Christopher D Shaffer: Biology, Washington University in St. Louis, St. Louis, United States; Madeena Shafiq: Ohio State University, Columbus, United States; Harsh Shah: Biology, University of Alabama Birmingham, Birmingham, United States; Zalak Shah: Virginia Commonwealth University, Richmond, United States; Sohum C Shah: Department of Biology, Baylor University, Waco, United States; Lindsey Shain: Western Kentucky University, Bowling Green, United States; Peter Shank: Biology and Medicine, Brown University, Providence, United States; Devon Shannonhouse-Wilde: University of California Santa Cruz, Santa Cruz, United States; Ananya Sharma: Western Kentucky University, Bowling Green, United States; Sanskriti Sharma: University of Florida, Gainsville, United States; Shaylen Sharp: Washington State University, Pullman, United States; Spencer Sharp: Biology, Nebraska Wesleyan University, Lincoln, Nebraska, United States; Nadia Sheen: Biological Sciences, Carnegie Mellon University, Pittsburgh, United States; Marshall G Sheide: Microbiology and Molecular Biology, Brigham Young University, Provo, United States; Kathryn E Sheldon: Biological Sciences, Carnegie Mellon University, Pittsburgh, United States; Michael A Shelfo: Microbiology and Molecular Biology, Brigham Young University, Provo, United States; Laura Shellooe: Biology, Gonzaga University, Spokane, United States; Matthew Sheltra: University of Maine, Honors College, Orono, United States; Adrie Shen: Biological Sciences, University of North Texas, Denton, United States; Jean Shen: Department of Microbiology, Immunology, and Molecular Genetics, University of California, Los Angeles, Los Angeles, United States; Morgan Sherer: Ohio State University, Columbus, United States; Andrea Sherod: Biology, Culver-Stockton College, Canton, United States; Chringma Sherpa: Biology, Spelman College, Atlanta, United States; Eileen Shi: Biology, University of California San Diego, La Jolla, United States; Rani Shiao: Biology, University of California San Diego, La Jolla, United States; Kelly S Shibuya: Biological Sciences, Carnegie Mellon University, Pittsburgh, United States; Hyo Jung Shin: Biological Sciences and Geology, Queensboro Community College, Bayside, United States; Marla Shipton: Ohio State University, Columbus, United States; Katharine Shively: Ohio State University, Columbus, United States; Breanne Short: Washington State University, Pullman, United States; Rahul Shrikanth: Biology, Illinois Wesleyan University, Bloomington, United States; Michael Shultz: The Evergreen State College, Olympia, United States; Shruthi Shyamala: Biology, Howard College, Washington, DC, United States; Zia Siddiqui: Biological Sciences, University of North Texas, Denton, United States; Hannah Sides: University of Florida, Gainsville, United States; Aziz Sidra: Natural Sciences, University of Houston-Downtown, Houston, United States; Talles Sidronio: University of Florida, Gainsville, United States; Christina Godfried Sie: Biological Sciences, Lehigh University, Bethlehem, United States; Satchel Siegel: Biology, Washington University in St. Louis, St. Louis, United States; Mary Siki: Biological Sciences, University of Pittsburgh, Pittsburgh, United States; Jeremy Silva: Biology, University of Texas at El Paso, El Paso, United States; Abigail Silva: Montclair State University, Montclair, United States; Ethan Sim: Department of Microbiology, Immunology, and Molecular Genetics, University of California, Los Angeles, Los Angeles, United States; Jacqueline Simeon: Biology and Chemistry, Nyack College, Nyack, United States; Nicholas J Simitzi: Department of Biology, Baylor University, Waco, United States; Abigail R Simmons: Biological Sciences, Carnegie Mellon University, Pittsburgh, United States; Jacob Simon: Biology, University of Louisiana at Monroe, Monroe, United States; Stephanie E Simon: Biological Sciences, University of North Texas, Denton, United States; Zachary Simon: University of California Santa Cruz, Santa Cruz, United States; Anita Simonian: Biology, Loyola Marymount University, Los Angeles, United States; Nathan Simpson: Biology, University of Louisiana at Monroe, Monroe, United States; Erika F Sims: Biology, Washington University in St. Louis, St. Louis, United States; Danielle Sin: Biological Sciences, Lehigh

University, Bethlehem, United States; Ramya Singireddy: Biology, University of Alabama Birmingham, Birmingham, United States; Samantha Siomko: Biology, Gettysburg College, Gettysburg, United States; Benjamin Siranosian: Biology and Medicine, Brown University, Providence, United States; Emily Sirek: Biology, University of Wisconsin-River Falls, River Falls, United States; Jordan Skinner: Ohio State University, Columbus, United States; Brittany Sklenar: School of Science and Technology, Georgia Gwinnett College, Lawrenceville, United States; Tyler Slade: biology, North Carolina Central University, Durham, United States; Lucas Slivicke: Biology, University of Wisconsin-River Falls, River Falls, United States; Katherine Smart: Biological Sciences, University of North Texas, Denton, United States; Megan Smeets: Biology, Illinois Wesleyan University, Bloomington, United States; Thomas J Smith: Biology, Saint Joseph's University, Philadelphia, United States; Abigail J Smith: Biology, College of Charleston, Charleston, United States; Damien Smith: Environmental and Biological Science, University of Maine, Machias, Machias, United States; Erica Smith: Biology, College of William and Mary, Williamsburg, United States; Elliott G Smith: Biology, College of Charleston, Charleston, United States; Joanna Smith: University of California Santa Cruz, Santa Cruz, United States; Jason P Smith: Biology, College of Charleston, Charleston, United States; Katherine A Smith: Biology, Saint Joseph's University, Philadelphia, United States; Kyle M Smith: Biology, Saint Joseph's University, Philadelphia, United States; Logan Smith: Biology, Jacksonville State University, Jacksonville, United States; Luke D Smith: Department of Biology, Baylor University, Waco, United States; Savanna K Smith: ISBT, LaSalle University, Philadelphia, United States; Colby Smith: Biology, Ouachita Baptist University, Arkadelphia, United States; Dennis Smith: The Evergreen State College, Olympia, United States; Jalen Smith: Morehouse College, Atlanta, United States; Kyle C Smith: Microbiology and Molecular Biology, Brigham Young University, Provo, United States; Veronica Smith: Science, Cabrini College, Radnor, United States; Brett Snyder: Biology, College of Charleston, Charleston, United States; Sarah L Sokol: Division of Natural and Health Sciences, Seton Hill University, Greensburg, United States; Divyakshi Solanki: University of Florida, Gainsville, United States; Vincent Sonderby: Biology, Gonzaga University, Spokane, United States; Robert Soohey: University of Maine, Honors College, Orono, United States; Stephen Soohey: University of Maine, Honors College, Orono, United States; Talia K Sopp: Biological Sciences, Carnegie Mellon University, Pittsburgh, United States; Samantha Sorenson: Biology, Illinois Wesleyan University, Bloomington, United States; Erin Sorge: Biology, University of Wisconsin-River Falls, River Falls, United States; Jesus Sotelo: Department of Biology, Baylor University, Waco, United States; Subada Soti: University of Colorado at Boulder, Boulder, United States; Rebecca Soto: Biology, University of Texas at El Paso, El Paso, United States; Kara Soucek: Biology, Gonzaga University, Spokane, United States; Hannah Souers: Biology, Gonzaga University, Spokane, United States; Maura J Southwell: Biology, Saint Joseph's University, Philadelphia, United States; Hannah Space: Biology, University of Wisconsin-River Falls, River Falls, United States; Alexia L Sparrow: Biology, Saint Joseph's University, Philadelphia, United States; Blaire Spaulding: Biology, Spelman College, Atlanta, United States; Kayla Spears: Biology, Carthage College, Kenosha, United States; Michael Clayton Speed: Natural Sciences, Del Mar College, Corpus Christi, United States; Shannon B Spencer: Biology, Saint Joseph's University, Philadelphia, United States; Lauren Spicer: Ohio State University, Columbus, United States; Preethy S Sridharan: Biological Sciences, Carnegie Mellon University, Pittsburgh, United States; Meghan K St. Cyr: Department of Biology, Baylor University, Waco, United States; Erika Stairs: Western Kentucky University, Bowling Green, United States; Katelyn J Stanley: Division of Natural and Health Sciences, Seton Hill University, Greensburg, United States; Julie Stanton: Washington State University, Pullman, United States; John Starner: Montclair State University, Montclair, United States; John Starnes: Western Kentucky University, Bowling Green, United States; Beth Statler: Ohio State University, Columbus, United States; Richard Stauffer: Biology, Carthage College, Kenosha, United States; Hernando Steidel: Pedagogy, University of Puerto Rico - Cayey, Cayey, United States; Jackson Steinberg: Department of Microbiology, Immunology, and Molecular Genetics, University of California, Los Angeles, Los Angeles, United States; Ivy J Stejskal: Department of Biology, Baylor University, Waco, United States; Oleg Stens: Biology, University of California San Diego, La Jolla, United States; Rachel Sternberg: Biological Sciences, Lehigh University, Bethlehem, United States; Leah Stetzel: Microbiology, Miami University, Oxford, United States; McKayla Stevens: Biology, College of Idaho, Caldwell, United States; Joseph Charles Steward: Biology, University of California San Diego, La Jolla, United States; Damion Stewart: School of Science and Technology, Georgia Gwinnett College, Lawrenceville, United States; Eric Stewart:

Xavier University of Louisiana, New Orleans, United States; Shawntavia Stewart: Biology, Calvin College, Grand Rapids, United States; Julianne J Sticha: Division of Natural and Health Sciences, Seton Hill University, Greensburg, United States; Jennifer Stiles (Beane): Microbiology and Biotechnology, North Carolina State University, Raleigh, United States; Julia Stimpfl: Ohio State University, Columbus, United States; Jonathan Stites: University of California Santa Cruz, Santa Cruz, United States; Timothy Stoddard: Biology, Gonzaga University, Spokane, United States; Kaitlyn Stoddart: Biological Sciences, University of North Texas, Denton, United States; Molly Storer: Purdue University, West Lafayette, United States; Elizabeth K Storm: Biology, Saint Joseph's University, Philadelphia, United States; Emily Stowe: Department of Biology, Bucknell University, Lewisburg, United States; Andrew Straszewski: Biology, Carthage College, Kenosha, United States; Clark Straub: University of California Santa Cruz, Santa Cruz, United States; Zachary Streeter: Biology, University of Louisiana at Monroe, Monroe, United States; William Strober: Biology, Washington University in St. Louis, St. Louis, United States; Alyssa Stubblefield: Biology, Ouachita Baptist University, Arkadelphia, United States; Joseph Stukey: Biology, Hope College, Holland, United States; Rachel Sturge: Department of Biological Sciences, University of Maryland, Baltimore County, Baltimore, United States; Trevor Sughrue: University of California Santa Cruz, Santa Cruz, United States; Niles Sulkko: University of Colorado at Boulder, Boulder, United States; Matthew Sullivan: University of Maine, Honors College, Orono, United States; Ryan Sullivan: Biological Sciences, University of North Texas, Denton, United States; David Sullivan: Microbiology, Miami University, Oxford, United States; Cassandra Sulski: Biology, Nebraska Wesleyan University, Lincoln, Nebraska, United States; Grace Sundeen: Biology, College of St. Scholastica, Duluth, United States; Nandini Surendranathan: Montclair State University, Montclair, United States; Minnu H Suresh: Division of Natural and Health Sciences, Seton Hill University, Greensburg, United States; Jacob A Surges: Department of Biology, Baylor University, Waco, United States; Theresia Sutherlin: Xavier University of Louisiana, New Orleans, United States; Sarah Swalley: Biology, University of Alabama Birmingham, Birmingham, United States; David Swartout: University of Florida, Gainsville, United States; Elliot Swartz: Biological Sciences, Lehigh University, Bethlehem, United States; Ramata Sy: Montclair State University, Montclair, United States; Najah Syed: Biological Sciences, University of North Texas, Denton, United States; Cole Sylvester: Biology, Trinity College, Hartford, United States; Charity Sylvester: Xavier University of Louisiana, New Orleans, United States; Jacqueline Synder: Biological Sciences, Lehigh University, Bethlehem, United States; Mary R Szurgot: Biology, Saint Joseph's University, Philadelphia, United States; Marta Szyszka: Biological Sciences and Geology, Queensboro Community College, Bayside, United States; Shandee Tachick: Biology, College of Idaho, Caldwell, United States; Kayla Taggard: Biology, Gonzaga University, Spokane, United States; Michael Taguiam: Department of Microbiology, Immunology, and Molecular Genetics, University of California, Los Angeles, Los Angeles, United States; Kareen J Taha: Biology, College of Charleston, Charleston, United States; Ahmed Tahseen: University of Colorado at Boulder, Boulder, United States; Katherine Tai: Department of Microbiology, Immunology, and Molecular Genetics, University of California, Los Angeles, Los Angeles, United States; George Taishin: Montclair State University, Montclair, United States; Shiori Takashima: Biology, University of Texas at El Paso, El Paso, United States; Jordan Takasugi: Biology, Gonzaga University, Spokane, United States; Jaee Tamhane: Biology, Loyola Marymount University, Los Angeles, United States; Kathleen Tan: Department of Microbiology, Immunology, and Molecular Genetics, University of California, Los Angeles, Los Angeles, United States; Tin-Yun Tang: Biology, University of California San Diego, La Jolla, United States; Natalie Tanke: Biology, Gettysburg College, Gettysburg, United States; Lucas Tans: Biology, Hope College, Holland, United States; Brian Tarbox: Marine Science, Southern Maine Community College, South Portland, United States; Lubaba Tasnim: Biological Sciences, University of North Texas, Denton, United States; Devon Taylor: Biological Sciences, Lehigh University, Bethlehem, United States; Rebecca Taylor: Biology, Smith College, Northampton, United States; Sarah Taylor: Biology and Medicine, Brown University, Providence, United States; Barbara J Taylor: Integrative Biology, Oregon State University, Corvallis, United States; Warren Taylor: Biology, University of Wisconsin-River Falls, River Falls, United States; Kristina Taynor: Microbiology, Miami University, Oxford, United States; Laura Teal: Biology, Hope College, Holland, United States; Gavin L Teichman: ISBT, LaSalle University, Philadelphia, United States; Shreya Tekumalla: University of Colorado at Boulder, Boulder, United States; David W Temme: Biology, Saint Joseph's University, Philadelphia, United States; Louise Temple: Integrated Science & Technology, James Madison University,

Harrisonburg, United States; Autumn Tendler: Washington State University, Pullman, United States; Mayra Terres: Biology, University of Texas at El Paso, El Paso, United States; Treyc Terry: Xavier University of Louisiana, New Orleans, United States; Michael Teti: Biology and Chemistry, Nyack College, Nyack, United States; Patricia M Thang: Biological Sciences, Carnegie Mellon University, Pittsburgh, United States; Franklin L Thelmo: Biology, Saint Joseph's University, Philadelphia, United States; Anapaula Themann: Biology, University of Texas at El Paso, El Paso, United States; Monique Theriault: University of Maine, Honors College, Orono, United States; Sarah Thibault: Department of Biology, Bucknell University, Lewisburg, United States; Arsema A Thomas: Biological Sciences, Carnegie Mellon University, Pittsburgh, United States; Ditte C Thomas: Biology, College of Charleston, Charleston, United States; Sherwin Thomas: Biology, University of Alabama Birmingham, Birmingham, United States; Athena Thomas: Montclair State University, Montclair, United States; Jennifer Thomas: Montclair State University, Montclair, United States; Katherine Thomas: Biology, University of Louisiana at Monroe, Monroe, United States; Jasper V Thompson: Biological Sciences, Carnegie Mellon University, Pittsburgh, United States; Juri Thompson: Biology, University of Louisiana at Monroe, Monroe, United States; Justin Thompson: University of Colorado at Boulder, Boulder, United States; Kayla Thompson: Biology, University of Alabama Birmingham, Birmingham, United States; Sara Thompson: Biology, Illinois Wesleyan University, Bloomington, United States; Christopher Thompson: Purdue University, West Lafayette, United States; Elyse Thompson: The Evergreen State College, Olympia, United States; Tasha Elizabeth Thompson: Biology, University of California San Diego, La Jolla, United States; Cheng Tian: Department of Microbiology, Immunology, and Molecular Genetics, University of California, Los Angeles, Los Angeles, United States; Taylor Tibbs: Biology, Carthage College, Kenosha, United States; Melissa Tighe: Biology, Gettysburg College, Gettysburg, United States; Ainsley Timmel: Biological Sciences, Lehigh University, Bethlehem, United States; Marsha W Timmermanm: ISBT, LaSalle University, Philadelphia, United States; Jasmine Ming Jing Ting: Biology, University of California San Diego, La Jolla, United States; Vishnu Tirumala: Western Kentucky University, Bowling Green, United States; Margaret Tish: Biological Sciences, University of Pittsburgh, Pittsburgh, United States; Mehrgol Tiv: Biological Sciences, University of Pittsburgh, Pittsburgh, United States; Andrew Tobias: Montclair State University, Montclair, United States; Deborah Tobiason: Biology, Carthage College, Kenosha, United States; Benjamin P Todd: Biology, Washington University in St. Louis, St. Louis, United States; Audrey Tolbert: Department of Biology, Bucknell University, Lewisburg, United States; Marc Tollis: Biology, CUNY, Queens College, Queens, United States; Skyler Tomisato: Biology, Gonzaga University, Spokane, United States; Brianne Tomko: Biology, Gettysburg College, Gettysburg, United States; Alex Tonthat: Department of Microbiology, Immunology, and Molecular Genetics, University of California, Los Angeles, Los Angeles, United States; Michael Torello: Biology, University of Alabama Birmingham, Birmingham, United States; Melissa Torres: Biology, Smith College, Northampton, United States; Sarah Torres: Biological Sciences, University of North Texas, Denton, United States; Javier Torresdey: Biology, University of Texas at El Paso, El Paso, United States; Elizabeth Toscan: Biology, Gonzaga University, Spokane, United States; Raymond Totah: Biology, Loyola Marymount University, Los Angeles, United States; Mariama Tounkara: Ohio State University, Columbus, United States; Monica Towler: School of Science and Technology, Georgia Gwinnett College, Lawrenceville, United States; Colby Townsend: Biological Sciences, University of North Texas, Denton, United States; Cody Townsend: Biology, University of Louisiana at Monroe, Monroe, United States; Khue Vi Tran: Biology, University of California San Diego, La Jolla, United States; Quy T Tran: Biology, College of Charleston, Charleston, United States; Brandon Tran: Biology, Washington University in St. Louis, St. Louis, United States; Connie Tran: Department of Microbiology, Immunology, and Molecular Genetics, University of California, Los Angeles, Los Angeles, United States; Alexandra Treadaway: Biology, University of Louisiana at Monroe, Monroe, United States; Marissa A Tremoglie: Biology, Saint Joseph's University, Philadelphia, United States; Kathryn Trentadue: Biology, Hope College, Holland, United States; Mason Trieu: University of Colorado at Boulder, Boulder, United States; Clayton Trisler: Biology, University of Louisiana at Monroe, Monroe, United States; Daryl R Trumbo: Washington State University, Pullman, United States; Quang Truong: Biological Sciences, University of North Texas, Denton, United States; Brendan Tsai: Purdue University, West Lafayette, United States; Edwin Tsay: Biology, University of Alabama Birmingham, Birmingham, United States; Jonathan Tsay: Biology, University of California San Diego, La Jolla, United States; Chung Him Tse: Biological Sciences and Geology, Queensboro Community College, Bayside, United States; Lanya

Tseng: Biological Sciences, Carnegie Mellon University, Pittsburgh, United States; Smaragdi Tsourapa: Biological Sciences and Geology, Queensboro Community College, Bayside, United States; Jonathan Tucker: The Evergreen State College, Olympia, United States; Joshua Tull: Montclair State University, Montclair, United States; Jacqueline C Tully: Biology, College of Charleston, Charleston, United States; Jessica Turen: University of Florida, Gainsville, United States; Chelsey Turner: Biology, University of Wisconsin-River Falls, River Falls, United States; Eric Turner: University of Florida, Gainsville, United States; Paul Turner: Biology, Culver-Stockton College, Canton, United States; Sydnie Turner: Xavier University of Louisiana, New Orleans, United States; Beth Tuttle: Biology, University of Alabama Birmingham, Birmingham, United States; Cassandra Tuttman: Biological Sciences, Lehigh University, Bethlehem, United States; Joy L Twentyman: Biology, Washington University in St. Louis, St. Louis, United States; Dionne Ubungen: Montclair State University, Montclair, United States; Dhruva Rajesh Udani: Biology, University of California San Diego, La Jolla, United States; Neha Udayakumar: Biology, University of Alabama Birmingham, Birmingham, United States; Yakntoro Udoumoh: Biology, Howard College, Washington, DC, United States; William Ueckermann: Biology, Gettysburg College, Gettysburg, United States; Sharon Uhder: Biology, Gonzaga University, Spokane, United States; Kristin Ullberg: Biology, Illinois Wesleyan University, Bloomington, United States; Hannah Ullery: Washington State University, Pullman, United States; Miranda Ulmer: Biology, Hope College, Holland, United States; Kendrick Underwood: Biology, University of Louisiana at Monroe, Monroe, United States; Tyler J Uppstrom: Biology, Washington University in St. Louis, St. Louis, United States; Carl Urbinati: Biology, Loyola Marymount and University of Detroit, Los Angeles, United States; Eduardo Urias: Biology, University of Texas at El Paso, El Paso, United States; Akif Uzman: Natural Sciences, University of Houston-Downtown, Houston, United States; Steven Valentino: University of Maine, Honors College, Orono, United States; Lourdes Valenzuela: University of California Santa Cruz, Santa Cruz, United States; Andrew Valesano: Biology, Hope College, Holland, United States; Braeden Van Deynze: Biology, Gonzaga University, Spokane, United States; David Van Doren: Biology, Gettysburg College, Gettysburg, United States; Dylan van Krieken: University of California Santa Cruz, Santa Cruz, United States; Vanessa Van Tongel: Biology and Chemistry, Nyack College, Nyack, United States; Rebecca Van Zanen: Biology, Calvin College, Grand Rapids, United States; Alexis Vance: Xavier University of Louisiana, New Orleans, United States; Peter VandeHaar: Biology, Calvin College, Grand Rapids, United States; Amy VanderStoep: Biology, Hope College, Holland, United States; Zoe Vann: University of Colorado at Boulder, Boulder, United States; Cassie VanWynen: Biology, Hope College, Holland, United States; Corina Vasquez: Biology, Gonzaga University, Spokane, United States; Josue Vasquez: Biology, Calvin College, Grand Rapids, United States; Gwendolyn S Vasquez: Biology, Merrimack College, North Andover, United States; Tremain Vass: biology, North Carolina Central University, Durham, United States; Anastasia Dmitrievna Vavilina: Biology, University of California San Diego, La Jolla, United States; Edwin Vazquez: Biology, University of Puerto Rico - Cayey, Cayey, United States; Alyssa Vecchio: Biological Sciences, Lehigh University, Bethlehem, United States; Quinn C Vega: Montclair State University, Montclair, United States; Rahulsimham Vegesna: Biology, University of Texas at El Paso, El Paso, United States; Paola Vela: Biology, University of Texas at El Paso, El Paso, United States; Rebecca Velasco: Biology, Gonzaga University, Spokane, United States; Kelsey Veldkamp: Biology, Calvin College, Grand Rapids, United States; Mariano Velez: ISBT, LaSalle University, Philadelphia, United States; Kara Venema: Biology, Calvin College, Grand Rapids, United States; Marissa Venero: University of Florida, Gainsville, United States; Yaamini Ranjani Venkataraman: Biology, University of California San Diego, La Jolla, United States; Vaidehi Venkataraman: University of Colorado at Boulder, Boulder, United States; Kristen Verdoorn: Biology, Carthage College, Kenosha, United States; Jake Verduzco: Biology, College of St. Scholastica, Duluth, United States; Daniel Vessells: Biology, Hope College, Holland, United States; Breeze Victor: Department of Biology, Bucknell University, Lewisburg, United States; McElroy Victoria: University of California Santa Cruz, Santa Cruz, United States; Steven Vigil Roach: University of Colorado at Boulder, Boulder, United States; Janahan Vijanderan: Department of Microbiology, Immunology, and Molecular Genetics, University of California, Los Angeles, Los Angeles, United States; Ammu Vijayakumar: Microbiology and Biotechnology, North Carolina State University, Raleigh, United States; Aishwarya Vijayan: University of Florida, Gainsville, United States; Albert Vill: Biology, Gettysburg College, Gettysburg, United States; Erika Villa: Biology, University of Texas at El Paso, El Paso, United States; Sabra Villa: University of California Santa Cruz, Santa Cruz, United States; Maria

Virginia Villadiego-Punto: Biological Sciences and Geology, Queensboro Community College, Bayside, United States; Jose Villagomez: Montclair State University, Montclair, United States; Ana Villanueva-Pereira: Biology, University of Puerto Rico - Cayey, Cayey, United States; Rachel Villegas: Biological Sciences, University of North Texas, Denton, United States; William Villella: Department of Microbiology, Immunology, and Molecular Genetics, University of California, Los Angeles, Los Angeles, United States; Yvette Villero Martinez: University of California Santa Cruz, Santa Cruz, United States; Kelly Vining: Molecular and Cell Biology Program, Oregon State University, Corvallis, United States; Meredith Virk: Biology, Gonzaga University, Spokane, United States; Bhula Vishal: Biology, Loyola Marymount University, Los Angeles, United States; Nikita Vispute: Biological Sciences, University of Pittsburgh, Pittsburgh, United States; Victoria Viveen: Biology, Hope College, Holland, United States; Melody Vo: Biological Sciences, University of North Texas, Denton, United States; Quynh Vo: Biological Sciences, University of North Texas, Denton, United States; Stephanie Vu: Department of Microbiology, Immunology, and Molecular Genetics, University of California, Los Angeles, Los Angeles, United States; John Vu: Biology, University of Louisiana at Monroe, Monroe, United States; Jamie Vulgamore: Natural Sciences, Del Mar College, Corpus Christi, United States; Chiraag Vyas: University of California Santa Cruz, Santa Cruz, United States; Kristen Wade: Virginia Commonwealth University, Richmond, United States; Marisa H Wagner: Biology, Saint Joseph's University, Philadelphia, United States; Shalia Wagner: Xavier University of Louisiana, New Orleans, United States; Ryan Wagner: Purdue University, West Lafayette, United States; Abigail Waidelich: Ohio State University, Columbus, United States; Andrew Wakefield: Microbiology, Miami University, Oxford, United States; Isaac Wakiro: Biology, College of St. Scholastica, Duluth, United States; Tyler Walburn: Biological Sciences, University of Pittsburgh, Pittsburgh, United States; Erin Walch: Western Kentucky University, Bowling Green, United States; Gretchen Walch: Western Kentucky University, Bowling Green, United States; Angie Waldron: The Evergreen State College, Olympia, United States; Loni Walker: Biology, Illinois Wesleyan University, Bloomington, United States; Savannah Walker: Microbiology, Miami University, Oxford, United States; Brenden Wall: Biology, Illinois Wesleyan University, Bloomington, United States; Jasmine C Wall: Biology, Saint Joseph's University, Philadelphia, United States; Lauren Beth Waller: Biology, University of California San Diego, La Jolla, United States; Rachel Walstead: Virginia Commonwealth University, Richmond, United States; Rachel S Walter: Honors Program, Florida Gulf Coast University, Fort Myers, United States; Dustin Walter: Biology, Ouachita Baptist University, Arkadelphia, United States; Andy Yixun Wang: Biology, University of California San Diego, La Jolla, United States; Da Wang: Biology, Illinois Wesleyan University, Bloomington, United States; Jay-Shing Wang: University of Florida, Gainesville, United States; Judy Jingxuan Wang: Biology, Washington University in St. Louis, St. Louis, United States; Hao-Yi Wang: Virginia Commonwealth University, Richmond, United States; Yuqi Wang: Biology, Washington University in St. Louis, St. Louis, United States; Vassie C Ware: Biological Sciences, Lehigh University, Bethlehem, United States; John Robert Warner: Biology, University of Louisiana at Monroe, Monroe, United States; David Warren: Biology and Medicine, Brown University, Providence, United States; Samantha Warwar: Ohio State University, Columbus, United States; Jacqueline M Washington: Biology and Chemistry, Nyack College, Nyack, United States; William Waterstreet: Purdue University, West Lafayette, United States; Colby Watkins: Biological Sciences, University of North Texas, Denton, United States; Laura C Watkins: Biology, Washington University in St. Louis, St. Louis, United States; Erin L Waugaman: Division of Natural and Health Sciences, Seton Hill University, Greensburg, United States; Brent Webb: Western Kentucky University, Bowling Green, United States; Jessica Webb: Biology, Culver-Stockton College, Canton, United States; Marc Webb: biology, North Carolina Central University, Durham, United States; Laurence Webb: Ohio State University, Columbus, United States; Jeffrey Wei: Biology, Washington University in St. Louis, St. Louis, United States; Christine Weir: University of California Santa Cruz, Santa Cruz, United States; Emilie Weisser: Biology, Washington University in St. Louis, St. Louis, United States; Holly Welfley: Ohio State University, Columbus, United States; Marie Wells: Biology, Spelman College, Atlanta, United States; Joshua Welsch: Biology, Hope College, Holland, United States; Braden Wenndt: Purdue University, West Lafayette, United States; John T Wertz: Biology, Calvin College, Grand Rapids, United States; Aliah S West: ISBT, LaSalle University, Philadelphia, United States; Daniel Westholm: Biology, College of St. Scholastica, Duluth, United States; Kathryn Weston: University of Florida, Gainesville, United States; Kathleen A Weston-Hafer: Biology, Washington University in St. Louis, St. Louis, United States; Victoria Westra: Biology, Calvin College, Grand

Rapids, United States; Abigail Whalen: Biology, Trinity College, Hartford, United States; Ellen Wheeler: Providence College, Providence, United States; James Wherley: Biology, Washington University in St. Louis, St. Louis, United States; Emily Whitaker: University of Maine, Honors College, Orono, United States; Dakota White: Biology, Gonzaga University, Spokane, United States; Laura L White: Department of Biology, Baylor University, Waco, United States; Lamanuel White: biology, North Carolina Central University, Durham, United States; Rebekah K White: Department of Biology, Baylor University, Waco, United States; Xander White: Biological Sciences, University of North Texas, Denton, United States; Stephen Whitfield: Biology, Illinois Wesleyan University, Bloomington, United States; Carmen Wickware: Purdue University, West Lafayette, United States; Peter Widitz: Biology, Calvin College, Grand Rapids, United States; Allison MD Wiedemeier: Biology, University of Louisiana at Monroe, Monroe, United States; Sophia R Wienbar: Biological Sciences, Carnegie Mellon University, Pittsburgh, United States; Dion Wigfall: ISBT, LaSalle University, Philadelphia, United States; Katherine Wikholm: Biology, Loyola Marymount University, Los Angeles, United States; Luke Wilde: Biology, Gonzaga University, Spokane, United States; William Wilde: Biology, Gonzaga University, Spokane, United States; Adrienne Wilen: Washington State University, Pullman, United States; Abigail Wilhelm: Biology, Culver-Stockton College, Canton, United States; Garrett Wilkerson: Biology, University of Louisiana at Monroe, Monroe, United States; Kellyn Wilkes: Biological Sciences, University of Pittsburgh, Pittsburgh, United States; Chantelle Willette: Biological Sciences, University of North Texas, Denton, United States; Chandler Williams: Morehouse College, Atlanta, United States; Ciera Williams: biology, North Carolina Central University, Durham, United States; Devin M Williams: Biology, Spelman College, Atlanta, United States; Drake Williams: Department of Microbiology, Immunology, and Molecular Genetics, University of California, Los Angeles, Los Angeles, United States; Kristina Williams: Biology, Loyola Marymount University, Los Angeles, United States; Lauren H Williams: Biological Sciences, Carnegie Mellon University, Pittsburgh, United States; Madelaine Williams: University of Colorado at Boulder, Boulder, United States; Richard Williams: Morehouse College, Atlanta, United States; Tyler Williams: Environmental and Biological Science, University of Maine, Machias, Machias, United States; Sara Williams: Biology, Ouachita Baptist University, Arkadelphia, United States; Kurt E Williamson: Biology, College of William and Mary, Williamsburg, United States; Sarah M Williamson: Virginia Commonwealth University, Richmond, United States; Whitney Willis: Biology, Ouachita Baptist University, Arkadelphia, United States; Monique Willoughby: Biological Sciences and Geology, Queensboro Community College, Bayside, United States; Alix C Wilson: Department of Biology, Baylor University, Waco, United States; Christine R Wilson: Department of Biology, Baylor University, Waco, United States; Elisa Wilson: Biology, Gonzaga University, Spokane, United States; John Wilson: Environmental and Biological Science, University of Maine, Machias, Machias, United States; Trevor Wilson: Biology, College of Idaho, Caldwell, United States; Lauren Wilson: Biology, Gettysburg College, Gettysburg, United States; Tyriana Wilson: Biology, University of Louisiana at Monroe, Monroe, United States; Justin Wimberly: Biology, University of Alabama Birmingham, Birmingham, United States; Danielle Winders: Ohio State University, Columbus, United States; Shane Wing: Biology, Gonzaga University, Spokane, United States; Sarah Winokur: Biology, Smith College, Northampton, United States; Victoria Marie Winslow: Biology, University of California San Diego, La Jolla, United States; Hannah S Wirtshafter: Biological Sciences, Carnegie Mellon University, Pittsburgh, United States; Eric Witherspoon: Morehouse College, Atlanta, United States; Evan Witt: Biology, Washington University in St. Louis, St. Louis, United States; Donna Wodarski: Science, Cabrini College, Radnor, United States; Meredith Wojcik: Microbiology and Biotechnology, North Carolina State University, Raleigh, United States; Victoria Wolf: Biology, Smith College, Northampton, United States; Brooke Wolff: Microbiology and Biotechnology, North Carolina State University, Raleigh, United States; Cody Wolterman: Biology, Loyola Marymount University, Los Angeles, United States; Michael J Wolyniak: Biology, Hampden-Sydney College, Farmville, United States; Adrienne Evelyn Wong: Biology, University of California San Diego, La Jolla, United States; Chung Ki Wong: Biology, Loyola Marymount University, Los Angeles, United States; Stephanie Wood: University of Maine, Honors College, Orono, United States; Mitchell Woodford: Biology and Chemistry, Nyack College, Nyack, United States; Andrew Woodruff: Microbiology, Miami University, Oxford, United States; Avery Woods: Morehouse College, Atlanta, United States; Dayton Wooldridge: Washington State University, Pullman, United States; Michael Woolford: Biological Sciences, Carnegie Mellon University, Pittsburgh, United States; Cassandra Worner: Microbiology, Miami University, Oxford, United States; Rebecka Worrell:

Biology, Smith College, Northampton, United States; Michael J Wozny: Honors Program, Florida Gulf Coast University, Fort Myers, United States; Nina Wren: Biology, Smith College, Northampton, United States; Brigham A Wright: Microbiology and Molecular Biology, Brigham Young University, Provo, United States; Bianca Wright: Biology, University of Texas at El Paso, El Paso, United States; Heather E Wright: Biology, College of Charleston, Charleston, United States; Spencer Wright: Western Kentucky University, Bowling Green, United States; Shelby Wright: Biology, University of Louisiana at Monroe, Monroe, United States; Kathryn Wrobel: Biology, Calvin College, Grand Rapids, United States; Cynthia E Wu: Biology, University of California San Diego, La Jolla, United States; Hao Wu: Department of Biology, Baylor University, Waco, United States; Kit Wu: Biology, University of California San Diego, La Jolla, United States; Xiangying Wu: Biological Sciences and Geology, Queensboro Community College, Bayside, United States; Jiewei Wu: Purdue University, West Lafayette, United States; Yi Shuan Wu: Biology, University of California San Diego, La Jolla, United States; Jalyn Wurm: Biology, Nebraska Wesleyan University, Lincoln, Nebraska, United States; Shacaria Wyke: biology, North Carolina Central University, Durham, United States; Kevin Wyllie: Biology, Loyola Marymount University, Los Angeles, United States; Kristen Wymore: Biological Sciences, Lehigh University, Bethlehem, United States; Christian J Xander: Biology, Saint Joseph's University, Philadelphia, United States; Xiao Xiao: Biological Sciences, University of Pittsburgh, Pittsburgh, United States; Helen Xun: Biology, Gonzaga University, Spokane, United States; Anastaciya Yakovenko: University of Florida, Gainsville, United States; Keianne Dale Yamada: Biology, University of California San Diego, La Jolla, United States; Kyoko Yamaguchi: Department of Microbiology, Immunology, and Molecular Genetics, University of California, Los Angeles, Los Angeles, United States; Keying Yan: University of Colorado at Boulder, Boulder, United States; Jonathan Yang: Department of Microbiology, Immunology, and Molecular Genetics, University of California, Los Angeles, Los Angeles, United States; Sophia Yang: Microbiology and Biotechnology, North Carolina State University, Raleigh, United States; Tsering Yangzom: Biological Sciences and Geology, Queensboro Community College, Bayside, United States; Rayce D Yanney: Department of Biology, Baylor University, Waco, United States; Tyler Yates: University of Colorado at Boulder, Boulder, United States; Chen Ye: Biology and Medicine, Brown University, Providence, United States; Brandon Yee: University of California Santa Cruz, Santa Cruz, United States; Sarah Yeend: Biology, Gonzaga University, Spokane, United States; Daniel Yehdego: Biology, University of Texas at El Paso, El Paso, United States; Justin Yen: Montclair State University, Montclair, United States; Benjamin Yoder: Biological Sciences, University of Pittsburgh, Pittsburgh, United States; Amber Yohn: Montclair State University, Montclair, United States; Priscilla Yong: Biology, Smith College, Northampton, United States; Santiago Yori: Honors Program, Florida Gulf Coast University, Fort Myers, United States; Alicia Young: Biology, University of Texas at El Paso, El Paso, United States; Elizabeth Young: Biology, Gonzaga University, Spokane, United States; Lauren K Young: Biology, Saint Joseph's University, Philadelphia, United States; Hannah Marie Youngwirth: Biology, University of California San Diego, La Jolla, United States; Hussain Yousaf: University of Florida, Gainsville, United States; Jullie Yu: Department of Microbiology, Immunology, and Molecular Genetics, University of California, Los Angeles, Los Angeles, United States; Victor Yu: Biological Sciences, University of Pittsburgh, Pittsburgh, United States; Eric Yu: Biology, Calvin College, Grand Rapids, United States; Stefan Yu: Biology, Washington University in St. Louis, St. Louis, United States; Han Yuan: Biology, Washington University in St. Louis, St. Louis, United States; Christine Zabel: Microbiology and Biotechnology, North Carolina State University, Raleigh, United States; Joleen Zackowski: Virginia Commonwealth University, Richmond, United States; David A Zaidins: Biological Sciences, Carnegie Mellon University, Pittsburgh, United States; Alice Zalan: Biology, University of California San Diego, La Jolla, United States; Stephanie Zamora: Biology, Loyola Marymount University, Los Angeles, United States; Rachel Zarchy: Biology, Illinois Wesleyan University, Bloomington, United States; Michael V Zavorski: ISBT, LaSalle University, Philadelphia, United States; Mariam Zayed: Xavier University of Louisiana, New Orleans, United States; Franck Zeba: Environmental and Biological Science, University of Maine, Machias, Machias, United States; Gerard Zegers: Environmental and Biological Science, University of Maine, Machias, Machias, United States; Michael Zehner: University of Colorado at Boulder, Boulder, United States; Lucas Zellmer: Biology, University of Wisconsin-River Falls, River Falls, United States; Manhao Zeng: Biology, Gettysburg College, Gettysburg, United States; Bruce H Zhang: Biology, University of California San Diego, La Jolla, United States; Bo Zhang: Biology, Washington University in St. Louis, St. Louis, United States;

Carolyn Min Zhang: Biology, University of California San Diego, La Jolla, United States; Daiyuan Zhang: Natural Sciences, Del Mar College, Corpus Christi, United States; Hairong Zhang: Biology, Washington University in St. Louis, St. Louis, United States; James Zhang: Biology, University of California San Diego, La Jolla, United States; Junhao Zhang: Department of Microbiology, Immunology, and Molecular Genetics, University of California, Los Angeles, Los Angeles, United States; Mitchell Jia Zhao: Biology, University of California San Diego, La Jolla, United States; Alec Zimmer: Biology, Washington University in St. Louis, St. Louis, United States; Zachary Zimmer: Biology, Illinois Wesleyan University, Bloomington, United States; Anastasia M Zimmerman: Biology, College of Charleston, Charleston, United States; Sarah Zimmermann: University of Colorado at Boulder, Boulder, United States; Tai Zollars: Biology, Nebraska Wesleyan University, Lincoln, Nebraska, United States; Melina Y Zuniga: Biology, Spelman College, Atlanta, United States;

**Phage Hunters Integrating Research and Education**

Amma Ababio: Department of Biological Sciences, University of Pittsburgh, Pittsburgh, United States; Jamil Alhassan: Department of Biological Sciences, University of Pittsburgh, Pittsburgh, United States; Zohair Azmi: Department of Biological Sciences, University of Pittsburgh, Pittsburgh, United States; Michelle Boyle: Department of Biological Sciences, University of Pittsburgh, Pittsburgh, United States; April Burch: Department of Biological Sciences, University of Pittsburgh, Pittsburgh, United States; Stephen Canton: Department of Biological Sciences, University of Pittsburgh, Pittsburgh, United States; Alexandra Cathcart: Department of Biological Sciences, University of Pittsburgh, Pittsburgh, United States; Brittany Dey: Department of Biological Sciences, University of Pittsburgh, Pittsburgh, United States; CourtneyBeth Dohl: Department of Biological Sciences, University of Pittsburgh, Pittsburgh, United States; Samantha Eppinger: Department of Biological Sciences, University of Pittsburgh, Pittsburgh, United States; Emma Fisher: Department of Biological Sciences, University of Pittsburgh, Pittsburgh, United States; Rodrigo Gonzalez: Department of Biological Sciences, University of Pittsburgh, Pittsburgh, United States; Forrest Guilfoile: Department of Biological Sciences, University of Pittsburgh, Pittsburgh, United States; David Hauser: Department of Biological Sciences, University of Pittsburgh, Pittsburgh, United States; Christina Hwang: Department of Biological Sciences, University of Pittsburgh, Pittsburgh, United States; Saikrishna Kothapalli: Department of Biological Sciences, University of Pittsburgh, Pittsburgh, United States; Darwin Leuba: Department of Biological Sciences, University of Pittsburgh, Pittsburgh, United States; Kohana Leuba: Department of Biological Sciences, University of Pittsburgh, Pittsburgh, United States; Sequoia Leuba: Department of Biological Sciences, University of Pittsburgh, Pittsburgh, United States; Austin Li: Department of Biological Sciences, University of Pittsburgh, Pittsburgh, United States; Edeline Loh: Department of Biological Sciences, University of Pittsburgh, Pittsburgh, United States; Matthew Luchansky: Department of Biological Sciences, University of Pittsburgh, Pittsburgh, United States; Nathaniel MacKenzie: Department of Biological Sciences, University of Pittsburgh, Pittsburgh, United States; Kaitlin Mitchell: Department of Biological Sciences, University of Pittsburgh, Pittsburgh, United States; Matthew Olm: Department of Biological Sciences, University of Pittsburgh, Pittsburgh, United States; Yein Park: Department of Biological Sciences, University of Pittsburgh, Pittsburgh, United States; Terence Parker: Department of Biological Sciences, University of Pittsburgh, Pittsburgh, United States; Kaitlin Price: Department of Biological Sciences, University of Pittsburgh, Pittsburgh, United States; Elina Roine: Department of Biological Sciences, University of Pittsburgh, Pittsburgh, United States; Rachel Rush: Department of Biological Sciences, University of Pittsburgh, Pittsburgh, United States; Paul Salamanca: Department of Biological Sciences, University of Pittsburgh, Pittsburgh, United States; Jennifer Schaub: Department of Biological Sciences, University of Pittsburgh, Pittsburgh, United States; Lauren Schmidt: Department of Biological Sciences, University of Pittsburgh, Pittsburgh, United States; Victoria Schneider: Department of Biological Sciences, University of Pittsburgh, Pittsburgh, United States; Ana Sencilo: Department of Biological Sciences, University of Pittsburgh, Pittsburgh, United States; Emilee Shine: Department of Biological Sciences, University of Pittsburgh, Pittsburgh, United States; Kailey Slavik: Department of Biological Sciences, University of Pittsburgh, Pittsburgh, United States; Shahwar Tariq: Department of Biological Sciences, University of Pittsburgh, Pittsburgh, United States; Waleed Tariq: Department of Biological Sciences, University of Pittsburgh, Pittsburgh, United States; Enoch Tse: Department of Biological Sciences, University of Pittsburgh, Pittsburgh, United States; Kathy Van Hoeck: Department of Biological Sciences, University of Pittsburgh,

Pittsburgh, United States; Alex Waldherr: Department of Biological Sciences, University of Pittsburgh, Pittsburgh, United States; Ana Wan: Department of Biological Sciences, University of Pittsburgh, Pittsburgh, United States; Brett Weingart: Department of Biological Sciences, University of Pittsburgh, Pittsburgh, United States; Albin Wells: Department of Biological Sciences, University of Pittsburgh, Pittsburgh, United States; Jakob Wells: Department of Biological Sciences, University of Pittsburgh, Pittsburgh, United States; Philip Williams: Department of Biological Sciences, University of Pittsburgh, Pittsburgh, United States; Randi Wilson: Department of Biological Sciences, University of Pittsburgh, Pittsburgh, United States; Gabriel Winbush: Department of Biological Sciences, University of Pittsburgh, Pittsburgh, United States; Amanda Yurick: Department of Biological Sciences, University of Pittsburgh, Pittsburgh, United States;

**Mycobacterial Genetics Course**

Naazneen Adam: University of KwaZulu-Natal, Durban, South Africa; Ashmita Arjoon: University of KwaZulu-Natal, Durban, South Africa; Lugani Bengani: University of KwaZulu-Natal, Durban, South Africa; Rooksana Carim: University of KwaZulu-Natal, Durban, South Africa; Nafiisah Chotun: University of KwaZulu-Natal, Durban, South Africa; Navisha Dookie: University of KwaZulu-Natal, Durban, South Africa; Nabila Essack: University of KwaZulu-Natal, Durban, South Africa; Karnishree Govender: University of KwaZulu-Natal, Durban, South Africa; Viveshree Govender: University of KwaZulu-Natal, Durban, South Africa; Nandini Gramoney: University of KwaZulu-Natal, Durban, South Africa; Jessica Hunter: University of KwaZulu-Natal, Durban, South Africa; Chernoh Jalloh: University of KwaZulu-Natal, Durban, South Africa; Afsana Kajee: University of KwaZulu-Natal, Durban, South Africa; Nathan Kieswetter: University of KwaZulu-Natal, Durban, South Africa; Michelle H Larsen: Microbiology and Immunology, Albert Einstein College of Medicine, Bronx, United States; Jared Mackenzie: University of KwaZulu-Natal, Durban, South Africa; Fiona Maiyo: University of KwaZulu-Natal, Durban, South Africa; Gugulethu Masondo: University of KwaZulu-Natal, Durban, South Africa; Mpilwenhle Mbanjwa: University of KwaZulu-Natal, Durban, South Africa; Yanga Mdleleni: University of KwaZulu-Natal, Durban, South Africa; Khanyisile Mngomezulu: University of KwaZulu-Natal, Durban, South Africa; Katherine Moccia: University of KwaZulu-Natal, Durban, South Africa; Chantal Molechan: University of KwaZulu-Natal, Durban, South Africa; Odessa Moodley: University of KwaZulu-Natal, Durban, South Africa; Zama Msibi: University of KwaZulu-Natal, Durban, South Africa; Tessa Naido: University of KwaZulu-Natal, Durban, South Africa; Anand Naranbhai: University of KwaZulu-Natal, Durban, South Africa; Vivek Naranbhai: University of KwaZulu-Natal, Durban, South Africa; Nomfundo Ncobeni: University of KwaZulu-Natal, Durban, South Africa; Fortunate Ndlandla: University of KwaZulu-Natal, Durban, South Africa; Bridget Nduna: University of KwaZulu-Natal, Durban, South Africa; Silindile Ngobese: University of KwaZulu-Natal, Durban, South Africa; Nokonwaba Nkondlo: University of KwaZulu-Natal, Durban, South Africa; Shirwin Pillay: University of KwaZulu-Natal, Durban, South Africa; Yathisha Ramlakhan: University of KwaZulu-Natal, Durban, South Africa; Nicole Reddy: University of KwaZulu-Natal, Durban, South Africa; Eric J Rubin: Department of Immunology and Infectious Diseases, Harvard School of Public Health, United States; Neo Sehloko: University of KwaZulu-Natal, Durban, South Africa; Shilisha Shanmugam: University of KwaZulu-Natal, Durban, South Africa; Sarisha Singh: University of KwaZulu-Natal, Durban, South Africa; Melisha Sukkhu: University of KwaZulu-Natal, Durban, South Africa; Po-Cheng Tang: University of KwaZulu-Natal, Durban, South Africa

## Funding

| Funder | Grant reference | Author |
| --- | --- | --- |
| Howard Hughes Medical Institute (HHMI) | 54308198 | Graham F Hatfull |
| National Institutes of Health (NIH) | GM51975 | Graham F Hatfull |
| Howard Hughes Medical Institute (HHMI) | 52007054 | Graham F Hatfull |
| Brigham Young University | | Sandra Burnett |
| Cabrini College | | David Dunbar |
| National Institutes of Health—INBRE | GM103408 | R Luke Daniels |

| Funder | Grant reference | Author |
| --- | --- | --- |
| National Science Foundation (NSF) | | John Dennehy |
| Queens College | | John Dennehy |
| Lehigh University | | Vassie Ware |
| Merrimack College | | Janine LeBlanc-Straceski |
| National Institutes of Health (NIH) | GM094712 | Nicanor Austriaco |
| National Institutes of Health—INBRE | GM103430 | Kathleen Cornely |
| Davis Foundational Grant | | Kathleen Cornely |
| Providence College | | Kathleen Cornely |
| St. Joseph's University | | Christina King Smith |
| University of Houston, Downtown | | Rachna Sadana |
| University of Maine, Honors College | | Keith Hutchinson |
| National Institutes of Health (NIH) | GM1003423 | Keith Hutchinson |
| Howard Hughes Medical Institute (HHMI) | | Michael Rubin, Kirk Anders, SEA-PHAGES program |
| University of Puerto Rico | | Michael Rubin |
| University of Wisconsin, River Falls | | Karen Klyczek |
| Western Kentucky University | | Claire Rinehart |
| Gatton Academy of Science and Mathematics | | Rodney King |
| Georgia College | | Indiren Pillay |
| Del Mar College | | John Hatherill |
| Miami University | | Iddo Friedberg |
| National Science Foundation (NSF) | DUE-1205059 | John Hatherill |
| National Science Foundation (NSF) | ABI-1146960 | Iddo Friedberg |
| Howard Hughes Medical Institute (HHMI) | 52007572 | SK Ireland |
| Doris Duke Charitable Foundation | | Michelle Larsen |
| Gonzaga University | | Kirk Anders |
| National Science Foundation (NSF) | DUE-1245778 | Kirk Anders |

David Asai, Kevin Bradley, and Lucia Barker (formerly) are (or were) employees of Howard Hughes Medical Institute who also provided support for the SEA-PHAGES and PHIRE programs. DA, KB, and LB contributed to the design of the programs and the systems for data collection.

## Author contributions

WHP, DJ-S, Conception and design, Acquisition of data, Analysis and interpretation of data, Drafting or revising the article, Contributed unpublished essential data or reagents; CAB, DAR, DJA, WRJ, RWH, JGL, GFH, Conception and design, Acquisition of data, Analysis and interpretation of data, Drafting or revising the article; SGC, Acquisition of data, Analysis and interpretation of data, Contributed unpublished essential data or reagents

## Additional files

### Supplementary files

• Supplementary file 1. List of 627 sequenced mycobacteriophages and cluster designations.

• Supplementary file 2. Full list of group author details.

### Major datasets

The following datasets were generated:

| Author(s) | Year | Dataset title | Dataset ID and/or URL | Database, license, and accessibility information |
|---|---|---|---|---|
| Russell D, Hatfull G | 2015 | Mycobacteriophage database | http://phagesdb.org | Publicly available at the Mycobacteriophage Database (http://phagesdb.org). |
| Bowman C, Cresawn S, Hatfull G | 2014 | Mykobacteriophage_627 | http://phamerator.webfactional.com/databases_Hatfull | Accessed via the program Phamerator, publicly available at http://phagesdb.org/Phamerator/faq/. |

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
