## [Decision Letter]

Thank you for sending your work entitled “Whole genome comparison of a large collection of mycobacteriophages reveals a continuum of phage genetic diversity” for consideration at *eLife*. Your article has been favorably evaluated by Richard Losick (Senior editor) and three reviewers, one of whom, Roberto Kolter, is a member of our Board of Reviewing Editors.

The Reviewing editor and the other reviewers discussed their comments before we reached this decision, and the Reviewing editor has assembled the following comments to help you prepare a revised submission.

We all felt that the manuscript has two important merits. First, this represents the first large scale phage genome sequencing effort of a collection of phages that infect a single host. The biological insights are important in that the genetic diversity extant in these 600+ genomes is seen as a continuum indicative of an active coming and going of genes within the large number of phages that can infect an individual host. Despite the remarkable diversity, it was possible to cluster the phage genomes and begin to approach a phage phylogeny, albeit a reticulate (networked) one rather than a straightforward branched one. Second, this work is the result of research largely carried out by college freshmen assembled in a structured yet geographically scattered science education program. This serves as an important example of how to carry out scientific research where students have a sense of ownership of the work.

However, we all felt that there were some issues that will have to be addressed in a revision:

1) The writing of the manuscript has two “personalities”. All of the descriptions of the educational aspects/benefits of the work were wonderfully written and could be followed with utmost ease due to the clarity of the written style. In contrast, the “comparative genomics” aspect of the manuscript was too littered with jargon and lack of rigor in defining terms that it made it quite difficult to follow for those not deeply steeped in the field. Thus, we would urge the senior author to take a “major overhaul” approach and thoroughly re-write the comparative genomics sections to make them as accessible to a broad scientific audience as the educational sections.

2) In the manuscript, the genomic data are presented, but in the webpage there are also images of each one of the bacteriophages, this information is valuable and should also be mentioned in this manuscript in order to inform the reader about the scope of the study.

3) Given the transcendence of the findings it is necessary that the authors provide detailed methods about the sequencing assembly process. The architecture of phage genomes and the mosaicism are features that could be influenced by the read assembly methods, for example de novo versus mapping.

4) One general disappointment was that no effort was made to identify and discuss the functions of protein families. With all these students working on these phages, it should be possible to identify many functions for families and also identify which families are associated with each other. While we are not asking for additional direct experimentation, some discussion as to possible function or routes to identifying such functions would be welcome.

---

## [Author Response]

1) The writing of the manuscript has two “personalities”. All of the descriptions of the educational aspects/benefits of the work were wonderfully written and could be followed with utmost ease due to the clarity of the written style. In contrast, the “comparative genomics” aspect of the manuscript was too littered with jargon and lack of rigor in defining terms that it made it quite difficult to follow for those not deeply steeped in the field. Thus, we would urge the senior author to take a “major overhaul” approach and thoroughly re-write the comparative genomics sections to make them as accessible to a broad scientific audience as the educational sections.

We recognize some of the issues that might have given rise to some lack of clarity, especially in the Introduction, in regard to the comparative genomics. Especially in the use of terms such as ‘cluster’, ‘population’, ‘group’, ‘type’, etc. We have revised the Abstract and Introduction with a major overhaul, and we think the revised version is much improved.

2) In the manuscript, the genomic data are presented, but in the webpage there are also images of each one of the bacteriophages, this information is valuable and should also be mentioned in this manuscript in order to inform the reader about the scope of the study.

We have included commentary on phage morphologies.

3) Given the transcendence of the findings it is necessary that the authors provide detailed methods about the sequencing assembly process. The architecture of phage genomes and the mosaicism are features that could be influenced by the read assembly methods, for example de novo versus mapping.

Details of sequencing methodologies are included in the Methods. Most importantly perhaps, all genomes are de novo assembled, and not scaffolded.

4) One general disappointment was that no effort was made to identify and discuss the functions of protein families. With all these students working on these phages, it should be possible to identify many functions for families and also identify which families are associated with each other. While we are not asking for additional direct experimentation, some discussion as to possible function or routes to identifying such functions would be welcome.

The manuscript deliberately focuses on specific aspects of the mycobacteriophage phage population as viewed through comparative genomics, coupled with the science education context with which these data are generated. Of course we have predicted numerous phage gene functions, much of which is done by student annotators. Providing a detailed description of this is simply way beyond the scope of this paper. We have, however, included a general comment about gene functions, and the high proportion of genes of unknown function.